# A Regime-Aware Trajectory Prediction Framework for 1000+ Systems Biology Models

**Heng Rao** [1]  **Jason Zipeng Zhang** [2]  **Yu Gu** [1]  **Zhenghao Liu** [1]  **Ge Yu** [1]  **Jeffrey Su** [3]  **Yang Cao** [4]  **Fan Yang** [2]  **Minghan Chen** [2]

## Abstract

Predicting long-horizon trajectories of biological dynamical systems remains challenging due to substantial system heterogeneity. Most existing machine learning approaches are system-specific, requiring retraining for each new system and exhibiting limited generalization across distinct biological regimes. To address this limitation, we create a large-scale benchmark of over 1,000 ODE-based systems biology models spanning diverse organisms, biological processes, and dynamical behaviors. Building on this benchmark, we propose a regime-aware trajectory prediction framework that enables cross-system generalization and uncertainty quantification for unseen systems. Our approach introduces structured initial states derived from biological regime priors, such as growth trends and oscillatory rhythms, into conditional flow matching, replacing the standard Gaussian source distribution. We provide theoretical justification for this initialization and empirically demonstrate state-of-the-art accuracy (31% MAE reduction), well-calibrated uncertainty (17% CRPS improvement), and efficient long-horizon inference across the benchmark.

## 1. Introduction

Predicting trajectories of biological dynamical systems is fundamental for understanding complex biological pro-

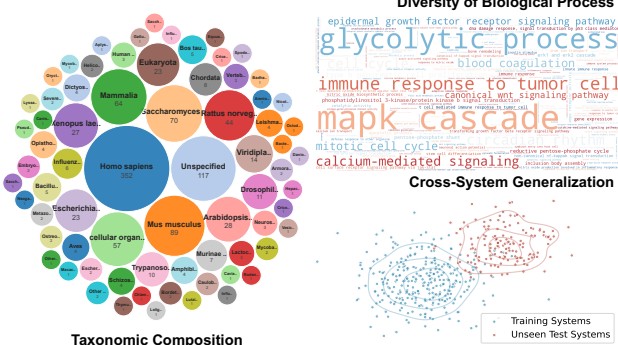

*Figure 1.* Diversity of SysBio-Traj across 1,050 biological models. **Left:** Taxonomic composition shows broad organism coverage (circle size indicates model count). **Top Right:** Gene Ontology biological process coverage (text size indicates model count). **Bottom Right:** Illustration of cross-system generalization task.

cesses (Ideker et al., 2001; Kitano, 2002), and these systems are traditionally modeled using ordinary differential equations (ODEs). While classical numerical solvers provide rigorous trajectory integration (Hairer & Wanner, 1996), they are hindered by high simulation costs and require explicit biological equations and ODE parameters that are usually incomplete, hypothetical, or difficult to specify in real-world biological discovery (Course & Nair, 2023). The advent of scientific machine learning has offered a promising alternative, with architectures such as Neural ODEs (Chen et al., 2018) and Neural Operators (Li et al., 2020) capable of approximating complex dynamics directly from observational data. However, these data-driven approaches are predominantly system-specific, trained to fit a single dynamical system and unable to generalize across heterogeneous biological systems (Vasiliauskaite & Antulov-Fantulin, 2024; Klötergens et al., 2025). This limitation becomes particularly prohibitive in large-scale settings involving thousands of biological systems (See our benchmark diversity in Figure 1), where per-system retraining is impractical. Consequently, a unified model is needed to handle trajectory prediction across heterogeneous biological systems.

To address the lack of scalability and transferability in system-specific models, general purpose time-series fore-

[1]College of Computer Science and Engineering, Northeastern University, Shenyang, Liaoning, China [2]Department of Computer Science, Wake Forest University, Winston-Salem, NC, USA [3]School of Computer Science, College of Computing, Georgia Institute of Technology, Atlanta, GA, USA [4]Department of Computer Science, Virginia Polytechnic Institute and State University, Blacksburg, VA, USA. Correspondence to: Yu Gu <guyu@mail.neu.edu.cn>, Minghan Chen <chenm@wfu.edu>.

*Proceedings of the $43^{rd}$ International Conference on Machine Learning*, Seoul, South Korea. PMLR 306, 2026. Copyright 2026 by the author(s).

casting architectures, such as Transformers (Liu et al., 2023; Zeng et al., 2023) and Mamba (Gu & Dao, 2024), have emerged as powerful system-agnostic tools to learn across heterogeneous datasets. While these approaches improve scalability, they are typically deterministic and produce point forecasts. Recent generative modeling paradigms, including diffusion models and flow matching approaches (Kollovieh et al., 2023; 2024), have been introduced to enable uncertainty quantification of trajectory prediction. Despite these advances, both deterministic and generative methods remain limited in long-horizon forecasting, as different future trajectories may share similar observed history, making the mapping from past to future ambiguous. Moreover, they may generate trajectories against known macroscopic behaviors, referred to as biological regimes. For example, irreversible disease progression exhibits monotonic dynamics; cell-cycle processes require sustained oscillations. This reflects a lack of biological inductive biases that enforce plausible dynamical regimes.

Motivated by these challenges, we propose a regime-aware trajectory prediction framework (RegimeFlow, see Figure 2), a unified architecture designed to master the diverse landscape of biological dynamics. Our key insight is that, despite massive heterogeneity, biological systems are not arbitrary but exhibit characteristic macroscopic behaviors (e.g., stable, monotonic, or oscillatory patterns) (Andrews et al., 2024; Tyson et al., 2003). Leveraging this, regime-level information is encoded directly in the generative prior, reducing reliance on heuristic regularization. Specifically, we construct regime-aware source distributions using Bayesian linear regression, initializing generation from biologically plausible states rather than isotropic Gaussian noise. This design significantly reduces the optimal transport distance between the source and target distributions, simplifying the learning process. To handle long-horizon dependencies, we employ a linear-complexity state-space backbone (Gu & Dao, 2024), modulated by adaptive conditioning to dynamically align the vector field with regime characteristics while preserving cross-system generalization. We evaluate RegimeFlow on a large collection of real-world biological systems (Figure 1), assessing its ability to generalize in long-horizon trajectory prediction with uncertainty quantification.

Our contributions are summarized as follows:

- We construct SysBio-Traj, a benchmark of 1,050 ODE-based biological systems spanning diverse organisms and pathways, and standardize it in a unified Python framework to support scalable evaluation of AI methods for systems biology.

- We propose RegimeFlow, a regime-aware trajectory prediction framework that encodes biological regime information through structural priors and adaptive con-

ditioning, enabling robust cross-system generalization with long-horizon uncertainty quantification.

- We provide theoretical justification and empirical evidence that regime-aware prior initialization yields a favorable starting distribution for conditional flow matching, leading to state-of-the-art accuracy (31% MAE reduction), well-calibrated uncertainty (17% CRPS improvement), and efficient long-horizon inference.

The resulting SysBio-Traj benchmark is publicly available at `https://huggingface.co/datasets/Heng Rao/SysBio-Traj`, and the RegimeFlow codebase and configurations are available at `https://github.com /hengrao02/RegimeFlow`.

## 2. Related Work

### 2.1. Point Forecasting in Time Series

Advancements in time-series forecasting have produced a diverse family of architectures, each navigating the fundamental trade-off between model expressivity and computational efficiency. High-capacity Transformer-based models (Nie, 2022; Liu et al., 2022; 2023) excel at capturing long-range dependencies through attention mechanisms, but this flexibility often incurs substantial computational and memory overhead, particularly for long sequences (Kim et al., 2025). In contrast, MLP-based models (Zeng et al., 2023; Ekambaram et al., 2023; Wang et al., 2024a) emphasize architectural simplicity and efficiency, though often at the expense of representational power. More recently, State Space Models (SSMs) (Liang et al., 2024; Ahamed & Cheng, 2024; Wang et al., 2025) have emerged as an effective middle ground. By leveraging linear recurrence, they achieve linear-time complexity while retaining the ability to model long-context dependencies (Liu et al., 2025a; Mei et al., 2025). Despite their success in general domains, these approaches face two challenges in biological contexts. First, they are inherently deterministic, producing single-point forecasts without predictive uncertainty that is essential for biological decision-making. Second, they struggle in long-horizon extrapolation under distribution shifts and sparse observations, which further degrade performance when transferring across heterogeneous biological systems.

### 2.2. Probabilistic Forecasting with Generative Models

Generative modeling has evolved significantly, progressing from the foundational architectures of GANs and VAEs (Kingma & Welling, 2013; Karras et al., 2019; Goodfellow et al., 2020) to the high-fidelity capabilities of Diffusion Probabilistic Models (Sohl-Dickstein et al., 2015). Diffusion-based frameworks (Rasul et al., 2021; Tashiro et al., 2021; Kollovieh et al., 2023) have recently advanced

conditional time-series forecasting, enabling uncertainty quantification through trajectory sampling. However, their iterative denoising process incurs prohibitive computational costs in real-world deployment (Lipman et al., 2022).

Flow Matching (FM) (Lipman et al., 2022; Tong et al., 2023b) offers a more efficient alternative by learning continuous probability paths grounded in optimal transport theory (McCann, 1997). By inducing nearly straight vector fields, FM reduces sampling cost and scales well to probabilistic forecasting. Yet, FM-based time-series models (Kollovieh et al., 2024; Hu et al., 2024; Zhang et al., 2024) typically rely on uninformative priors, neglecting critical temporal dependencies. Several recent works have attempted to introduce structural bias into generative forecasting, but limitations persist. TSFlow (Kollovieh et al., 2024) employs dataset-specific Gaussian Process priors that cannot adapt to heterogeneous regimes, while CGFM (Xu et al., 2025) constructs its source distribution using external auxiliary forecasters, increasing computational overhead. In contrast, our framework internalizes prior and conditioning adaptation within a unified architecture, eliminating the need for external predictors and providing robust, self-contained modeling across diverse biological dynamics.

## 3. Method

Figure 2 illustrates RegimeFlow, a regime-aware trajectory prediction framework for biological models. The key challenge is to generalize long-horizon predictions from short observation windows across heterogeneous systems. We address this by constructing a regime-aware prior and mapping it to target trajectory distributions via conditional flow matching, with adaptive conditioning to align the vector field with regime characteristics.

### 3.1. Preliminary: Conditional Flow Matching

Our approach builds on the conditional flow matching (CFM) framework (Lipman et al., 2022; Tong et al., 2023a), which constructs a time-dependent vector field to transport a source distribution $q_0$ to a target distribution $q_1$. Formally, CFM defines a vector field $u_t : [0, 1] \times \mathbb{R}^d \to \mathbb{R}^d$ that governs the trajectory of samples through $d\mathbf{x}_t = u_t(\mathbf{x}_t) \, dt$. This ODE induces a probability path $\{p_t\}_{t \in [0,1]}$ satisfying the boundary conditions $p_0 = q_0$ and $p_1 = q_1$. To circumvent the intractability of the marginal vector field generating $p_t$, CFM learns a neural approximation $v_\theta$ by regressing onto tractable conditional vector fields.

Following the method proposed in (Lipman et al., 2022), we define the conditional probability path via linear interpolation between a source sample $\mathbf{x}_0 \sim q_0$ and a target sample $\mathbf{x}_1 \sim q_1$. The resulting trajectory is expressed as $\mathbf{x}_t = (1 - t)\mathbf{x}_0 + t\mathbf{x}_1$ with $t \in [0, 1]$.

This choice yields straight conditional paths with constant conditional vector field $u_t(\mathbf{x}_t \mid \mathbf{x}_0, \mathbf{x}_1) = \mathbf{x}_1 - \mathbf{x}_0$. Consequently, learning reduces to a regression problem that matches $v_\theta(t, \mathbf{x}_t)$ to this computable target field:

$$\mathcal{L}_{\text{CFM}}(\theta) = \mathbb{E}_{\substack{t \sim \mathcal{U}[0,1] \\ \mathbf{x}_0 \sim q_0, \mathbf{x}_1 \sim q_1}} \left[ \|v_\theta(t, \mathbf{x}_t) - (\mathbf{x}_1 - \mathbf{x}_0)\|_2^2 \right].$$

### 3.2. Problem Formulation

We consider a universe of heterogeneous biological ODE systems $\mathcal{S}$. To rigorously evaluate the model's capability to generalize across distinct systems, we partition $\mathcal{S}$ into two disjoint sets: a source training set $\mathcal{S}_{\text{train}}$ and a target set $\mathcal{S}_{\text{test}}$ reserved for evaluation, ensuring that $\mathcal{S}_{\text{train}} \cap \mathcal{S}_{\text{test}} = \emptyset$.

In this framework, **biological regime** (denoted by $\mathcal{C}$) refers to a qualitative class of dynamical behavior exhibited by a system over time. Unlike prescribing specific trajectories or microscopic parameters (e.g., reaction rates), regimes characterize the macroscopic pattern that a biological system is constrained to follow. These include sustained oscillations (e.g., circadian rhythms or cell cycles), monotonic trends (e.g., irreversible disease progression or tumor growth), and convergence to stable equilibria (e.g., homeostatic regulation or metabolic processes). Systems within the same regime may differ in transient dynamics, but share similar long-term qualitative behavior.

For any system $S^{(i)} \in \mathcal{S}$, let $\mathbf{y} \in \mathbb{R}^{L+T}$ represent a univariate trajectory composed of a historical observation $\mathbf{y}_p \in \mathbb{R}^L$ and a future trajectory $\mathbf{y}_f \in \mathbb{R}^T$. Our goal is to learn a parameterized model on $\mathcal{S}_{\text{train}}$ that approximates the conditional target distribution $q_1(\mathbf{y}|\mathbf{y}_p, \mathcal{C})$ for unseen systems in $\mathcal{S}_{\text{test}}$, conditioned on the observation and biological regime. The conditional generative process is formalized as:

$$p_\theta(\mathbf{y}|\mathbf{y}_p, \mathcal{C}) = \int p_\theta(\mathbf{y}|\mathbf{x}_0, \mathbf{y}_p, \mathcal{C}) q_0(\mathbf{x}_0|\mathbf{y}_p, \mathcal{C}) \, d\mathbf{x}_0. \quad (1)$$

We introduce a regime-aware prior $q_0(\mathbf{x}_0|\mathbf{y}_p, \mathcal{C})$ that conditions the source distribution on the biological regime and the observed history via basis functions, thereby simplifying the learning of vector field $v_\theta$. Incorporating this prior into flow matching yields a specialized training objective:

$$\mathcal{L}(\theta) = \mathbb{E}_{t, \mathbf{x}_0, \mathbf{x}_1} \left[ \|v_\theta(t, \mathbf{x}_t, \mathbf{y}_p, \mathcal{C}) - (\mathbf{x}_1 - \mathbf{x}_0)\|_2^2 \right], \quad (2)$$

where $\mathbf{x}_0 \sim q_0(\cdot|\mathbf{y}_p, \mathcal{C})$ and $\mathbf{x}_1 \sim q_1(\cdot|\mathbf{y}_p, \mathcal{C})$.

### 3.3. Regime-Aware Prior Construction

In contrast to standard generative frameworks that rely on uninformative Gaussian source distributions, biological trajectories exhibit characteristic macroscopic behaviors, including stable, monotonic, and oscillatory regimes. We encode this information into a regime-aware prior, replacing

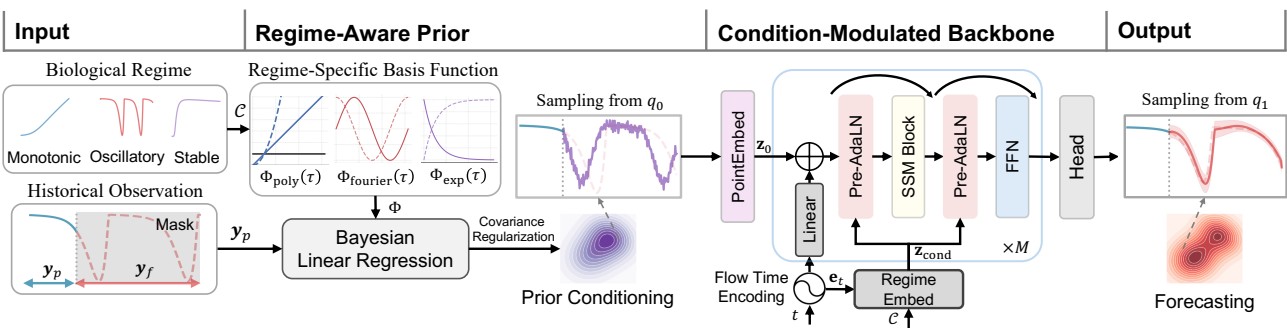

*Figure 2.* Overview of regime-aware trajectory prediction framework. Given historical observations $\mathbf{y}_p$, RegimeFlow encodes biological regime information via structured basis functions and Bayesian linear regression to infer a regime-aware prior $q_0$. Sampling from the prior initializes conditional flow matching. The backbone, adaptively conditioned on regime characteristics, learns a vector field $v_\theta$ to generate future long-horizon trajectories $\mathbf{y}_p$ via ODE integration.

generic stochastic initialization with structured initial states that are consistent with the system's dynamical patterns. The resulting transport process is biased toward biologically plausible trajectories from the outset.

### 3.3.1. DESIGN OF REGIME-SPECIFIC BASIS FUNCTIONS

We ground our trajectory modeling in classical function approximation theory. The Stone-Weierstrass Theorem (Stone, 1948) establishes that continuous functions on compact intervals can be uniformly approximated to arbitrary precision by sufficiently rich families of functions, such as polynomial or trigonometric bases. While our construction employs finite, structured basis sets, this result provides theoretical intuition for representing trajectories through basis superpositions.

Accordingly, we model an observed trajectory segment $\mathbf{y}_p$ as a weighted superposition of explicit basis functions, augmented with structural inductive biases to reflect regime priors. We formalize this basis projection as a Bayesian linear regression (BLR) (Box & Tiao, 2011) problem. Let $\mathbf{\Phi}(\tau) = [\phi_1(\tau), \ldots, \phi_K(\tau)]^\top \in \mathbb{R}^K$ denote the vector of basis functions evaluated at the relative time index $\tau$ of the trajectory. We specify the **likelihood function** as:

$$\mathbf{y}_p = \mathbf{\Phi}\mathbf{w} + \boldsymbol{\epsilon}, \quad \boldsymbol{\epsilon} \sim \mathcal{N}(\mathbf{0}, \beta^{-1}\mathbf{I}), \tag{3}$$

where $\mathbf{w} \in \mathbb{R}^K$ is the weight vector, $\beta$ denotes the noise precision, and $\boldsymbol{\epsilon}$ accounts for both aleatoric measurement noise and approximation errors due to basis truncation.

We construct the basis $\mathbf{\Phi}(\tau)$ as a composite library of functional primitives, explicitly tailored to span canonical biological dynamical regimes. **(i)** For stable trajectories characterized by asymptotic relaxation toward steady states, we employ a dictionary of exponential decay terms $\{1 - e^{-\gamma_j \tau}\}_j$ over a discretized set of rates $\{\gamma_j\}$, with scaling absorbed by the regression weights. **(ii)** To represent oscillatory patterns, we adopt a Fourier basis $\{\sin(k\omega\tau), \cos(k\omega\tau)\}_k$, where the fundamental frequency $\omega$ is informed by domain knowledge of the biological system when available (e.g., circadian

rhythms exhibit a 24-hour period). **(iii)** Monotonic trends are captured using polynomial basis $\{1, \tau, \tau^2, \ldots\}$. In complex scenarios where the dynamic pattern does not fall into any of the predefined regime categories, we revert to the degenerate basis $\{1\}$, yielding a time-invariant isotropic Gaussian prior. This fallback avoids imposing spurious structure when regime information is ambiguous. Regime labels are assigned per coordinate using an automated detector followed by manual validation; details are provided in Appendix C.4.

### 3.3.2. BAYESIAN INFERENCE

Given the Gaussian likelihood and the conjugate isotropic prior $p(\mathbf{w}) = \mathcal{N}(\mathbf{0}, \alpha^{-1}\mathbf{I})$, the posterior distribution $p(\mathbf{w} \mid \mathbf{y}_p)$ remains Gaussian and admits a closed-form solution:

$$p(\mathbf{w} \mid \mathbf{y}_p) = \mathcal{N}(\mathbf{w}; \boldsymbol{\mu}_\mathbf{w}, \boldsymbol{\Sigma}_\mathbf{w}). \tag{4}$$

The posterior covariance and mean are given by

$$\boldsymbol{\Sigma}_\mathbf{w} = \left(\alpha\mathbf{I} + \beta\mathbf{\Phi}_p^\top\mathbf{\Phi}_p\right)^{-1}, \quad \boldsymbol{\mu}_\mathbf{w} = \beta\boldsymbol{\Sigma}_\mathbf{w}\mathbf{\Phi}_p^\top\mathbf{y}_p, \tag{5}$$

where $\mathbf{\Phi}_p \in \mathbb{R}^{L \times K}$ denotes the design matrix formed by evaluating $K$ basis functions at the $L$ observed time points. $\alpha$ and $\beta$ denote the prior precision and the observation noise, respectively. As a basis for our prior, we first derive the standard Bayesian predictive distribution by marginalizing over the weight uncertainty. Define $\mathbf{\Phi}_f \in \mathbb{R}^{T \times K}$ as the basis matrix evaluated over the future $T$ time points. The resulting distribution is:

$$\mathcal{N}(\mathbf{x}_0; \mathbf{\Phi}_f\boldsymbol{\mu}_\mathbf{w}, \ \beta^{-1}\mathbf{I} + \mathbf{\Phi}_f\boldsymbol{\Sigma}_\mathbf{w}\mathbf{\Phi}_f^\top). \tag{6}$$

The predictive trajectory mean $\boldsymbol{\mu}_{\text{prior}} = \mathbf{\Phi}_f\boldsymbol{\mu}_\mathbf{w}$ provides a regime-informed extrapolation of the observed trajectory that reflects the macroscopic biological behavior implied by the chosen basis functions.

### 3.3.3. COVARIANCE REGULARIZATION

While the Bayesian linear regression provides a full covariance matrix, directly using this dense structure for long-horizon prediction ($T > L$) can be problematic. In particular, finite basis extrapolation over extended horizons may induce numerical instability and inflated variance estimates (Bishop & Nasrabadi, 2006), potentially misleading the generative process through poorly scaled priors. To address this instability, we retain the predictive mean $\boldsymbol{\mu}_{\text{prior}}$ and regularize the covariance used to initialize the flow. We replace the anisotropic predictive covariance with an isotropic approximation to improve numerical stability.

This simplification is supported by the Wasserstein-2 decomposition in Appendix A (Eq. 19), which separates the effect of mean alignment from the centered distribution term. Thus, fixing an isotropic covariance preserves the BLR mean guidance while stabilizing the source distribution; the empirical comparison with the full BLR covariance is provided in Appendix Table 3.

Formally, we define the regime-aware prior as

$$q_0(\mathbf{x}_0|\mathbf{y}_p, \mathcal{C}) = \mathcal{N}(\mathbf{x}_0; \boldsymbol{\mu}_{\text{prior}}, \sigma^2_{\text{init}}\mathbf{I}), \qquad (7)$$

where $\sigma^2_{\text{init}}$ is a scalar variance hyperparameter.

This covariance regularization preserves the extrapolated trend of the trajectory mean while avoiding numerical artifacts arising from long-range uncertainty propagation. Empirically, the residuals ($\mathbf{y}_f - \boldsymbol{\mu}_{\text{prior}}$) are approximately Gaussian, as shown in Appendix Figure 9.

### 3.3.4. THEORETICAL JUSTIFICATION

We provide a geometric characterization of the proposed prior, demonstrating that aligning the prior with the regime subspace yields provable advantages. Specifically, we establish that under an explicit Signal-to-Noise Ratio (SNR) condition, our approach strictly reduces the expected Optimal Transport cost relative to the standard Gaussian.

First, we analyze the estimator's variance. Let $\boldsymbol{\Sigma}_{\text{design}} := \boldsymbol{\Phi}_f(\boldsymbol{\Phi}_p^\top \boldsymbol{\Phi}_p)^{-1}\boldsymbol{\Phi}_f^\top$ denote the predictive covariance matrix determined solely by the basis geometry.

**Proposition 3.1** (Error Reduction under Regime-Aware Initialization). *In the limit of vanishing regularization ($\alpha \to 0$), the regime-aware prior yields a strictly lower mean squared error for trajectory initialization than the uninformative Gaussian prior, provided the following SNR condition holds:*

$$\frac{\|\boldsymbol{\mu}_Q\|^2}{\sigma^2_\epsilon} > \text{Tr}(\boldsymbol{\Sigma}_{\text{design}}), \qquad (8)$$

*where $\|\boldsymbol{\mu}_Q\|^2$ denotes the energy of the target mean trajectory and $\sigma^2_\epsilon = \beta^{-1}$ is the observation noise variance.*

Proposition 3.1 establishes that when the projected trajectory energy dominates the extrapolation variance induced by the basis geometry, regime-aware initialization improves estimation accuracy. Geometrically, this corresponds to initializing the flow closer to the target manifold, thereby shortening the transport path required for generation.

**Proposition 3.2** (Reduction of Expected Wasserstein-2 Cost). *Under the condition in Proposition 3.1, for any target distribution $Q$ with finite variance, the regime-aware prior $P_{\text{BLR}}$ yields a strictly smaller expected Wasserstein-2 distance to $Q$ than an uninformative Gaussian prior $P_{\text{GA}}$:*

$$\mathbb{E}_{\mathbf{y}_p}\left[W_2^2(P_{\text{BLR}}(\mathbf{y}_p), Q)\right] < W_2^2(P_{\text{GA}}, Q). \qquad (9)$$

Detailed proofs are provided in Appendix A, complemented by empirical verification in Appendix B.1. Together, these results formalize how regime-aware initialization reduces the geometric complexity of the transport problem by providing an initial trajectory distribution closer to the target dynamics. By effectively contracting the transport distance, this approach alleviates the model burden to learn highly complex mappings, thereby improving optimization stability and mitigating the ill-posedness commonly encountered in long-horizon forecasting.

**A Computable SNR Criterion.** The SNR condition in Proposition 3.1 depends on the unknown target mean $\boldsymbol{\mu}_Q$, and therefore cannot be checked directly at test time. We introduce a computable surrogate based on the observed history and the selected basis family:

$$\hat{\Delta}(\mathbf{y}_p) := \frac{\|\boldsymbol{\mu}_{\text{prior}}(\mathbf{y}_p)\|^2}{\hat{\sigma}^2_\epsilon} - \text{Tr}(\boldsymbol{\Sigma}_{\text{design}}), \qquad (10)$$

where $\boldsymbol{\mu}_{\text{prior}}(\mathbf{y}_p)$ is the predicted trajectory mean, used as a plug-in estimate of the target mean, and $\hat{\sigma}^2_\epsilon$ is estimated from the residuals obtained when fitting the observed window. This diagnostic operationalizes the SNR condition using quantities available from the observed history and the geometry of the selected basis. A positive value suggests that the regime-specific basis can provide an informative source initialization, whereas a non-positive value indicates potential regime mismatch or insufficient confidence, in which case regime-agnostic inference serves as a natural fallback or a more refined basis family may be warranted.

### 3.4. Condition-Modulated Backbone

To parameterize the neural vector field $v_\theta$, we propose the Condition-Modulated backbone. This backbone harnesses the linear time complexity of Selective State Space Model (SSM) (Gu & Dao, 2024) to capture long-range temporal dependencies efficiently.

### 3.4.1. INPUT EMBEDDING AND CONDITION ENCODING

We employ a channel-independent architecture to handle biological models of varying dimensions. Each dimension is processed as a univariate sequence derived from the intermediate flow state $\mathbf{x}_t$, while all dimensions share the same backbone parameters. This flow state $\mathbf{x}_t$, along with the flow time $t$ and biological regime $\mathcal{C}$, is projected into a shared latent space $\mathbb{R}^D$. The univariate input is encoded into the latent space $\mathbf{z}_0$ via an MLP-based point embedding: $\mathbf{z}_0 = \mathrm{MLP}_{\mathrm{emd}}(\mathbf{x}_t)$. Concurrently, the flow time $t \in [0, 1]$ is encoded via sinusoidal positional embeddings to yield $\mathbf{e}_t \in \mathbb{R}^D$ (Kollovieh et al., 2024). The initial hidden state of the backbone ($\mathbf{h}_0$) is formed by injecting a layer-specific time bias: $\mathbf{h}_0 = \mathbf{z}_0 + \mathrm{Linear}_{\mathrm{time}}(\mathbf{e}_t)$, where $\mathrm{Linear}_{\mathrm{time}}$ denotes the time linear projection.

To encode the biological regime $\mathcal{C} = (c_{\mathrm{pat}}, c_{\mathrm{freq}})$, we retrieve pattern embeddings $\mathbf{e}_{\mathrm{pat}}$ for the discrete pattern variable $c_{\mathrm{pat}} \in \{1, \ldots, N_{\mathrm{pat}}\}$ from a learnable matrix, and project the continuous frequency $c_{\mathrm{freq}} \in \mathbb{R}_{\geq 0}$ into embeddings $\mathbf{e}_{\mathrm{freq}}$ via a Fourier feature mapping with learnable frequencies (Tancik et al., 2020).

To facilitate classifier-free guidance (Ho & Salimans, 2022), regime embeddings (pattern and oscillatory frequency) are randomly masked during training and replaced with learnable null tokens, enabling the model to learn an explicit representation for missing regime information and capture both regime-conditioned and regime-agnostic dynamics. The processed regime features $(\widetilde{\mathbf{e}}_{\mathrm{pat}}, \widetilde{\mathbf{e}}_{\mathrm{freq}})$ are concatenated with the flow time embedding $\mathbf{e}_t$ and projected into a global conditioning vector: $\mathbf{z}_{\mathrm{cond}} = \mathrm{MLP}_{\mathrm{cond}}([\widetilde{\mathbf{e}}_{\mathrm{pat}}, \widetilde{\mathbf{e}}_{\mathrm{freq}}, \mathbf{e}_t])$. $\mathbf{z}_{\mathrm{cond}}$ serves to regulate the vector field dynamics.

### 3.4.2. CONDITION-MODULATED LAYER

The backbone consists of $M$ stacked condition-modulated layers designed to integrate regime prior directly into the latent dynamics. Each layer utilizes a pre-normalized residual structure, comprising an SSM block followed by a Feed-Forward Network (FFN). Conditioning is achieved using Adaptive Layer Normalization (AdaLN) (Perez et al., 2018), which modulates intermediate representations based on the global conditioning vector $\mathbf{z}_{\mathrm{cond}}$. For the $l$-th layer, AdaLN computes dimension-wise scale ($\mathbf{s}_l$) and shift ($\mathbf{d}_l$) from $\mathbf{z}_{\mathrm{cond}}$ and applies them to normalized hidden states:

$$
\begin{aligned}
(\mathbf{s}_l, \mathbf{d}_l) &= \mathrm{Linear}_{\mathrm{ada}}^{(l)}(\mathbf{z}_{\mathrm{cond}}), \\
\mathrm{AdaLN}(\mathbf{h}, \mathbf{z}_{\mathrm{cond}}) &= \mathrm{RMSNorm}(\mathbf{h}) \odot (1 + \mathbf{s}_l) + \mathbf{d}_l.
\end{aligned}
\tag{11}
$$

Following (Peebles & Xie, 2023), we initialize the projection $\mathrm{Linear}_{\mathrm{ada}}^{(l)}$ with zeros, starting the modulation as an identity transformation ($\mathbf{s}_l = \mathbf{0}, \mathbf{d}_l = \mathbf{0}$), allowing the backbone to initially train in a regime-agnostic manner and improving convergence stability.

With this modulation, the state updates for the $l$-th layer are:

$$
\begin{aligned}
\mathbf{h}_l' &= \mathbf{h}_{l-1} + \mathrm{SSM}\left(\mathrm{AdaLN}(\mathbf{h}_{l-1}, \mathbf{z}_{\mathrm{cond}})\right), \\
\mathbf{h}_l &= \mathbf{h}_l' + \mathrm{FFN}\left(\mathrm{AdaLN}(\mathbf{h}_l', \mathbf{z}_{\mathrm{cond}})\right).
\end{aligned}
\tag{12}
$$

Finally, to preserve multi-scale representations across network depths, the target vector field is predicted by decoding the aggregated sum of all layer outputs, rather than relying solely on the final hidden state.

## 4. Experiments

We evaluate RegimeFlow using our benchmark designed to rigorously assess long-horizon trajectory inference across diverse biological systems. All reported metrics are presented as the mean and standard deviation ($\mu \pm \sigma$) over three independent seeds, providing a quantitative measure of predictive stability and robustness.

### 4.1. Benchmark Construction and Task Formulation

To evaluate performance across the broad spectrum of biological complexity, we assemble SysBio-Traj, a large-scale benchmark of 1,050 ODE systems spanning diverse taxonomic domains and dynamical regimes sourced from (Malik-Sheriff et al., 2020) (see Appendix D for the full list of biological systems). As illustrated in Appendix Figure 15, this benchmark exhibits substantial heterogeneity, with system dimensions ranging from 1 to 1,000 and reaction counts spanning three orders of magnitude. For each system, ground-truth trajectories are generated by numerically solving the ODEs and sampling the solution at 512 uniformly spaced time points.

To emulate real-world scenarios where observational data is limited, we formulate a challenging long-horizon forecasting task: predicting trajectories over $T = 256$ time points from a given $L = 96$ observed points. Specifically for oscillatory systems, we forecast one full cycle from observations covering only part of a cycle, testing the model's ability to extrapolate full oscillatory dynamics when periodicity is not explicitly apparent. We adopt a randomized system-wise train/validation/test split (70%/10%/20%), such that models are trained and evaluated on disjoint sets of biological systems. This setup enforces a rigorous out-of-distribution (OOD) generalization. To accommodate the wide variation in system dimensionality, we employ a channel-independent modeling strategy with instance-wise normalization.

### 4.2. Baselines and Implementation Details

We compare our framework against two prominent paradigms: point forecasting and probabilistic generative models. Point forecasting baselines include MLP-based models such as DLinear (Zeng et al., 2023) and TimeMixer (Wang et al., 2024a), Transformer-based models

*Table 1.* Cross-system generalization performance on MSE and MAE across regime-specific subsets, complex subset outside the three regimes, and overall. Each entry reports the mean with standard deviation. Best and second-best performances are highlighted in bold and underline, respectively. Models marked with † additionally condition on biological regime, including pattern and frequency.

| Model | Overall | | Complex | | Stable | | Oscillatory | | Monotonic | |
|---|---|---|---|---|---|---|---|---|---|---|
| | MSE↓ | MAE↓ | MSE↓ | MAE↓ | MSE↓ | MAE↓ | MSE↓ | MAE↓ | MSE↓ | MAE↓ |
| *Point Forecasting* | | | | | | | | | | |
| DLinear | $0.061_{\pm0.002}$ | $0.174_{\pm0.004}$ | $0.079_{\pm0.003}$ | $0.188_{\pm0.010}$ | $0.046_{\pm0.001}$ | $0.154_{\pm0.005}$ | $0.156_{\pm0.013}$ | $0.290_{\pm0.011}$ | $0.051_{\pm0.005}$ | $0.176_{\pm0.013}$ |
| iTransformer | $0.045_{\pm0.002}$ | $0.100_{\pm0.005}$ | $0.070_{\pm0.010}$ | $0.138_{\pm0.020}$ | $0.032_{\pm0.002}$ | $0.072_{\pm0.004}$ | $0.127_{\pm0.028}$ | $0.233_{\pm0.018}$ | $0.030_{\pm0.008}$ | $0.100_{\pm0.018}$ |
| PatchTST | $0.041_{\pm0.001}$ | $0.095_{\pm0.005}$ | $0.062_{\pm0.006}$ | $0.133_{\pm0.021}$ | $0.028_{\pm0.001}$ | $0.070_{\pm0.004}$ | $0.128_{\pm0.024}$ | $0.233_{\pm0.016}$ | $0.021_{\pm0.006}$ | $0.085_{\pm0.016}$ |
| TimeMixer | $0.051_{\pm0.004}$ | $0.124_{\pm0.008}$ | $0.070_{\pm0.008}$ | $0.145_{\pm0.017}$ | $0.038_{\pm0.004}$ | $0.096_{\pm0.009}$ | $0.112_{\pm0.013}$ | $0.234_{\pm0.014}$ | $0.053_{\pm0.010}$ | $0.163_{\pm0.020}$ |
| TimeXer | $0.043_{\pm0.001}$ | $0.100_{\pm0.007}$ | $0.069_{\pm0.009}$ | $0.139_{\pm0.020}$ | $0.030_{\pm0.003}$ | $0.074_{\pm0.008}$ | $0.123_{\pm0.028}$ | $0.231_{\pm0.014}$ | $0.025_{\pm0.006}$ | $0.098_{\pm0.015}$ |
| SMamba | $0.045_{\pm0.001}$ | $0.105_{\pm0.004}$ | $0.068_{\pm0.010}$ | $0.141_{\pm0.023}$ | $0.033_{\pm0.003}$ | $0.078_{\pm0.005}$ | $0.114_{\pm0.016}$ | $0.228_{\pm0.013}$ | $0.035_{\pm0.005}$ | $0.115_{\pm0.010}$ |
| BiMamba4TS | $0.045_{\pm0.001}$ | $0.099_{\pm0.004}$ | $0.070_{\pm0.010}$ | $0.140_{\pm0.021}$ | $0.031_{\pm0.002}$ | $0.070_{\pm0.004}$ | $0.125_{\pm0.022}$ | $0.237_{\pm0.015}$ | $0.030_{\pm0.005}$ | $0.102_{\pm0.013}$ |
| NSformer | $0.028_{\pm0.002}$ | $0.079_{\pm0.005}$ | $0.048_{\pm0.002}$ | $0.116_{\pm0.007}$ | $0.017_{\pm0.001}$ | $0.055_{\pm0.005}$ | $0.100_{\pm0.008}$ | $0.221_{\pm0.006}$ | $0.011_{\pm0.001}$ | $0.053_{\pm0.002}$ |
| *Time-Series Foundation Models (Zero-Shot)* | | | | | | | | | | |
| TimeMoE | $0.120_{\pm0.004}$ | $0.231_{\pm0.010}$ | $0.099_{\pm0.013}$ | $0.191_{\pm0.032}$ | $0.118_{\pm0.003}$ | $0.221_{\pm0.010}$ | $0.138_{\pm0.005}$ | $0.268_{\pm0.005}$ | $0.138_{\pm0.012}$ | $0.289_{\pm0.023}$ |
| Sundial | $0.111_{\pm0.003}$ | $0.217_{\pm0.007}$ | $0.095_{\pm0.012}$ | $0.185_{\pm0.031}$ | $0.106_{\pm0.002}$ | $0.203_{\pm0.008}$ | $0.141_{\pm0.009}$ | $0.270_{\pm0.009}$ | $0.133_{\pm0.011}$ | $0.281_{\pm0.022}$ |
| TimesFM | $0.099_{\pm0.005}$ | $0.152_{\pm0.004}$ | $0.109_{\pm0.009}$ | $0.145_{\pm0.021}$ | $0.086_{\pm0.002}$ | $0.138_{\pm0.006}$ | $0.249_{\pm0.032}$ | $0.299_{\pm0.014}$ | $0.045_{\pm0.004}$ | $0.112_{\pm0.007}$ |
| Chronos | $0.097_{\pm0.005}$ | $0.167_{\pm0.004}$ | $0.144_{\pm0.021}$ | $0.181_{\pm0.012}$ | $0.078_{\pm0.003}$ | $0.147_{\pm0.002}$ | $0.234_{\pm0.013}$ | $0.309_{\pm0.007}$ | $0.054_{\pm0.004}$ | $0.153_{\pm0.005}$ |
| *Probabilistic Forecasting w/o Regime Condition (Best Performing: Ours)* | | | | | | | | | | |
| CSDI | $0.099_{\pm0.020}$ | $0.254_{\pm0.034}$ | $0.100_{\pm0.030}$ | $0.250_{\pm0.048}$ | $0.103_{\pm0.022}$ | $0.264_{\pm0.038}$ | $0.137_{\pm0.012}$ | $0.304_{\pm0.014}$ | $0.053_{\pm0.013}$ | $0.172_{\pm0.030}$ |
| TSDiff | $0.042_{\pm0.003}$ | $0.118_{\pm0.004}$ | $0.058_{\pm0.007}$ | $0.146_{\pm0.014}$ | $0.030_{\pm0.003}$ | $0.096_{\pm0.004}$ | $0.124_{\pm0.010}$ | $0.263_{\pm0.012}$ | $0.023_{\pm0.001}$ | $0.092_{\pm0.004}$ |
| TSFlow | $0.026_{\pm0.003}$ | $0.075_{\pm0.011}$ | $0.044_{\pm0.001}$ | $0.111_{\pm0.009}$ | $0.017_{\pm0.004}$ | $0.055_{\pm0.012}$ | $0.088_{\pm0.007}$ | $0.208_{\pm0.007}$ | $0.008_{\pm0.002}$ | $0.044_{\pm0.008}$ |
| Ours | $0.023_{\pm0.002}$ | $0.067_{\pm0.004}$ | $0.039_{\pm0.001}$ | $0.102_{\pm0.008}$ | $0.015_{\pm0.002}$ | $0.046_{\pm0.003}$ | $0.081_{\pm0.008}$ | $0.196_{\pm0.009}$ | $0.008_{\pm0.002}$ | $0.041_{\pm0.005}$ |
| *Probabilistic Forecasting w/ Regime Condition (Best Performing: Ours)* | | | | | | | | | | |
| CSDI† | $0.076_{\pm0.010}$ | $0.204_{\pm0.012}$ | $0.100_{\pm0.009}$ | $0.236_{\pm0.011}$ | $0.072_{\pm0.015}$ | $0.196_{\pm0.019}$ | $0.105_{\pm0.011}$ | $0.257_{\pm0.024}$ | $0.060_{\pm0.024}$ | $0.183_{\pm0.046}$ |
| TSDiff† | $0.034_{\pm0.008}$ | $0.109_{\pm0.017}$ | $0.088_{\pm0.033}$ | $0.201_{\pm0.054}$ | $0.014_{\pm0.003}$ | $0.068_{\pm0.011}$ | $0.118_{\pm0.002}$ | $0.292_{\pm0.001}$ | $0.014_{\pm0.001}$ | $0.078_{\pm0.009}$ |
| TSFlow† | $0.016_{\pm0.004}$ | $0.064_{\pm0.013}$ | $0.039_{\pm0.005}$ | $0.111_{\pm0.022}$ | $0.008_{\pm0.002}$ | $0.046_{\pm0.010}$ | $0.044_{\pm0.005}$ | $0.139_{\pm0.017}$ | $0.008_{\pm0.001}$ | $0.050_{\pm0.005}$ |
| Ours† | **$0.012_{\pm0.002}$** | **$0.044_{\pm0.006}$** | **$0.034_{\pm0.002}$** | **$0.096_{\pm0.011}$** | **$0.006_{\pm0.001}$** | **$0.027_{\pm0.003}$** | **$0.035_{\pm0.002}$** | **$0.111_{\pm0.008}$** | **$0.006_{\pm0.000}$** | **$0.033_{\pm0.002}$** |

*Table 2.* Uncertainty quantification measured by CRPS (↓) across regime-specific subsets, complex subset, and overall. Best results are in **bold**. (†: condition on biological regime.)

| Model | Overall | Complex | Stable | Oscillatory | Monotonic |
|---|---|---|---|---|---|
| CSDI† | 0.138 | 0.166 | 0.127 | 0.186 | 0.135 |
| | (±0.013) | (±0.018) | (±0.019) | (±0.015) | (±0.040) |
| TSDiff† | 0.077 | 0.151 | 0.048 | 0.198 | 0.057 |
| | (±0.015) | (±0.049) | (±0.009) | (±0.001) | (±0.006) |
| TSFlow† | 0.047 | 0.086 | 0.034 | 0.100 | 0.037 |
| | (±0.010) | (±0.016) | (±0.008) | (±0.011) | (±0.003) |
| Ours† | **0.039** | **0.086** | **0.023** | **0.098** | **0.028** |
| | (±0.006) | (±0.010) | (±0.003) | (±0.007) | (±0.002) |

We assess point prediction accuracy using mean squared error (MSE) and mean absolute error (MAE). To evaluate distributional fidelity, we employ the continuous ranked probability score (CRPS) (Gneiting & Raftery, 2007), which is approximated using 100 Monte Carlo samples per test instance. All experiments were implemented in PyTorch and executed on a single NVIDIA A6000 GPU.

### 4.3. Main Results

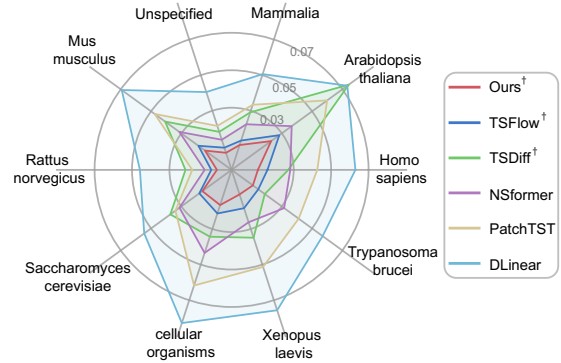

*Figure 3.* MSE comparison between Ours† and five competitive baselines across ten most frequent biological system taxonomies.

including iTransformer (Liu et al., 2023), PatchTST (Nie, 2022), TimeXer (Wang et al., 2024b), and NSformer (Liu et al., 2022), as well as state-space models such as SMamba (Wang et al., 2025) and BiMamba4TS (Liang et al., 2024). For probabilistic baselines, we consider diffusion models, TSDiff (Kollovieh et al., 2023) and CSDI (Tashiro et al., 2021), and flow matching methods, TS-Flow (Kollovieh et al., 2024). We additionally evaluate four zero-shot time-series foundation models (TSFMs), including TimeMoE (Shi et al., 2025), Sundial (Liu et al., 2025b), TimesFM (Das et al., 2024), and Chronos (Ansari et al., 2024); detailed settings and analysis are provided in Appendix B.7. Full baseline descriptions and implementation settings are provided in Appendix C.

Table 1 summarizes the forecasting performance on unseen biological systems. When incorporating regime information (denoted by †), our method consistently outperforms all

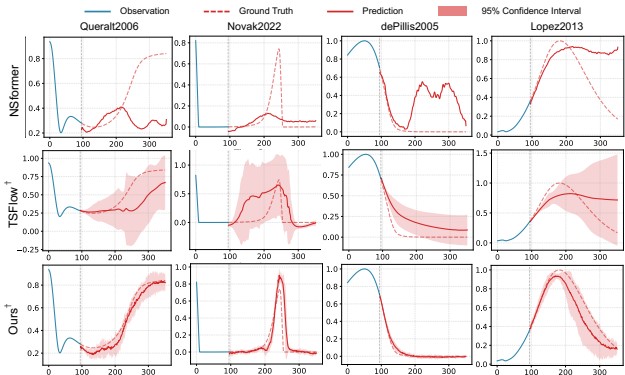

*Figure 4.* Visualization of trajectory forecasts by Ours[†], TSFlow[†], and NSformer on four representative biological systems.

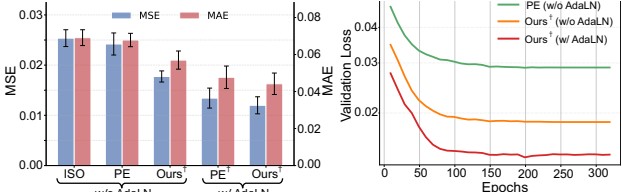

*Figure 5.* **Left:** Ablation analysis of prior designs (ISO, PE, Our regime-aware) and adaptive conditioning (AdaLN). **Right:** Validation loss under different prior initializations and conditioning.

SOTA baselines across individual regimes, complex systems outside the three canonical regimes, and overall. It achieves overall reductions of 25% in MSE and 31% in MAE relative to the best-performing baseline, TSFlow[†]. These improvements persist across diverse regimes (MSE: 14%∼32%), underscoring our superior capability for cross-system generalization. Figure 3 further shows our method attains the lowest MSE across all ten most frequent biological system taxonomies compared to baselines.

To demonstrate robustness and uncertainty across distinct dynamical behaviors, we visualize predictions on four representative biological systems: bistable mitotic exit switch (Queralt et al., 2006), oscillatory endoreplication (Novak & Tyson, 2022), tumor-immune interactions (de Pillis et al., 2005), and a high-dimensional apoptosis model (Lopez et al., 2013). As shown in Figure 4, our framework produces the most accurate predictions with well-calibrated uncertainty. In contrast, TSFlow[†] exhibits degraded predictive performance with substantially wider confidence intervals, likely due to its reliance on fixed, system-agnostic priors, while NSformer struggles to accommodate diverse regimes. Consistent with these qualitative observations, our approach achieves an average 17% reduction in CRPS relative to the strongest baseline, TSFlow[†] (Table 2), demonstrating superior uncertainty calibration and closer alignment with the target distribution.

Notably, even without regime information, our framework remains competitive and consistently outperforms all baselines across individual subsets and overall. RegimeFlow reduces MSE and MAE by 12% and 11% relative to the best-performing baseline, TSFlow. This advantage likely stems from the input-dependent dynamics of the SSM-based backbone (Gu et al., 2021; Gu & Dao, 2024). Point forecasting architectures, while effective on single-domain benchmarks (Lai et al., 2018; Makridakis et al., 2020), exhibit fundamental limitations in this cross-system setting and often collapse to trivial mean trends across diverse regimes.

Among point forecasting baselines, NSformer achieves the strongest performance, likely due to its series stationarization and de-stationary attention mechanisms, which enhance robustness to scale shifts. Conversely, methods that rely heavily on inter-variable correlations, such as iTransformer (Liu et al., 2023), are disadvantaged as biological coupling relationships often vary drastically across systems.

Table 1 also reports zero-shot results from four representative TSFMs. Despite their larger parameter scales (128–200 M), these models remain substantially less accurate on SysBio-Traj; the strongest TSFM, Chronos, obtains an overall MSE of 0.097, compared with 0.012 for Ours[†]. Detailed analyses are provided in Appendix B.7.

### 4.4. Effectiveness of Regime-Aware Conditioning

To assess the contribution of each component, we independently ablate the regime-aware prior and adaptive conditioning design. In Figure 5 (left), replacing the Isotropic (ISO) kernel with the Periodic (PE) kernel yields only a marginal improvement in accuracy. In contrast, substituting the above kernels with our regime-aware prior leads to a substantially larger reduction in error (MSE: 0.025→0.018), indicating that our prior provides more informative guidance for modeling complex dynamical behaviors than dataset-specific priors. Under our prior, introducing adaptive conditioning (AdaLN) further reduces the error (MSE: 0.018→0.012), highlighting its importance in leveraging prior information. Figure 5 (right) shows that our initialization converges to a consistently lower validation loss than the PE kernel under identical model settings, highlighting the optimization benefits of the regime-aware prior.

### 4.5. Sensitivity, Efficiency, and Extrapolation Analysis

We first assess the model's sensitivity to the regime information availability. During training, regime information is randomly masked as a form of regularization, discouraging over-reliance on explicit prior metadata and encouraging the backbone to infer regime-relevant features directly from raw time-series observations. When regime information is masked, system dynamics are treated as unknown and the source distribution is initialized from the observed history as

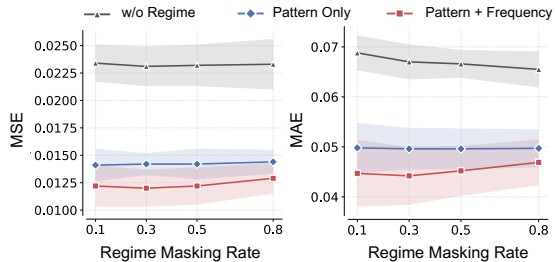

Figure 6. Sensitivity to regime information availability. Our model is trained with different regime masking rates and tested under varying regime conditions (no regime, pattern, pattern & frequency).

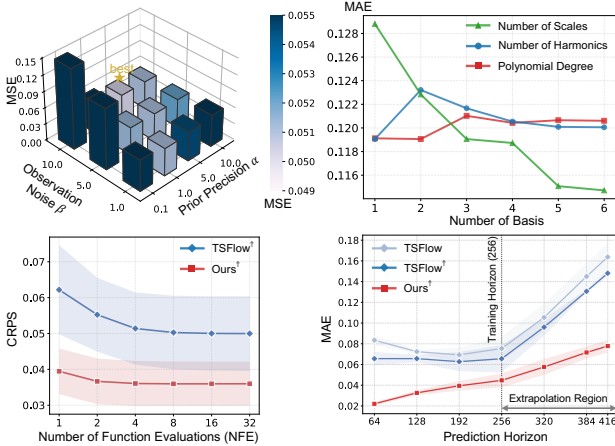

Figure 7. **Top left:** Sensitivity to BLR hyperparameters. **Top right:** Effect of basis complexity (numbers of harmonics, scales, and polynomial degrees). **Bottom left:** Inference efficiency (CRPS vs. Number of Function Evaluation). **Bottom right:** Extrapolation performance under extended forecasting horizons.

$\mathcal{N}(\mathbf{y}_L, \sigma_{\text{init}}^2 \mathbf{I})$. Figure 6 reveals a trade-off: higher masking improves regime-agnostic accuracy but slightly degrades regime-aware performance. A masking rate of 0.3 offers a good balance, maintaining low regime-aware error while preserving competitive regime-agnostic performance.

We further investigate the sensitivity of Bayesian linear regression (BLR) hyperparameters. Figure 7 (top left) identifies prior precision $\alpha = 5.0$ and observation noise $\beta = 10.0$ as the optimal choices. The top-right panel shows that a quadratic polynomial coupled with a single harmonic component offers sufficient performance, while adding additional scales further improves accuracy. In terms of inference efficiency (bottom left), our method achieves near-optimal CRPS with a single function evaluation step, significantly outperforming TSFlow[†], which requires $\geq 16$ steps. This efficiency suggests that the proposed prior initializes the flow closer to the target distribution, thereby reducing the complexity of ODE integration. A full computational cost comparison across all models is provided in Appendix B.6, where RegimeFlow is $9.3\times$ faster than TSFlow[†] during inference with a single function evaluation. Finally, the extrapolation analysis (bottom right) demonstrates that our method maintains low error when extending the prediction horizon from 256 to 416 time steps. In contrast, TSFlow exhibits a much steeper error increase, highlighting the improved robustness of regime-aware initialization under horizon shifts.

### 4.6. Additional Robustness Studies

Beyond the main benchmark results and sensitivity analyses discussed above, Appendix B provides supplementary stress tests and component-level ablations. Under measurement noise and irregular sampling, RegimeFlow maintains consistent advantages over TSFlow[†] (Appendix B.10); on representative transient and hybrid trajectories, it accommodates dynamics beyond canonical regimes, supporting that the regime-aware prior acts as soft guidance rather than a rigid constraint (Appendix B.8).

The appendix also provides component-level ablations on

covariance regularization, backbone choice, and TSFlow kernel configurations (Appendix Sections B.2, B.3, and B.4). For the NFE, extrapolation, and theory-bound analyses discussed above, Appendix Sections B.5, B.9, and B.1 provide detailed protocols and additional quantitative results. Together, these studies show that RegimeFlow remains effective beyond the clean benchmark setting, while the ablations confirm the complementary benefits of the regime-aware prior and sequence backbone.

## 5. Conclusion

This work introduced RegimeFlow, a unified framework for cross-system biological trajectory prediction. By replacing an uninformative source distribution with a regime-aware prior, RegimeFlow improves long-horizon forecasting accuracy, uncertainty calibration, and inference efficiency across SysBio-Traj, a large curated benchmark of biological systems. Beyond model performance, SysBio-Traj provides a standardized testbed for evaluating cross-system generalization in systems biology. Several limitations motivate future work. RegimeFlow currently adopts a channel-independent formulation to support transfer across systems with heterogeneous dimensions and system-specific interaction topologies, leaving topology-aware relational modeling as an important extension. Although representative transient and hybrid behaviors can be handled through soft regime guidance (Appendix B.8), parameter-dependent regime transitions and highly irregular or chaotic dynamics may require more adaptive, data-driven regime priors beyond the current basis library. Future work will also extend SysBio-Traj and RegimeFlow toward stochastic, spatio-temporal, and multi-modal biological settings.

## Impact Statement

This paper presents work whose goal is to advance the field of Machine Learning and its application to Systems Biology. There are many potential societal consequences of our work, none which we feel must be specifically highlighted here.

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

# A. Theoretical Analysis and Proofs

In this appendix, we formally derive the theoretical conditions under which the regime-aware prior yields a strictly lower estimation risk than an uninformative baseline. We focus our analysis on the limit of vanishing regularization ($\alpha \to 0$). In this regime, the prediction is governed solely by the projection of the history onto the mechanism subspace, allowing us to isolate and quantify the geometric advantage of the proposed framework.

## A.1. Proof of Proposition 3.1

**Proposition 3.1 (Error Reduction under Regime-Aware Initialization).** *Consider the observation regime where the history length $L$ is sufficient to identify the $K$ basis coefficients ($L \geq K$ and $\Phi_p$ has full column rank). Let $\Sigma_{\text{design}} := \Phi_f (\Phi_p^\top \Phi_p)^{-1} \Phi_f^\top$ denote the predictive covariance matrix determined solely by the basis geometry. In the limit of vanishing regularization ($\alpha \to 0$), the Regime-Aware Prior strictly reduces the Mean Squared Error (MSE) compared to the uninformed zero-mean prior if and only if the Signal-to-Noise Ratio (SNR) satisfies:*

$$\frac{\|\boldsymbol{\mu}_Q\|^2}{\sigma_\epsilon^2} > \text{Tr}(\Sigma_{\text{design}}), \tag{13}$$

*where $\|\boldsymbol{\mu}_Q\|^2$ denotes the energy of the ground truth trajectory, and $\sigma_\epsilon^2$ represents the variance of the observation noise.*

*Proof.* We assess the efficacy of the regime-aware prior by comparing the MSE of the proposed estimator against that of the zero-mean baseline. For the uninformative estimator $\hat{\boldsymbol{\mu}}_{\text{GA}} = \mathbf{0}$, the risk corresponds strictly to the energy of the ground truth trajectory $\boldsymbol{\mu}_Q = \Phi_f \mathbf{w}^*$:

$$R_{\text{GA}} := \mathbb{E}_{\mathbf{y}_p}[\|\mathbf{0} - \boldsymbol{\mu}_Q\|^2] = \|\boldsymbol{\mu}_Q\|^2. \tag{14}$$

Next, consider the regime-aware prior $\boldsymbol{\mu}_{\text{prior}} = \Phi_f \hat{\mathbf{w}}$. In the limit of vanishing regularization ($\alpha \to 0$), then:

$$\hat{\mathbf{w}} = (\Phi_p^\top \Phi_p)^{-1} \Phi_p^\top \mathbf{y}_p. \tag{15}$$

Under the linear observation model $\mathbf{y}_p = \Phi_p \mathbf{w}^* + \boldsymbol{\epsilon}$, this estimator is unbiased, satisfying $\mathbb{E}[\hat{\mathbf{w}}] = \mathbf{w}^*$. Consequently, the bias term in the risk decomposition vanishes, and the total risk is governed solely by the predictive variance:

$$\begin{aligned} R_{\text{BLR}} &:= \mathbb{E}_{\mathbf{y}_p}[\|\Phi_f(\hat{\mathbf{w}} - \mathbf{w}^*)\|^2] \\ &= \mathbb{E}_{\boldsymbol{\epsilon}}[\|\Phi_f(\Phi_p^\top \Phi_p)^{-1} \Phi_p^\top \boldsymbol{\epsilon}\|^2]. \end{aligned} \tag{16}$$

Using the property that $\mathbf{x}^\top \mathbf{x} = \text{Tr}(\mathbf{x}\mathbf{x}^\top)$ and the noise assumption $\mathbb{E}[\boldsymbol{\epsilon}\boldsymbol{\epsilon}^\top] = \sigma_\epsilon^2 \mathbf{I}$, we expand the expectation:

$$\begin{aligned} R_{\text{BLR}} &= \mathbb{E}[\text{Tr}\left(\Phi_f(\Phi_p^\top \Phi_p)^{-1} \Phi_p^\top \boldsymbol{\epsilon}\boldsymbol{\epsilon}^\top \Phi_p (\Phi_p^\top \Phi_p)^{-1} \Phi_f^\top\right)] \\ &= \text{Tr}\left(\Phi_f(\Phi_p^\top \Phi_p)^{-1} \Phi_p^\top \mathbb{E}[\boldsymbol{\epsilon}\boldsymbol{\epsilon}^\top] \Phi_p (\Phi_p^\top \Phi_p)^{-1} \Phi_f^\top\right) \\ &= \sigma_\epsilon^2 \text{Tr}\left(\Phi_f(\Phi_p^\top \Phi_p)^{-1} (\Phi_p^\top \Phi_p)(\Phi_p^\top \Phi_p)^{-1} \Phi_f^\top\right) \\ &= \sigma_\epsilon^2 \text{Tr}\left(\Phi_f(\Phi_p^\top \Phi_p)^{-1} \Phi_f^\top\right) \\ &= \sigma_\epsilon^2 \text{Tr}(\Sigma_{\text{design}}). \end{aligned} \tag{17}$$

Therefore, the condition for strict superiority, $R_{\text{BLR}} < R_{\text{GA}}$, holds if and only if $\sigma_\epsilon^2 \text{Tr}(\Sigma_{\text{design}}) < \|\boldsymbol{\mu}_Q\|^2$. Rearranging terms yields the stated inequality. $\square$

## A.2. Proof of Proposition 3.2

**Proposition 3.2 (Reduction of Expected Wasserstein-2 Cost).** *Under the condition in Proposition 3.1, for target measure $Q$ with finite second moments (not necessarily Gaussian), the regime-aware prior $P_{BLR}$ strictly reduces the expected Wasserstein-2 distance to $Q$ compared to the standard Gaussian prior $P_{GA}$:*

$$\mathbb{E}_{\mathbf{y}_p}\left[W_2^2(P_{BLR}(\mathbf{y}_p), Q)\right] < W_2^2(P_{GA}, Q). \tag{18}$$

*Proof.* We compare the transport costs associated with the uninformed prior $P_{\text{GA}} = \mathcal{N}(\mathbf{0}, \sigma_{\text{init}}^2 \mathbf{I})$ and the regime-aware prior $P_{\text{BLR}}(\mathbf{y}_p) = \mathcal{N}(\boldsymbol{\mu}_{\text{prior}}, \sigma_{\text{init}}^2 \mathbf{I})$. Observe that both distributions share the same isotropic covariance $\sigma_{\text{init}}^2 \mathbf{I}$, differing only in their first moments.

We invoke the translation invariance property of the Wasserstein-2 distance. For any probability measure $\mu$ with finite second moments and any target measure $Q$ on a Euclidean space, the squared distance decomposes orthogonally into the distance between their means and the distance between their centered distributions (Remark 2.19 in (Peyré et al., 2019)):

$$W_2^2(\mu, Q) = \|\mathbb{E}[\mu] - \mathbb{E}[Q]\|_2^2 + W_2^2(\bar{\mu}, \bar{Q}), \tag{19}$$

where $\bar{\mu}$ and $\bar{Q}$ denote the centered distributions. Since the covariance of $P_{\text{BLR}}$ is fixed to $\sigma_{\text{init}}^2 \mathbf{I}$ independent of the observation $\mathbf{y}_p$, its centered distribution is deterministic and identical to that of $P_{\text{GA}}$. Specifically, $\bar{P}_{\text{BLR}} = \bar{P}_{\text{GA}} = \mathcal{N}(\mathbf{0}, \sigma_{\text{init}}^2 \mathbf{I})$. Consequently, the shape component of the transport cost is a constant, which we denote by $C_{\text{shape}} := W_2^2(\mathcal{N}(\mathbf{0}, \sigma_{\text{init}}^2 \mathbf{I}), Q)$.

For the deterministic uninformed prior, the transport cost is thus:

$$W_2^2(P_{\text{GA}}, Q) = \|\mathbf{0} - \boldsymbol{\mu}_Q\|^2 + C_{\text{shape}} = \|\boldsymbol{\mu}_Q\|^2 + C_{\text{shape}}. \tag{20}$$

For the regime-aware prior, the cost is a random variable dependent on the historical observation $\mathbf{y}_p$. Taking the expectation over the observation noise yields:

$$\begin{aligned} \mathbb{E}_{\mathbf{y}_p}\left[W_2^2(P_{\text{BLR}}(\mathbf{y}_p), Q)\right] &= \mathbb{E}_{\mathbf{y}_p}\left[\|\boldsymbol{\mu}_{\text{prior}} - \boldsymbol{\mu}_Q\|^2 + C_{\text{shape}}\right] \\ &= R_{\text{BLR}} + C_{\text{shape}}. \end{aligned} \tag{21}$$

Defining the reduction in expected transport cost as $\Delta\mathcal{E} := W_2^2(P_{\text{GA}}, Q) - \mathbb{E}_{\mathbf{y}_p}[W_2^2(P_{\text{BLR}}, Q)]$, we find:

$$\Delta\mathcal{E} = (\|\boldsymbol{\mu}_Q\|^2 + C_{\text{shape}}) - (R_{\text{BLR}} + C_{\text{shape}}) = \|\boldsymbol{\mu}_Q\|^2 - R_{\text{BLR}}. \tag{22}$$

Under the condition established in Proposition 3.1, we have $R_{\text{BLR}} < \|\boldsymbol{\mu}_Q\|^2$, which implies $\Delta\mathcal{E} > 0$.

Thus, the regime-aware prior strictly reduces the expected Wasserstein-2 transport cost under the conditions of Proposition 3.1.

$\square$

# B. Additional Experiments

## B.1. Empirical Validation of Theoretical Bounds

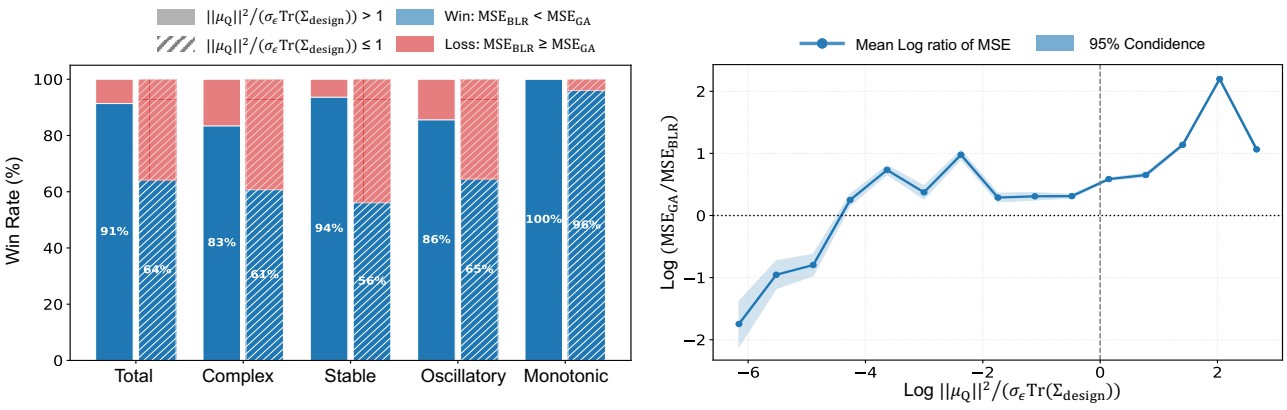

*Figure 8.* **Left:** Empirical win rate of the our BLR-induced regime-aware prior over an isotropic Gaussian initialization (ISO, $\mathcal{N}(\mathbf{0}, \sigma_{\text{init}}^2 \mathbf{I})$) across regimes. **Right:** Empirical relationship between the theoretical bound term in Proposition 3.1 and the observed error ratio.

To assess the practical utility of Proposition 3.1 in the finite-sample regime, we analyze the relationship between the theoretical bounds and the empirical performance gap of the BLR-induced and isotropic Gaussian (ISO, $\mathcal{N}(\mathbf{0}, \sigma_{\text{init}}^2 \mathbf{I})$)

*Table 3.* Comparing a fixed-variance prior to the BLR-induced covariance $\Sigma_{\text{BLR}}$ on our benchmark. Each entry is reported as mean and $\pm$std. Best mean results are in **bold** and second-best are underlined.

| | Overall | | | Complex | | | Stable | | | Oscillatory | | | Monotonic | | |
|---|---|---|---|---|---|---|---|---|---|---|---|---|---|---|---|
| Model | MSE↓ | MAE↓ | CRPS↓ | MSE↓ | MAE↓ | CRPS↓ | MSE↓ | MAE↓ | CRPS↓ | MSE↓ | MAE↓ | CRPS↓ | MSE↓ | MAE↓ | CRPS↓ |
| Ours$^{\dagger}$ ($\sigma_{\text{init}}^2$) | **0.012** | **0.044** | **0.039** | **0.034** | **0.096** | **0.086** | **0.006** | **0.027** | **0.023** | **0.035** | **0.111** | **0.098** | **0.006** | **0.033** | **0.028** |
| | ±0.002 | ±0.006 | ±0.006 | ±0.002 | ±0.011 | ±0.010 | ±0.001 | ±0.003 | ±0.003 | ±0.002 | ±0.008 | ±0.007 | ±0.000 | ±0.002 | ±0.002 |
| Ours$^{\dagger}$ ($\Sigma_{\text{BLR}}$) | 0.014 | 0.050 | 0.046 | 0.042 | 0.110 | 0.103 | 0.007 | 0.032 | 0.029 | 0.036 | 0.113 | 0.107 | 0.006 | 0.036 | 0.033 |
| | ±0.003 | ±0.007 | ±0.007 | ±0.007 | ±0.018 | ±0.018 | ±0.001 | ±0.002 | ±0.002 | ±0.003 | ±0.007 | ±0.008 | ±0.001 | ±0.001 | ±0.001 |

*Table 4.* Backbone ablation comparing SSM-based and self-attention models on our benchmark. Each entry is reported as mean and ±std. Best mean results are in **bold** and second-best are underlined.

| | Overall | | | Complex | | | Stable | | | Oscillatory | | | Monotonic | | |
|---|---|---|---|---|---|---|---|---|---|---|---|---|---|---|---|
| Model | MSE↓ | MAE↓ | CRPS↓ | MSE↓ | MAE↓ | CRPS↓ | MSE↓ | MAE↓ | CRPS↓ | MSE↓ | MAE↓ | CRPS↓ | MSE↓ | MAE↓ | CRPS↓ |
| Ours$^{\dagger}$ (SSM) | **0.012** | **0.044** | **0.039** | 0.034 | **0.096** | **0.086** | **0.006** | **0.027** | **0.023** | **0.035** | **0.111** | **0.098** | **0.006** | **0.033** | **0.028** |
| | ±0.002 | ±0.006 | ±0.006 | ±0.002 | ±0.011 | ±0.010 | ±0.001 | ±0.003 | ±0.003 | ±0.002 | ±0.008 | ±0.007 | ±0.000 | ±0.002 | ±0.002 |
| Ours$^{\dagger}$ (Attention) | 0.012 | 0.046 | 0.041 | **0.033** | 0.097 | 0.087 | 0.006 | 0.028 | 0.025 | 0.036 | 0.114 | 0.100 | 0.006 | 0.034 | 0.030 |
| | ±0.001 | ±0.006 | ±0.005 | ±0.000 | ±0.007 | ±0.006 | ±0.001 | ±0.003 | ±0.003 | ±0.001 | ±0.008 | ±0.007 | ±0.001 | ±0.002 | ±0.002 |

initializations. As illustrated in Figure 8 (Left), the BLR prior outperforms the ISO baseline across the majority of dynamic regimes. Crucially, this empirical advantage exhibits a strong alignment with our theoretical derivation. Figure 8 (Right) reveals a positive correlation between the magnitude of the theoretical bound and the observed log-error ratio. This result indicates that although Proposition 3.1 is derived under an asymptotic limit ($\alpha \to 0$), it serves as a robust and informative proxy for $\alpha > 0$ settings. Thus, this bound effectively identifies scenarios where the mechanistic prior yields the most significant reduction in the Wasserstein transport cost, further validating the BLR initialization's superiority over the uninformed isotropic baseline.

## B.2. Empirical Benefits of a Covariance-Regularized Prior

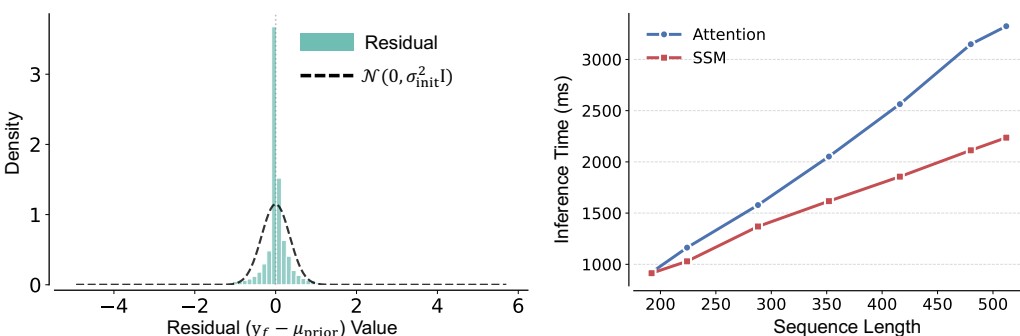

*Figure 9.* **Left:** Residual distribution of $\mathbf{y} - \boldsymbol{\mu}_{\text{prior}}$ under the regime-aware prior. **Right:** Inference-time scaling of SSM and self-attention backbones with increasing prediction horizon, highlighting the superior long-horizon efficiency of SSM-based models.

In practice, we instantiate the prior with a fixed variance $\sigma_{\text{init}}^2$ rather than the BLR-induced $\Sigma_{\text{BLR}}$. As shown in Table 3, using the full BLR predictive covariance already yields strong accuracy (overall MSE 0.014), confirming that the Bayesian predictive distribution provides useful source information. Regularizing this covariance to an isotropic form further improves overall MSE/MAE/CRPS from $0.014/0.050/0.046$ to $0.012/0.044/0.039$, consistent with the mean-alignment argument in Section 3.3.4. Furthermore, Figure 9(left) shows that the residual $\mathbf{y} - \boldsymbol{\mu}_{\text{prior}}$ is well-approximated by a Gaussian, supporting the constant-variance approximation in this regime.

## B.3. Architecture Analysis: Attention and SSM

We ablate the sequence module by replacing the Selective State Space Model (SSM) with a standard self-attention block. As shown in Table 4, the accuracy is largely preserved under this substitution (MSE/MAE/CRPS: $0.012/0.044/0.039$ for SSM

*Table 5.* Performance comparison of TSFlow with different kernels at NFE=1 and NFE=16. Each entry is reported as mean and ±std. Best mean results are in **bold** and second-best are underlined.

| Model | Overall | | | Complex | | | Stable | | | Oscillatory | | | Monotonic | | |
|---|---|---|---|---|---|---|---|---|---|---|---|---|---|---|---|
| | MSE↓ | MAE↓ | CRPS↓ | MSE↓ | MAE↓ | CRPS↓ | MSE↓ | MAE↓ | CRPS↓ | MSE↓ | MAE↓ | CRPS↓ | MSE↓ | MAE↓ | CRPS↓ |
| *NFE = 1* | | | | | | | | | | | | | | | |
| TSFlow-ISO | 0.026 ±0.002 | 0.081 ±0.017 | 0.074 ±0.015 | 0.042 ±0.001 | **0.112** ±0.013 | 0.104 ±0.011 | 0.017 ±0.003 | 0.062 ±0.021 | 0.057 ±0.019 | 0.085 ±0.013 | 0.208 ±0.007 | 0.197 ±0.008 | 0.009 ±0.004 | 0.051 ±0.020 | 0.045 ±0.018 |
| TSFlow-OU | **0.025** ±0.003 | 0.081 ±0.012 | 0.074 ±0.010 | 0.041 ±0.002 | 0.115 ±0.011 | 0.107 ±0.010 | 0.017 ±0.005 | 0.063 ±0.016 | 0.057 ±0.013 | 0.079 ±0.008 | 0.200 ±0.005 | 0.190 ±0.006 | 0.009 ±0.002 | 0.050 ±0.007 | 0.044 ±0.005 |
| TSFlow-SE | 0.026 ±0.001 | 0.082 ±0.014 | 0.074 ±0.011 | **0.040** ±0.001 | 0.112 ±0.013 | **0.103** ±0.011 | 0.018 ±0.002 | 0.065 ±0.017 | 0.058 ±0.014 | **0.079** ±0.013 | **0.198** ±0.012 | **0.187** ±0.013 | 0.009 ±0.003 | 0.053 ±0.017 | 0.046 ±0.014 |
| TSFlow-PE | 0.025 ±0.003 | **0.078** ±0.011 | **0.072** ±0.010 | 0.042 ±0.001 | 0.112 ±0.009 | 0.104 ±0.009 | **0.017** ±0.003 | **0.059** ±0.012 | **0.054** ±0.011 | 0.084 ±0.006 | 0.208 ±0.006 | 0.195 ±0.005 | **0.008** ±0.002 | **0.047** ±0.010 | **0.041** ±0.009 |
| *NFE = 16* | | | | | | | | | | | | | | | |
| TSFlow-ISO | 0.027 ±0.002 | 0.080 ±0.017 | 0.059 ±0.011 | 0.043 ±0.002 | 0.112 ±0.012 | 0.086 ±0.008 | 0.018 ±0.004 | 0.060 ±0.020 | 0.043 ±0.014 | 0.088 ±0.013 | 0.208 ±0.008 | 0.159 ±0.012 | 0.010 ±0.004 | 0.051 ±0.019 | 0.040 ±0.014 |
| TSFlow-OU | **0.026** ±0.003 | 0.077 ±0.012 | 0.056 ±0.008 | 0.043 ±0.002 | 0.113 ±0.010 | 0.086 ±0.013 | 0.018 ±0.005 | 0.057 ±0.015 | 0.039 ±0.010 | 0.083 ±0.007 | 0.199 ±0.004 | 0.152 ±0.007 | 0.010 ±0.002 | 0.049 ±0.010 | 0.037 ±0.009 |
| TSFlow-SE | 0.027 ±0.002 | 0.079 ±0.010 | 0.057 ±0.004 | **0.043** ±0.001 | 0.112 ±0.012 | **0.085** ±0.007 | 0.019 ±0.002 | 0.061 ±0.012 | 0.042 ±0.005 | **0.081** ±0.011 | **0.198** ±0.011 | **0.150** ±0.013 | 0.010 ±0.004 | 0.053 ±0.016 | 0.040 ±0.010 |
| TSFlow-PE | 0.026 ±0.003 | **0.075** ±0.011 | **0.055** ±0.007 | 0.044 ±0.001 | **0.111** ±0.009 | 0.086 ±0.008 | **0.017** ±0.004 | **0.055** ±0.012 | **0.038** ±0.008 | 0.088 ±0.007 | 0.208 ±0.007 | 0.158 ±0.006 | **0.008** ±0.002 | **0.044** ±0.008 | **0.033** ±0.006 |

vs. 0.012/0.046/0.041 for Attention), indicating that the principal error reductions are not driven solely by the choice of sequence model. However, Figure 9 (right) indicates that SSM achieves lower inference time as the prediction length grows, making it preferable for long-term trajectory generation when computational scaling is a dominant constraint.

## B.4. TSFlow Baseline Results under Different Prior Kernels

To establish a rigorous baseline, we evaluated the TSFlow architecture against the full spectrum of kernel configurations proposed in the original study. As detailed in Table 5, the variant equipped with the periodic kernel (TSFlow-PE) demonstrates superior pointwise accuracy compared to alternative instantiations on overall results. Crucially, this performance advantage extends to distributional metrics. Table 5 also confirms that TSFlow-PE achieves the lowest overall CRPS, validating its selection as the primary comparator for both deterministic and probabilistic assessments.

## B.5. Sensitivity to Number of Function Evaluations

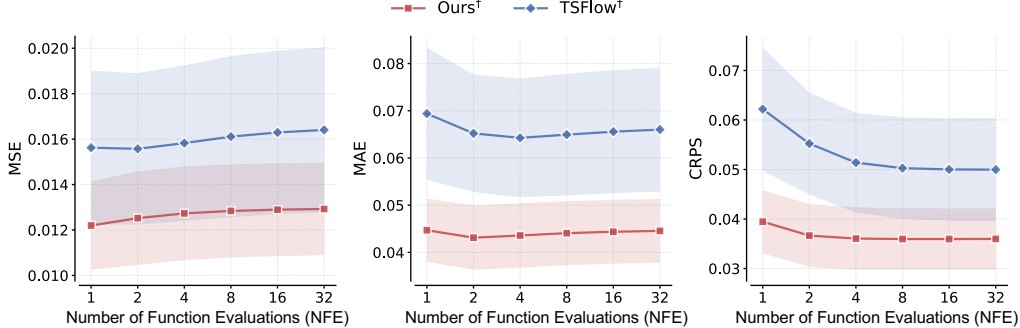

*Figure 10.* Impact of the Number of Function Evaluations (NFE) on forecasting performance. Our method achieves near-optimal overall results with a single step (NFE=1). In contrast, the baseline TSFlow requires significantly more steps (NFE ≥ 16) to converge to the target distribution.

We analyze the impact of the Number of Function Evaluations (NFE) in Figure 10. Our method achieves near-optimal deterministic and probabilistic performance at NFE = 1. This rapid convergence stems from the regime-aware prior, which structurally approximates the target to minimize the source data distributional mismatch, maximizing inference speed while maintaining competitive accuracy. In contrast, TSFlow initiates with uninformative prior, necessitating significantly more

*Table 6.* Computational cost comparison on a single NVIDIA RTX A6000 GPU. Training time is reported in hours, and inference time is reported per sample.

| Model | Family | Params | Train (h) | Infer. (ms) | NFE |
|---|---|---|---|---|---|
| DLinear | Point | 49.7 K | 0.3 | 0.092 | – |
| iTransformer | Point | 4.9 M | 0.5 | 0.966 | – |
| PatchTST | Point | 4.7 M | 0.6 | 0.800 | – |
| TimeMixer | Point | 133 K | 0.5 | 3.135 | – |
| TimeXer | Point | 11.5 M | 0.5 | 2.104 | – |
| SMamba | Point | 3.0 M | 0.5 | 2.300 | – |
| BiMamba4TS | Point | 3.1 M | 1.5 | 2.496 | – |
| NSformer | Point | 2.0 M | 5 | 3.000 | – |
| CSDI[†] | Diffusion | 413 K | 8 | 90.8 | 50 |
| TSDiff[†] | Diffusion | 946 K | 10 | 552.6 | 100 |
| TSFlow[†] | Flow | 1.9 M | 6 | 71.9 | 16 |
| **Ours[†]** | **Flow** | **1.8 M** | **7** | **7.7** | **1** |

*Table 7.* Comparison with zero-shot time-series foundation models. Each entry reports the mean with standard deviation. Best and second-best performances are highlighted in **bold** and underline, respectively.

| Model | Overall | | Complex | | Stable | | Oscillatory | | Monotonic | |
|---|---|---|---|---|---|---|---|---|---|---|
| | MSE↓ | MAE↓ | MSE↓ | MAE↓ | MSE↓ | MAE↓ | MSE↓ | MAE↓ | MSE↓ | MAE↓ |
| TimeMoE | $0.120_{\pm 0.004}$ | $0.231_{\pm 0.010}$ | $0.099_{\pm 0.013}$ | $0.191_{\pm 0.032}$ | $0.118_{\pm 0.003}$ | $0.221_{\pm 0.010}$ | $\underline{0.138}_{\pm 0.005}$ | $\underline{0.268}_{\pm 0.005}$ | $0.138_{\pm 0.012}$ | $0.289_{\pm 0.023}$ |
| Sundial | $0.111_{\pm 0.003}$ | $0.217_{\pm 0.007}$ | $\underline{0.095}_{\pm 0.012}$ | $0.185_{\pm 0.031}$ | $0.106_{\pm 0.002}$ | $0.203_{\pm 0.008}$ | $0.141_{\pm 0.009}$ | $0.270_{\pm 0.009}$ | $0.133_{\pm 0.011}$ | $0.281_{\pm 0.022}$ |
| TimesFM | $0.099_{\pm 0.005}$ | $\underline{0.152}_{\pm 0.004}$ | $0.109_{\pm 0.009}$ | $\underline{0.145}_{\pm 0.021}$ | $0.086_{\pm 0.002}$ | $\underline{0.138}_{\pm 0.006}$ | $0.249_{\pm 0.032}$ | $0.299_{\pm 0.014}$ | $\underline{0.045}_{\pm 0.004}$ | $\underline{0.112}_{\pm 0.007}$ |
| Chronos | $\underline{0.097}_{\pm 0.004}$ | $0.167_{\pm 0.000}$ | $0.144_{\pm 0.021}$ | $0.181_{\pm 0.012}$ | $\underline{0.078}_{\pm 0.003}$ | $0.147_{\pm 0.002}$ | $0.234_{\pm 0.013}$ | $0.309_{\pm 0.007}$ | $0.054_{\pm 0.004}$ | $0.153_{\pm 0.005}$ |
| **Ours[†]** | $\mathbf{0.012}_{\pm 0.002}$ | $\mathbf{0.044}_{\pm 0.006}$ | $\mathbf{0.034}_{\pm 0.002}$ | $\mathbf{0.096}_{\pm 0.011}$ | $\mathbf{0.006}_{\pm 0.001}$ | $\mathbf{0.027}_{\pm 0.003}$ | $\mathbf{0.035}_{\pm 0.002}$ | $\mathbf{0.111}_{\pm 0.008}$ | $\mathbf{0.006}_{\pm 0.000}$ | $\mathbf{0.033}_{\pm 0.002}$ |

steps (NFE $\geq$ 16) to adequately resolve distributional features. Consequently, to ensure a fair comparison at peak capacity, we adopt NFE = 1 for our framework and NFE = 16 for TSFlow (similarly to (Kollovieh et al., 2024)) in all experiments.

## B.6. Computational Cost Comparison

We provide a comprehensive computational cost comparison across all evaluated models in Table 6. All measurements were conducted on a single NVIDIA RTX A6000 GPU.

When considered together with the predictive results in Table 1, these measurements show a clear trade-off between predictive performance and computational cost. Point forecasting methods provide fast training and inference (e.g., DLinear: 0.3 h training and 0.092 ms per sample), but have higher predictive errors on the main benchmark. Existing probabilistic baselines provide uncertainty-aware forecasts but often incur substantially higher inference cost, with diffusion-based CSDI[†] and TSDiff[†] requiring 50–100 function evaluations and over 90 ms per sample. TSFlow[†] reduces the NFE to 16 but still requires 71.9 ms per sample. In contrast, RegimeFlow (Ours[†]) performs inference in 7.7 ms with a single function evaluation (NFE = 1), making it 9.3× faster than TSFlow[†]. With 1.8 M parameters and approximately 7 h training time, RegimeFlow keeps training cost comparable to standard probabilistic baselines while substantially reducing inference overhead.

## B.7. Comparison with Time-Series Foundation Models

We further compare RegimeFlow against four representative zero-shot time-series foundation models (TSFMs): TimeMoE (Shi et al., 2025), Sundial (Liu et al., 2025b), TimesFM (Das et al., 2024), and Chronos (Ansari et al., 2024). These models are pretrained on large-scale general-purpose time-series corpora and support zero-shot inference, making them natural baselines for evaluating the advantage of domain-specific biological priors in cross-system generalization. All TSFMs are evaluated in the zero-shot setting on the same test trajectories and prediction horizons used in the main benchmark, without task-specific fine-tuning.

Table 7 reports the results. Despite their massive pretraining scale (128–200 M parameters), all four TSFMs yield substantially higher errors than RegimeFlow. The strongest foundation model in the overall comparison, Chronos, achieves an overall MSE of 0.097, compared to 0.012 for Ours[†] with only 1.8 M parameters. The gap is also pronounced on the Oscillatory subset:

*Table 8.* Performance on transient and hybrid dynamical regimes under the $L = 96, T = 416$ extrapolation setting. Each entry reports the mean.

| Case | System | Ours[†] | | TSFlow[†] | |
|---|---|---|---|---|---|
| | | MSE↓ | MAE↓ | MSE↓ | MAE↓ |
| Complex Oscil. | Golomb2006 | 0.024 | 0.092 | 0.135 | 0.275 |
| Oscil.→Stable | Sharp2013 | 0.035 | 0.116 | 0.046 | 0.159 |
| Damped Oscil. | Radulescu2008 | 0.051 | 0.151 | 0.086 | 0.216 |

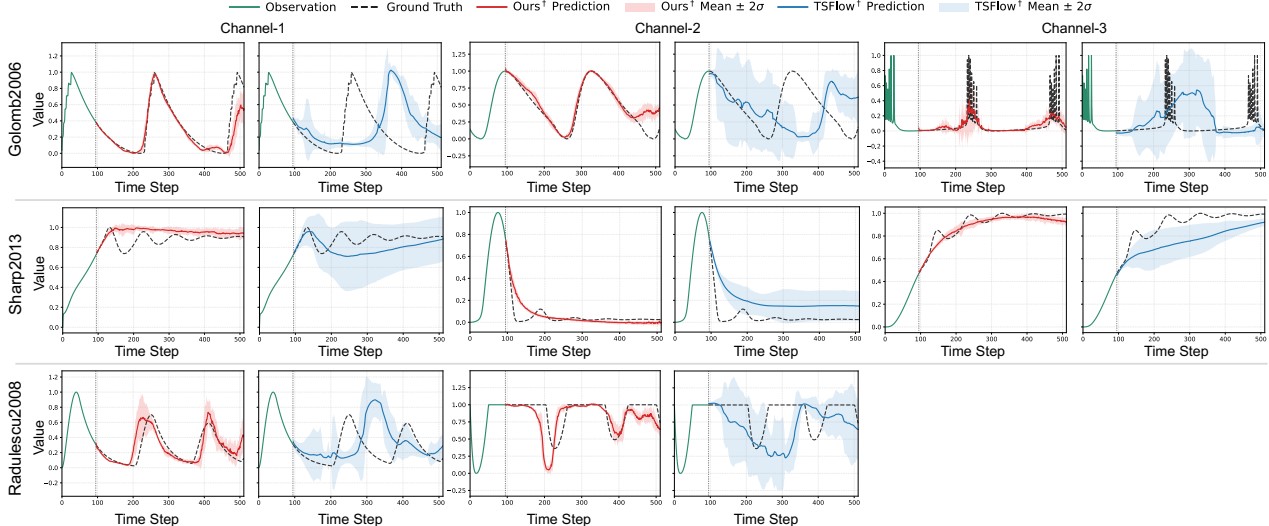

*Figure 11.* Representative predictions on transient and hybrid regimes under the $L = 96, T = 416$ extrapolation setting. The selected cases include complex oscillations (Golomb2006), oscillation-to-stable transition (Sharp2013), and damped oscillations (Radulescu2008).

the best TSFM reaches an MSE of $0.138$, whereas Ours[†] achieves $0.035$. Chronos and TimesFM show even larger errors on this regime, suggesting that generic pretraining alone does not reliably capture sustained biological oscillations. This pattern indicates that general-purpose TSFMs, while strong zero-shot forecasting baselines, do not encode the biological regime structure required by this benchmark. In contrast, RegimeFlow combines a regime-aware prior with a condition-modulated sequence backbone, providing structural guidance tailored to long-horizon biological dynamics.

### B.8. Validation on Transient and Hybrid Dynamical Regimes

To assess whether RegimeFlow remains effective beyond the three canonical regime categories, we evaluate it on three representative biological systems that exhibit transient or hybrid dynamical behaviors under a longer extrapolation setting with $L = 96$ observed points and a $T = 416$ prediction horizon: **Golomb2006** (complex oscillations with no dominant single-regime pattern) (Golomb et al., 2006), **Sharp2013** (oscillatory dynamics that transition to a stable steady state) (Sharp et al., 2013), and **Radulescu2008** (decreasing-amplitude oscillations that dampen over time) (Radulescu et al., 2008). These cases are deliberately chosen because they do not fit neatly into a single macroscopic regime, testing whether the regime-aware prior provides flexible guidance rather than a rigid constraint. As shown in Table 8 and Figure 11, RegimeFlow obtains lower errors than TSFlow[†] across all three representative transient cases. On Golomb2006 (complex oscillation), RegimeFlow reduces MSE from $0.135$ to $0.024$, an 82% improvement, suggesting that the structured prior remains useful even when the dynamics do not follow a single canonical pattern. On Sharp2013 (oscillation-to-stable transition), both methods show higher error, as expected for a regime-switching system; RegimeFlow's MSE of $0.035$ versus $0.046$ for TSFlow[†] suggests that the prior provides partial guidance without over-constraining the flow. On Radulescu2008 (decreasing oscillation), RegimeFlow achieves an MSE of $0.051$ compared to $0.086$ for TSFlow[†], a 41% improvement. These results indicate that RegimeFlow's regime-aware prior can still capture the underlying structure of the dynamics, even when the system exhibits transient or hybrid behaviors that do not fit neatly into a single macroscopic regime category.

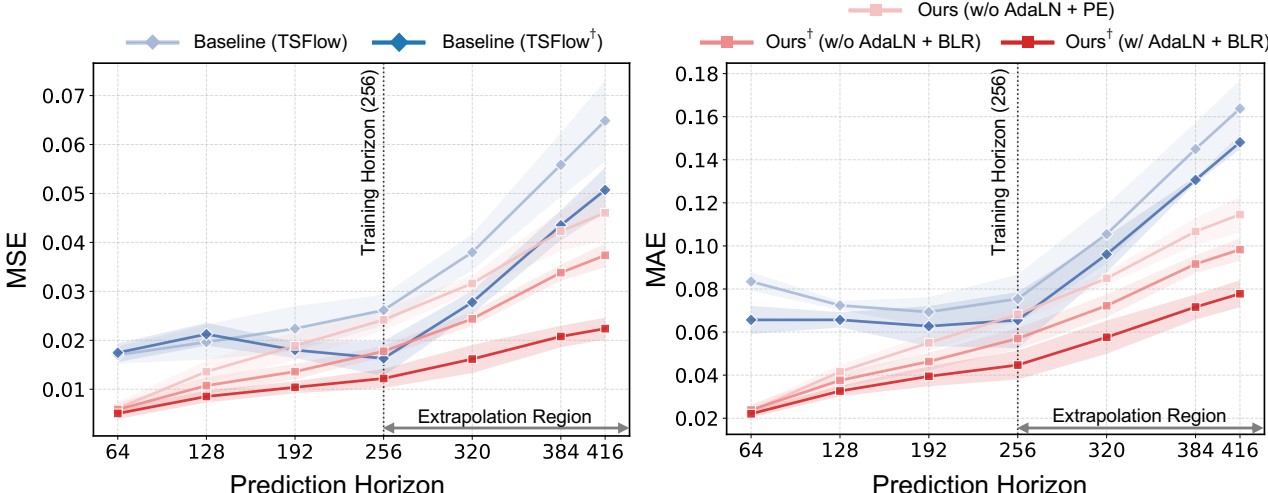

*Figure 12.* Extrapolation performance across extended prediction lengths. PE denotes the baseline initialized with a Periodic Kernel prior, while BLR represents our proposed regime-aware prior induced by Bayesian Linear Regression. Models trained on a fixed length ($T = 256$) are evaluated on longer trajectories, demonstrating the superior generalization of our proposed framework.

## B.9. Robustness to Extrapolation Length

### B.9.1. QUANTITATIVE EVALUATION OF LONG-HORIZON EXTRAPOLATION

To evaluate the generalization capability of our framework, we investigate its performance when extrapolating to prediction lengths significantly beyond the training setup ($T_{\text{train}} = 256$). As illustrated in Figure 12, our full model (w/ AdaLN + BLR) maintains high predictive fidelity across extended lengths, effectively mitigating the degradation typically observed in autoregressive or iterative generation.

Disentangling the drivers of this robustness yields two primary insights. First, the architectural superiority of our framework is demonstrated by the performance gap between our Periodic (PE) variant (w/o AdaLN + PE) and the TSFlow baseline. Unlike the standard S4-based architecture employed in TSFlow, our backbone leverages the enhanced receptivity of modern Selective State Space Models to capture long-range dependencies. Second, a comparison of the conditioned variants underscores the essential role of the prior. The model initialized with the BLR-induced distribution (w/o AdaLN + BLR) consistently outperforms both the TSFlow[†] model and our own model initialized with the PE kernel (w/o AdaLN + PE). This result highlights the superior efficacy of the BLR prior in capturing underlying dynamic patterns compared to the PE kernel. This indicates that the mechanistic prior encodes invariant dynamic properties that persist beyond the training window, whereas the standard PE kernel struggles to generalize to unseen temporal coordinates.

### B.9.2. QUALITATIVE VISUALIZATION OF EXTENDED TRAJECTORY PREDICTIONS

As shown in Figure 13, we select five dynamical systems with qualitatively different regime patterns, each of which can be described by mechanistic ordinary differential equation models commonly used in prior studies (Tomasz et al., 2004; Fribourg et al., 2014; Pritchard & Birch, 2014; Yildirim & Mackey, 2003; Bungay et al., 2006).

Overall, our method demonstrates strong adaptability: even with a long extrapolation horizon of $T = 416$, it can still capture the global trajectory trend with high fidelity. When the underlying series exhibits regime changes, Ours[†](w/o AdaLN) already tracks most regime-wise trend variations well, highlighting the effectiveness of the regime-aware prior during extrapolation. When the series is relatively regime-agnostic, the comparison between Ours-PE(w/o AdaLN) and TSFlow shows that our backbone yields more accurate predictions, supporting the advantage of our backbone design.

As illustrated in Figure 14, we selected five representative oscillatory systems exhibiting distinct temporal patterns, including cell division cycles and genetic network dynamics (Tyson, 1991; Locke et al., 2005; Li et al., 2009; Adams et al., 2012; Conradie et al., 2010).

The comparative analysis reveals that our proposed method consistently achieves superior predictive fidelity across these

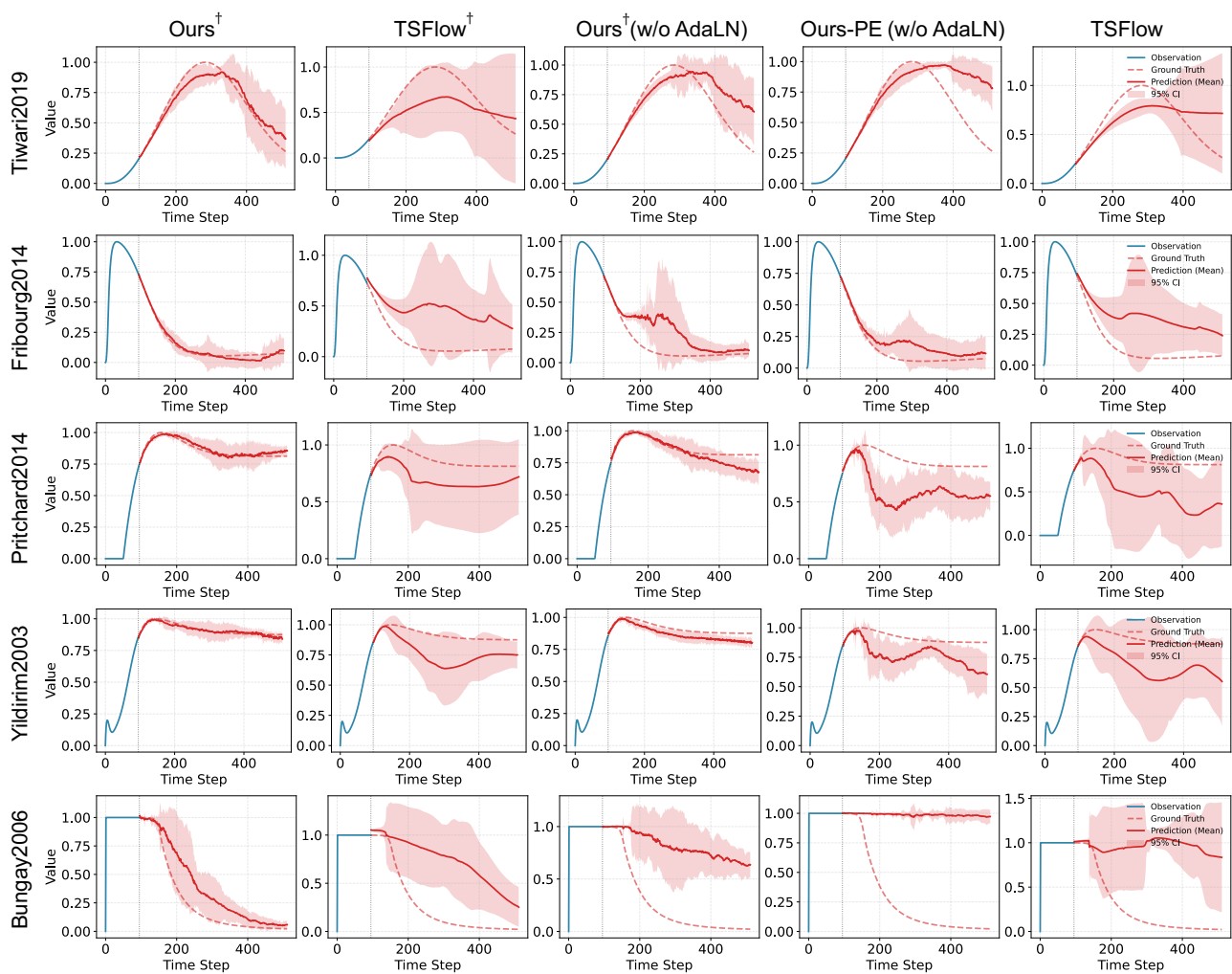

*Figure 13.* Qualitative comparison of extrapolation trajectories ($T = 416$) across five systems exhibiting distinct regimes. We compare Ours[†] against the state-of-the-art baseline TSFlow and two ablations: Ours[†] (w/o AdaLN) and Ours-PE (w/o AdaLN). Ours-PE denotes the variant initialized with a standard Periodic Kernel prior, while Ours[†] employs our proposed regime-aware prior. The visualization highlights the model's capability to capture long-term trends under diverse dynamic patterns.

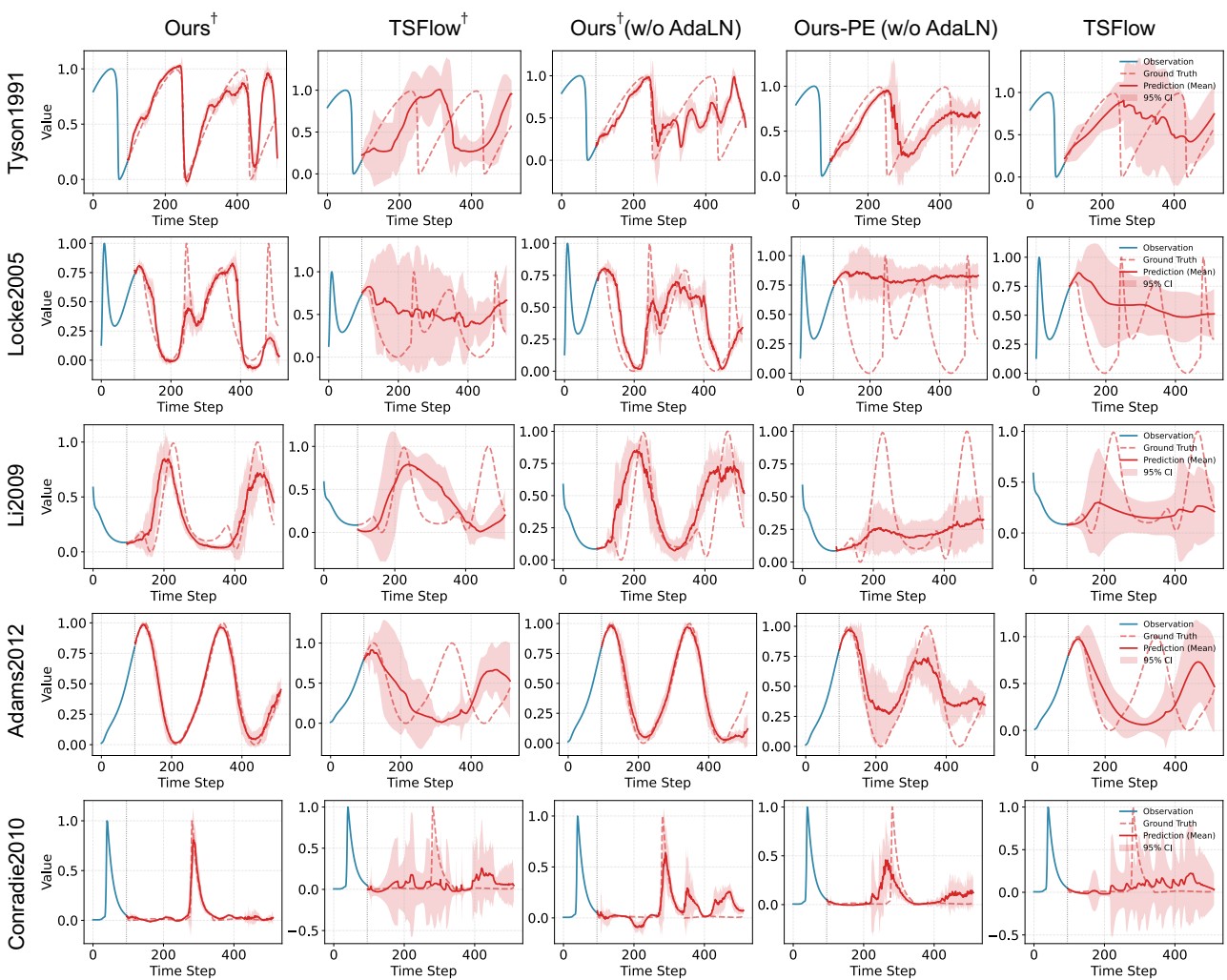

*Figure 14.* Qualitative visualization of predicted trajectories across five distinct oscillatory biological systems. We compare the extrapolation performance ($T = 416$) of Ours[†] against the baseline TSFlow[†] and ablation variants (Ours[†] w/o AdaLN, Ours-PE w/o AdaLN). The results demonstrate that our regime-aware prior initialization significantly enhances the capture of diverse oscillatory patterns compared to standard periodic kernel methods.

*Table 9.* Robustness evaluation under measurement noise and irregular sampling. Each entry reports the mean with standard deviation. $\Delta$MSE is computed on the overall MSE as $(1 - \text{Ours}^{\dagger}/\text{TSFlow}^{\dagger}) \times 100\%$. The Clean setting from the main benchmark (Table 1) is included as reference.

| Setting | Model | Overall | | | Complex | | Stable | | Oscillatory | | Monotonic | |
|---|---|---|---|---|---|---|---|---|---|---|---|---|
| | | MSE↓ | MAE↓ | ΔMSE | MSE↓ | MAE↓ | MSE↓ | MAE↓ | MSE↓ | MAE↓ | MSE↓ | MAE↓ |
| Clean | Ours$^{\dagger}$ | $0.012_{\pm0.002}$ | $0.044_{\pm0.006}$ | 25.0% | $0.034_{\pm0.002}$ | $0.096_{\pm0.011}$ | $0.006_{\pm0.001}$ | $0.027_{\pm0.003}$ | $0.035_{\pm0.002}$ | $0.111_{\pm0.008}$ | $0.006_{\pm0.000}$ | $0.033_{\pm0.002}$ |
| | TSFlow$^{\dagger}$ | $0.016_{\pm0.004}$ | $0.064_{\pm0.013}$ | | $0.039_{\pm0.005}$ | $0.111_{\pm0.022}$ | $0.008_{\pm0.002}$ | $0.046_{\pm0.010}$ | $0.044_{\pm0.005}$ | $0.139_{\pm0.017}$ | $0.008_{\pm0.001}$ | $0.050_{\pm0.005}$ |
| Noisy-0.1 | Ours$^{\dagger}$ | $0.041_{\pm0.004}$ | $0.134_{\pm0.009}$ | 8.9% | $0.089_{\pm0.019}$ | $0.200_{\pm0.024}$ | $0.031_{\pm0.005}$ | $0.117_{\pm0.013}$ | $0.074_{\pm0.014}$ | $0.188_{\pm0.017}$ | $0.027_{\pm0.003}$ | $0.121_{\pm0.004}$ |
| | TSFlow$^{\dagger}$ | $0.045_{\pm0.003}$ | $0.140_{\pm0.006}$ | | $0.089_{\pm0.011}$ | $0.199_{\pm0.018}$ | $0.031_{\pm0.005}$ | $0.117_{\pm0.009}$ | $0.081_{\pm0.007}$ | $0.197_{\pm0.015}$ | $0.046_{\pm0.010}$ | $0.154_{\pm0.019}$ |
| Noisy-0.3 | Ours$^{\dagger}$ | $0.059_{\pm0.002}$ | $0.178_{\pm0.003}$ | 11.9% | $0.098_{\pm0.014}$ | $0.233_{\pm0.015}$ | $0.050_{\pm0.002}$ | $0.164_{\pm0.008}$ | $0.099_{\pm0.012}$ | $0.234_{\pm0.012}$ | $0.040_{\pm0.010}$ | $0.160_{\pm0.022}$ |
| | TSFlow$^{\dagger}$ | $0.067_{\pm0.014}$ | $0.187_{\pm0.021}$ | | $0.104_{\pm0.018}$ | $0.230_{\pm0.025}$ | $0.048_{\pm0.012}$ | $0.161_{\pm0.023}$ | $0.110_{\pm0.009}$ | $0.242_{\pm0.007}$ | $0.094_{\pm0.044}$ | $0.232_{\pm0.053}$ |
| Irregular-48 | Ours$^{\dagger}$ | $0.038_{\pm0.005}$ | $0.123_{\pm0.010}$ | 29.6% | $0.074_{\pm0.013}$ | $0.181_{\pm0.025}$ | $0.027_{\pm0.002}$ | $0.103_{\pm0.008}$ | $0.102_{\pm0.006}$ | $0.241_{\pm0.008}$ | $0.014_{\pm0.002}$ | $0.075_{\pm0.005}$ |
| | TSFlow$^{\dagger}$ | $0.054_{\pm0.009}$ | $0.163_{\pm0.018}$ | | $0.097_{\pm0.010}$ | $0.214_{\pm0.016}$ | $0.039_{\pm0.007}$ | $0.144_{\pm0.016}$ | $0.124_{\pm0.010}$ | $0.263_{\pm0.013}$ | $0.042_{\pm0.014}$ | $0.139_{\pm0.026}$ |
| Irregular-24 | Ours$^{\dagger}$ | $0.080_{\pm0.004}$ | $0.202_{\pm0.008}$ | 15.8% | $0.111_{\pm0.015}$ | $0.234_{\pm0.021}$ | $0.072_{\pm0.001}$ | $0.195_{\pm0.005}$ | $0.150_{\pm0.006}$ | $0.297_{\pm0.007}$ | $0.036_{\pm0.005}$ | $0.131_{\pm0.009}$ |
| | TSFlow$^{\dagger}$ | $0.095_{\pm0.012}$ | $0.233_{\pm0.017}$ | | $0.133_{\pm0.016}$ | $0.271_{\pm0.023}$ | $0.074_{\pm0.011}$ | $0.210_{\pm0.017}$ | $0.195_{\pm0.012}$ | $0.348_{\pm0.011}$ | $0.089_{\pm0.019}$ | $0.226_{\pm0.027}$ |

diverse oscillation modes. Crucially, the performance gap between Ours$^{\dagger}$ (including its variant w/o AdaLN) and the regime-agnostic Ours-PE (w/o AdaLN) underscores the critical role of the regime-aware prior. This regime-aware prior provides a robust structural initialization that is essential for accurate long-term extrapolation. In contrast, although TSFlow$^{\dagger}$ also incorporates regime information, its performance is constrained by a less accurate initial trajectory shape relative to our approach. This discrepancy in initialization induces optimization bias in later stages, preventing the baseline from fully recovering the true oscillatory dynamics.

### B.10. Robustness to Measurement Noise and Irregular Sampling

To assess the robustness of RegimeFlow under conditions that approximate real-world experimental data, we evaluate both RegimeFlow (Ours$^{\dagger}$) and TSFlow$^{\dagger}$ under two forms of data degradation: (1) **measurement noise**, where additive Gaussian perturbations $x + s\epsilon$ with $\epsilon \sim \mathcal{N}(0,1)$ and $s \in \{0.1, 0.3\}$ are applied to the observed trajectories; and (2) **irregular sampling**, where the observation window is randomly subsampled, retaining only 48 or 24 out of the original 96 time points.

As shown in Table 9, RegimeFlow consistently outperforms TSFlow$^{\dagger}$ across all overall robustness settings. Under mild noise ($s=0.1$), the regime-aware prior preserves an 8.9% MSE advantage; under stronger noise ($s=0.3$), the advantage remains meaningful at 11.9%, indicating that the regime-aware prior partially attenuates noise through its closed-form projection onto the basis subspace. Under irregular sampling, RegimeFlow exhibits particularly strong robustness: with only 24 irregular observations (Irregular-24), RegimeFlow achieves an MSE of 0.080 compared to 0.095 for TSFlow$^{\dagger}$ (15.8% improvement), and with 48 observations the improvement reaches 29.6%. This resilience is attributable to the regime-aware prior's ability to recover the underlying trend from sparse observations via the selected basis family, providing a structurally informed initialization even when data density is low. Table 9 also reports regime-wise breakdowns for completeness, while $\Delta$MSE is computed from the overall results.

## C. Implementation Details

In this section, we provide a comprehensive overview of the experimental setup to facilitate reproducibility. We detail the baseline models, hyperparameter configurations, and training protocols used in our evaluation. All experiments were implemented in PyTorch and executed on a workstation equipped with a single NVIDIA A6000 GPU.

### C.1. Baseline Descriptions

To evaluate the effectiveness of our proposed framework, we benchmarked against a diverse set of state-of-the-art methods, categorized into Point Time-Series Forecasting architectures and Generative Probabilistic Forecasting models.

C.1.1. POINT TIME-SERIES FORECASTING.

We acknowledge the *Time-Series-Library*[1] for providing a unified codebase for time-series forecasting. All point forecasting baselines reported in this work were implemented based on this framework to ensure a fair and standardized comparison.

- **DLinear** (Zeng et al., 2023): A simple yet effective decomposition-based linear model that separates time series into trend and seasonal components.

- **iTransformer** (Liu et al., 2023): An inverted Transformer architecture that embeds the entire temporal sequence of each variate as a token to capture multivariate correlations.

- **PatchTST** (Nie, 2022): A channel-independent Transformer model that utilizes patching to segment time series into sub-series tokens, preserving local semantic information.

- **TimeMixer** (Wang et al., 2024a): An MLP-based architecture designed for multi-scale time-series mixing.

- **TimeXer** (Wang et al., 2024b): A Transformer-based framework specialized in empowering time-series forecasting by effectively incorporating exogenous variables.

- **NSformer** (Liu et al., 2022): A Transformer-based model specifically designed to address the challenges of non-stationary time series.

- **SMamba** (Wang et al., 2025): A selective state-space model (SSM) adapted for time-series forecasting, leveraging the efficiency of the Mamba architecture.

- **BiMamba4TS** (Liang et al., 2024): A bidirectional state-space model that extends Mamba to capture temporal dependencies from both forward and backward directions.

C.1.2. GENERATIVE PROBABILISTIC FORECASTING.

- **CSDI** (Tashiro et al., 2021): A conditional score-based diffusion model originally designed for time-series imputation.

- **TSDiff** (Kollovieh et al., 2023): A diffusion-based model that introduces a self-guidance mechanism to regularize the generation process.

- **TSFlow** (Kollovieh et al., 2024): A conditional flow matching model for probabilistic time series forecasting that combines Gaussian process priors.

C.1.3. TIME-SERIES FOUNDATION MODELS (ZERO-SHOT).

We evaluate four representative zero-shot time-series foundation models (TSFMs) to assess how large-scale general-purpose pretraining performs relative to domain-specific biological priors in cross-system trajectory prediction. All TSFMs are evaluated in the zero-shot setting on the same test trajectories.

- **TimeMoE** (Shi et al., 2025): A decoder-only sparse mixture-of-experts (MoE) Transformer, pretrained on over 300 billion time points spanning nine domains. Public checkpoint: `https://huggingface.co/Maple728/TimeMoE-200M`.

- **Sundial** (Liu et al., 2025b): A patch-based decoder-only Transformer, trained with a flow-matching objective (TimeFlow Loss) on one trillion time points. Public checkpoint: `https://huggingface.co/thuml/sundial-base-128m`.

- **TimesFM** (Das et al., 2024): A decoder-only patched Transformer, pretrained by Google Research on approximately 100 billion time points from web, retail, traffic, and energy domains. Public checkpoint: `https://huggingface.co/google/timesfm-1.0-200m`.

- **Chronos** (Ansari et al., 2024): A T5-based encoder-decoder Transformer that tokenizes time series via scaling and quantization, trained with a cross-entropy loss on 84 billion tokens from public datasets augmented with synthetic Gaussian process data. Public checkpoint: `https://huggingface.co/amazon/chronos-t5-base`.

---

[1]`https://github.com/thuml/Time-Series-Library`

*Table 10.* Hyperparameter configurations for RegimeFlow and baseline models. **Family** categorizes the modeling paradigm. **NFE** denotes the Number of Function Evaluations per inference step. $D$ and $M$ correspond to the backbone's hidden dimension and layer count, respectively.

| Model | Family | Hidden Dim ($D$) | Layers ($M$) | LR | NFE |
|---|---|---|---|---|---|
| DLinear | Point | 512 | 2 | $1 \times 10^{-3}$ | - |
| iTransformer | Point | 512 | 3 | $2 \times 10^{-4}$ | - |
| PatchTST | Point | 512 | 2 | $2 \times 10^{-4}$ | - |
| TimeMixer | Point | 16 | 3 | $2 \times 10^{-4}$ | - |
| TimeXer | Point | 512 | 4 | $1 \times 10^{-4}$ | - |
| NSformer | Point | 256 | 2 | $2 \times 10^{-4}$ | - |
| S-Mamba | Point | 256 | 4 | $1 \times 10^{-4}$ | - |
| BiMamba4TS | Point | 256 | 4 | $1 \times 10^{-4}$ | - |
| CSDI | Diffusion | 64 | 4 | $1 \times 10^{-3}$ | 50 |
| TSDiff | Diffusion | 128 | 6 | $1 \times 10^{-4}$ | 100 |
| TSFlow | Flow | 256 | 4 | $8 \times 10^{-4}$ | 16 |
| **Ours** | **Flow** | **96** | **4** | $\mathbf{8 \times 10^{-4}}$ | **1** |

## C.2. Hyperparameter Settings

We summarize the key hyperparameters used for Ours (RegimeFlow) and all baselines in Table 10. For Ours, we use $M = 4$ condition-modulated residual blocks and the backbone hidden size is $D = 96$. We employ a BLR-induced regime-aware prior with $(\alpha, \beta) = (5, 10)$. Specifically, we employ an exponential basis set $\{1 - e^{-0.2\tau}, 1 - e^{-2.0\tau}, 1 - e^{-5.0\tau}\}$, augmented with polynomial terms of degree 2 and a single harmonic component. The prior scale is initialized to $\sigma_{\text{init}} = 0.247$. All models are optimized with AdamW.

## C.3. Training Protocol

### C.3.1. OPTIMIZATION CONFIGURATION.

All experiments were conducted using 32-bit floating-point precision (FP32). The models were trained using learning rates ranging from $10^{-4}$ to $10^{-3}$ and a weight decay of $10^{-6}$. To ensure training stability, we employed a maximum batch size of 2048 and applied Exponential Moving Average (EMA) to the model parameters with a decay rate of 0.995, commencing after 200 optimization steps. The learning schedule included a warmup period of 50 epochs, and validation evaluations were performed every 10 epochs.

### C.3.2. DATA AND TRAINING SETUP.

The dataset was partitioned into training, validation, and test sets with ratios of 0.7, 0.1, and 0.2, respectively. We configured the data sequences with an input observation length of 96, a prediction horizon of 256, and a sliding window stride of 16. All models were trained for a maximum of 500 epochs, utilizing an early stopping strategy monitoring the validation MSE loss with a patience of 5 epochs.

## C.4. Regime Assignment Procedure

Regime labels are assigned **per coordinate trajectory**, not per biological system. For a model with multiple state variables, each coordinate is analyzed independently, so different variables from the same system may receive different $(c_{\text{pat}}, c_{\text{freq}})$ labels. The assignment follows a two-step procedure.

**Automated trajectory analysis.** For each coordinate trajectory, we first apply a rule-based detector to assign the discrete pattern label $c_{\text{pat}}$ and, when applicable, the frequency label $c_{\text{freq}}$. Because biological trajectories can be sinusoidal, pulse-like, square-wave, damped, or only partially observed, we combine multiple criteria rather than relying on a single test. Oscillatory trajectories are detected using peak/valley periodicity, zero-crossing regularity around the trajectory mean, and FFT-based spectral evidence for a dominant periodic component. For oscillatory coordinates, $c_{\text{freq}}$ is estimated from the dominant periodic component; otherwise, $c_{\text{freq}} = 0$. Monotonic trajectories are identified by a persistent global trend and consistent directional changes across local segments. Stable trajectories are identified by near-zero tail slope and decreasing

local variation, indicating convergence to a steady state. Coordinates that do not satisfy these criteria are assigned to the Complex category, for which we use the intercept-only fallback basis $\Phi = \{1\}$.

**Manual validation.** All automatically generated labels were manually reviewed by visually inspecting trajectory plots, with particular attention to borderline cases such as weak oscillations, damped oscillations, and slow convergence. When the automated label was clearly inconsistent with the observed trajectory shape, it was corrected.

The six fine-grained trajectory types are mapped to the four regime groups used in the paper as follows: complex → Complex; increasing-stable and decreasing-stable → Stable; oscillation → Oscillatory; monotonic increasing and monotonic decreasing → Monotonic. In the released SysBio-Traj metadata, these assignments are stored for each coordinate trajectory together with the trajectory type, bounds, and estimated period when applicable.

## D. Benchmark Statistics and Specifications for 1,050 Systems Biology Models

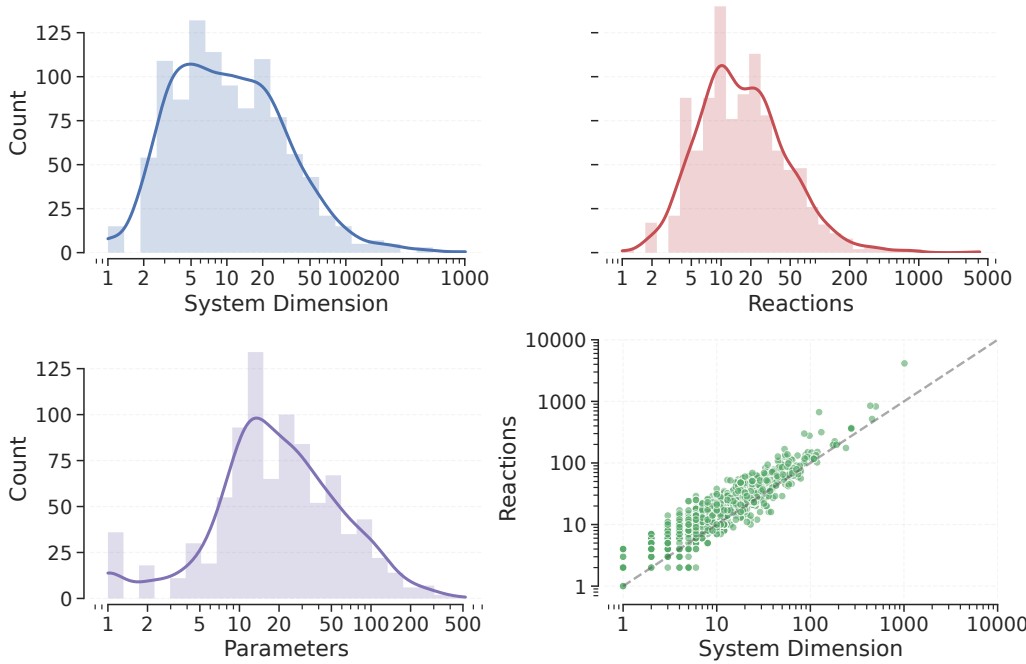

*Figure 15.* Statistical overview of the curated biological ODE benchmark. **Top Left:** Distribution of system dimensionality (number of state variables). **Top Right:** Distribution of reaction counts per system. **Bottom Left:** Distribution of parameter space size (number of ODE parameters). **Bottom Right:** Joint distribution (scatter plot) of system dimensions versus biological reactions.

In this section we analyze the structural and dynamical composition of our benchmark suite, which contains 1,050 distinct biological ODE systems sourced from (Malik-Sheriff et al., 2020).

### D.1. Benchmark Overview

Figure 1 (left) shows that the dataset spans a broad taxonomic spectrum, with mammalian systems accounting for the majority, that particularly those modeling *Homo sapiens* and *Mus musculus*. Figure 15 illustrates the statistical profile of our benchmark, comprising 1,050 biological ODE systems. It highlights the structural heterogeneity across system dimensions, reaction counts, and parameter distributions.

To rigorously assess generalization, we adopt a strict system-wise split in which training and test sets contain disjoint biological systems. This partition forces the model to learn system invariant dynamical patterns rather than memorizing system specific behavior, ensuring that predictive performance reflects genuine extrapolation capability.

**Trajectory generation.** Each system is modeled in the Systems Biology Markup Language (SBML) format and simulated using the Tellurium library (Medley et al., 2018). The SBML file is parsed to recover the governing ODEs, parameters, and initial conditions; the system dynamics are then numerically integrated to produce trajectories sampled at 512 uniformly spaced time points. Regime labels are assigned per coordinate using the automated detection and manual validation procedure described in Appendix C.4, and are stored alongside the trajectory data and model source files in the released benchmark.

## D.2. Systems Biology Model Specifications and Metadata

*Table 11.* Full list of 1,050 ODE-based models, including model ID, paper reference, biological domain, system and parameter dimensions, and Trajectory Time Span. "-" indicates unclassified taxonomy or biological process.

| Model ID | Reference | Domain | Sys. Dim. | Par. Dim. | Time Span |
|---|---|---|---|---|---|
| BIOMD0000000003 | Goldbeter1991 | Taxonomy-Amphibia | 3 | 15 | [0.0, 60.0] |
| BIOMD0000000004 | Goldbeter1991 | Taxonomy-Amphibia | 5 | 15 | [0.0, 70.0] |
| BIOMD0000000005 | Tyson1991 | Taxonomy-Opisthokonta | 6 | 10 | [0.0, 100.0] |
| BIOMD0000000006 | Tyson1991 | Taxonomy-Opisthokonta | 2 | 5 | [0.0, 110.0] |
| BIOMD0000000008 | Gardner1998 | Taxonomy-Amphibia | 5 | 23 | [0.0, 40.0] |
| BIOMD0000000009 | Huang1996 | Taxonomy-Xenopus laevis | 26 | 31 | [0.0, 80.0] |
| BIOMD0000000010 | Kholodenko2000 | Taxonomy-Xenopus laevis | 8 | 22 | [0.0, 3500.0] |
| BIOMD0000000011 | Levchenko2000 | Taxonomy-Xenopus laevis | 22 | 30 | [0.0, 290.2] |
| BIOMD0000000012 | Elowitz2000 | Taxonomy-Escherichia coli | 6 | 16 | [0.0, 450.0] |
| BIOMD0000000013 | Poolman2004 | Taxonomy-Nicotiana tabacum | 18 | 77 | [0.0, 0.4] |
| BIOMD0000000014 | Levchenko2000 | Taxonomy-Xenopus laevis | 86 | 300 | [0.0, 241.0] |
| BIOMD0000000015 | Curto1998 | Taxonomy-Homo sapiens | 16 | 111 | [0.0, 20000.0] |
| BIOMD0000000016 | Goldbeter1995 | Taxonomy-Drosophila melanogaster | 6 | 18 | [0.0, 60.0] |
| BIOMD0000000017 | Hoefnagel2002 | Taxonomy-Lactococcus lactis | 11 | 71 | [0.0, 0.4] |
| BIOMD0000000018 | Morrison1989 | Taxonomy-Homo sapiens\|KEGG Pathway-One carbon pool by folate\|KEGG Pathway-Folate biosynthesis | 20 | 107 | [0.0, 500000.0] |
| BIOMD0000000019 | Schoeberl2002 | Taxonomy-Homo sapiens | 93 | 90 | [0.0, 300.0] |
| BIOMD0000000020 | Hodgkin1952 | Taxonomy-Loligo forbesii\|Gene Ontology-giant axon | 4 | 27 | [0.0, 25.0] |
| BIOMD0000000021 | Leloup1999 | Taxonomy-Drosophila melanogaster | 10 | 48 | [0.0, 80.0] |
| BIOMD0000000022 | Ueda2001 | Taxonomy-Drosophila melanogaster | 10 | 66 | [0.0, 70.0] |
| BIOMD0000000023 | Rohwer2001 | Taxonomy-Saccharum officinarum | 5 | 54 | [0.0, 1949.2] |
| BIOMD0000000026 | Markevich2004 | Taxonomy-Xenopus laevis | 11 | 16 | [0.0, 1698.4] |
| BIOMD0000000027 | Markevich2004 | Taxonomy-Xenopus laevis | 5 | 9 | [0.0, 1327.2] |
| BIOMD0000000028 | Markevich2004 | Taxonomy-Xenopus laevis | 16 | 27 | [0.0, 1760.4] |
| BIOMD0000000029 | Markevich2004 | Taxonomy-Xenopus laevis | 6 | 15 | [0.0, 1843.2] |
| BIOMD0000000030 | Markevich2004 | Taxonomy-Xenopus laevis | 18 | 32 | [0.0, 1243.4] |
| BIOMD0000000031 | Markevich2004 | Taxonomy-Xenopus laevis | 6 | 9 | [0.0, 250.0] |
| BIOMD0000000032 | Kofahl2004 | Taxonomy-Saccharomyces cerevisiae\|Gene Ontology-MAPK cascade | 36 | 47 | [0.0, 225.0] |
| BIOMD0000000033 | Brown2004 | Taxonomy-cellular organisms\|Gene Ontology-epidermal growth factor receptor signaling pathway\|Gene Ontology-neurotrophin TRK receptor signaling pathway | 32 | 48 | [0.0, 100.0] |
| BIOMD0000000035 | Vilar2002 | Taxonomy-cellular organisms | 9 | 16 | [0.0, 60.0] |
| BIOMD0000000036 | Tyson1999 | Taxonomy-Drosophila melanogaster | 2 | 11 | [0.0, 60.0] |
| BIOMD0000000037 | Marwan2003 | Taxonomy-Physarum polycephalum | 11 | 12 | [0.0, 95.6] |
| BIOMD0000000038 | Rohwer2000 | Taxonomy-Escherichia coli | 13 | 20 | [0.0, 0.004] |
| BIOMD0000000039 | Marhl2000 | Taxonomy-Xenopus laevis | 5 | 11 | [0.0, 30.0] |
| BIOMD0000000041 | Kongas2007 | Taxonomy-Oryctolagus | 10 | 25 | [0.0, 0.5] |
| BIOMD0000000042 | Nielsen1998 | Taxonomy-Saccharomyces cerevisiae | 15 | 25 | [0.0, 30.0] |
| BIOMD0000000043 | Borghans1997 | Taxonomy-Rattus | 5 | 13 | [0.0, 5.0] |
| BIOMD0000000044 | Borghans1997 | Taxonomy-Rattus | 4 | 17 | [0.0, 7.0] |
| BIOMD0000000045 | Borghans1997 | Taxonomy-Rattus | 4 | 15 | [0.0, 55.0] |
| BIOMD0000000046 | Olsen2003 | Taxonomy-Armoracia rusticana | 12 | 15 | [0.0, 1000.0] |
| BIOMD0000000047 | Oxhamre2005 | Taxonomy-Rattus | 3 | 17 | [0.0, 50.0] |
| BIOMD0000000048 | Kholodenko1999 | Taxonomy-Rattus norvegicus | 23 | 50 | [0.0, 103.2] |

| Model ID | Reference | Domain | Sys. Dim. | Par. Dim. | Time Span |
|---|---|---|---|---|---|
| BIOMD0000000049 | Sasagawa2005 | Taxonomy-Rattus\|Gene Ontology-MAPK cascade\|Gene Ontology-epidermal growth factor receptor signaling pathway\|Gene Ontology-Ras protein signal transduction\|Gene Ontology-neurotrophin TRK receptor signaling pathway | 94 | 234 | [0.0, 1385.2] |
| BIOMD0000000050 | Martins2003 | Taxonomy-cellular organisms | 14 | 16 | [0.0, 351.8] |
| BIOMD0000000051 | Chassagnole2002 | Taxonomy-Escherichia coli\|Gene Ontology-glycolytic process\|Gene Ontology-pentose-phosphate shunt\|Gene Ontology-phosphoenolpyruvate-dependent sugar phosphotransferase system\|KEGG Pathway-Glycolysis / Gluconeogenesis\|KEGG Pathway-Pentose phosphate pathway | 18 | 142 | [0.0, 800.0] |
| BIOMD0000000052 | Brands2002 | Taxonomy-cellular organisms | 11 | 11 | [0.0, 1487.2] |
| BIOMD0000000054 | Ataullahkhanov 1996 | Taxonomy-Homo sapiens\|Gene Ontology-regulation of sodium ion transport\|Gene Ontology-regulation of glycolytic process\|Gene Ontology-AMP biosynthetic process | 3 | 10 | [0.0, 253.6] |
| BIOMD0000000055 | Locke2005 | Taxonomy-Arabidopsis | 13 | 67 | [0.0, 52.0] |
| BIOMD0000000056 | Chen2004 | Taxonomy-Saccharomyces cerevisiae | 50 | 164 | [0.0, 225.0] |
| BIOMD0000000057 | Sneyd2002 | Taxonomy-Rattus rattus | 6 | 33 | [0.0, 4.6] |
| BIOMD0000000058 | Bindschadler 2001 | Taxonomy-Rattus norvegicus\|Gene Ontology-calcium-mediated signaling\|Gene Ontology-inositol phosphate-mediated signaling | 4 | 29 | [0.0, 20.0] |
| BIOMD0000000059 | Fridlyand2003 | Taxonomy-Mus musculus\|Gene Ontology-regulation of insulin secretion\|Gene Ontology-regulation of calcium-mediated signaling | 8 | 97 | [0.0, 220000.0] |
| BIOMD0000000060 | Keizer1996 | Taxonomy-cellular organisms\|Gene Ontology-ryanodine-sensitive calcium-release channel activity\|Gene Ontology-calcium-mediated signaling\|Gene Ontology-calcium-induced calcium release activity | 4 | 11 | [0.0, 8.0] |
| BIOMD0000000061 | Hynne2001 | Taxonomy-Saccharomyces cerevisiae | 22 | 69 | [0.0, 3.0] |
| BIOMD0000000062 | Bhartiya2003 | Taxonomy-Escherichia coli | 3 | 18 | [0.0, 168.0] |
| BIOMD0000000063 | Galazzo1990 | Taxonomy-Saccharomyces cerevisiae\|Gene Ontology-glycolytic process | 5 | 61 | [0.0, 0.8] |
| BIOMD0000000064 | Teusink2000 | Taxonomy-Saccharomyces cerevisiae | 19 | 85 | [0.0, 2.4] |
| BIOMD0000000065 | Yildirim2003 | Taxonomy-Escherichia coli | 8 | 23 | [0.0, 200.4] |
| BIOMD0000000067 | Fung2005 | Taxonomy-Escherichia coli (strain K12) | 7 | 21 | [0.0, 100.0] |
| BIOMD0000000068 | Curien2003 | Taxonomy-Arabidopsis thaliana\|Gene Ontology-threonine biosynthetic process\|Gene Ontology-L-methionine biosynthetic process from O-phospho-L-homoserine and cystathionine\|KEGG Pathway-Glycine, serine and threonine metabolism\|KEGG Pathway-map00271 | 4 | 6 | [0.0, 800.0] |
| BIOMD0000000069 | Fuss2006 | Taxonomy-Homo sapiens | 10 | 20 | [0.0, 100.0] |

*Continued on next page*

| Model ID | Reference | Domain | Sys. Dim. | Par. Dim. | Time Span |
|---|---|---|---|---|---|
| BIOMD0000000070 | Holzhutter2004 | Taxonomy-Homo sapiens\|Gene Ontology-glycolytic process\|Gene Ontology-pentose-phosphate shunt\|Gene Ontology-glutathione metabolic process\|KEGG Pathway-Glycolysis / Gluconeogenesis - Homo sapiens (human)\|KEGG Pathway-Pentose phosphate pathway - Homo sapiens (human) | 40 | 166 | [0.0, 9.6] |
| BIOMD0000000071 | Bakker2001 | Taxonomy-Trypanosoma brucei | 23 | 72 | [0.0, 0.8] |
| BIOMD0000000072 | Yi2003 | Taxonomy-Saccharomyces cerevisiae | 7 | 8 | [0.0, 1500.0] |
| BIOMD0000000073 | Leloup2003 | Taxonomy-Mammalia | 16 | 80 | [0.0, 52.0] |
| BIOMD0000000074 | Leloup2003 | Taxonomy-Mammalia | 19 | 95 | [0.0, 70.0] |
| BIOMD0000000075 | Xu2003 | Taxonomy-Mus musculus | 10 | 30 | [0.0, 87.8] |
| BIOMD0000000076 | Cronwright2002 | Taxonomy-Saccharomyces cerevisiae | 1 | 18 | [0.0, 1.6] |
| BIOMD0000000077 | Blum2000 | Taxonomy-cellular organisms | 8 | 10 | [0.0, 12.0] |
| BIOMD0000000078 | Leloup2003 | Taxonomy-Mammalia | 16 | 81 | [0.0, 60.0] |
| BIOMD0000000079 | Goldbeter2006 | Taxonomy-Homo sapiens | 3 | 11 | [0.0, 23.0] |
| BIOMD0000000081 | Suh2004 | Taxonomy-Homo sapiens\|Gene Ontology-lipid metabolic process\|Gene Ontology-phospholipase C-activating G-protein coupled acetylcholine receptor signaling pathway | 21 | 35 | [0.0, 12.0] |
| BIOMD0000000083 | Leloup2003 | Taxonomy-Mammalia | 19 | 64 | [0.0, 100.0] |
| BIOMD0000000084 | Hornberg2005 | Taxonomy-Mammalia\|KEGG Pathway-MAPK signaling pathway - Rattus norvegicus (rat) | 8 | 18 | [0.0, 34.0] |
| BIOMD0000000089 | Locke2006 | Taxonomy-Arabidopsis thaliana | 16 | 78 | [0.0, 55.0] |
| BIOMD0000000090 | Wolf2001 | Taxonomy-Saccharomyces cerevisiae | 13 | 28 | [0.0, 85.0] |
| BIOMD0000000091 | Proctor2005 | Taxonomy-cellular organisms | 15 | 21 | [0.0, 150.0] |
| BIOMD0000000093 | Yamada2003 | Taxonomy-Mus musculus | 33 | 72 | [0.0, 1000.0] |
| BIOMD0000000094 | Yamada2003 | Taxonomy-Mus musculus | 33 | 70 | [0.0, 4500.0] |
| BIOMD0000000096 | Zeilinger2006 | Taxonomy-Arabidopsis | 19 | 94 | [0.0, 80.0] |
| BIOMD0000000097 | Zeilinger2006 | Taxonomy-Arabidopsis | 19 | 93 | [0.0, 75.0] |
| BIOMD0000000098 | Goldbeter1990 | Taxonomy-cellular organisms | 2 | 13 | [0.0, 5.0] |
| BIOMD0000000099 | Laub1998 | Taxonomy-Dictyostelium | 7 | 14 | [0.0, 14.5] |
| BIOMD0000000100 | Rozi2003 | Taxonomy-cellular organisms\|Gene Ontology-phosphorylase kinase regulator activity\|KEGG Pathway-Calcium signaling pathway - Homo sapiens (human) | 4 | 30 | [0.0, 3.0] |
| BIOMD0000000101 | Vilar2006 | Taxonomy-cellular organisms\|Reactome-Signaling by TGF-beta Receptor Complex\|KEGG Pathway-TGF-beta signaling pathway - Homo sapiens (human) | 6 | 9 | [0.0, 1935.6] |
| BIOMD0000000102 | Legewie2006 | Taxonomy-cellular organisms | 13 | 41 | [0.0, 2000.0] |
| BIOMD0000000103 | Legewie2006 | Taxonomy-cellular organisms | 17 | 49 | [0.0, 2000.0] |
| BIOMD0000000104 | Klipp2002 | Taxonomy-cellular organisms | 6 | 2 | [0.0, 10.4] |
| BIOMD0000000105 | Proctor2007 | Taxonomy-Eukaryota | 35 | 17 | [0.0, 300000.0] |
| BIOMD0000000106 | Yang2007 | Taxonomy-Homo sapiens | 25 | 54 | [0.0, 49.6] |
| BIOMD0000000107 | Novak1993 | Taxonomy-Amphibia | 9 | 36 | [0.0, 200.0] |
| BIOMD0000000109 | Haberichter2007 | Taxonomy-Mammalia | 57 | 61 | [0.0, 1981.6] |
| BIOMD0000000110 | Qu2003 | Taxonomy-Eukaryota | 15 | 30 | [0.0, 150.0] |
| BIOMD0000000111 | Novak2001 | Taxonomy-Schizosaccharomycetaceae | 10 | 53 | [0.0, 300.0] |
| BIOMD0000000112 | Clarke2006 | Taxonomy-Neovison vison | 10 | 17 | [0.0, 600.0] |
| BIOMD0000000113 | Dupont1992 | Taxonomy-cellular organisms | 4 | 19 | [0.0, 2.5] |
| BIOMD0000000114 | Somogyi1990 | Taxonomy-Rattus | 2 | 8 | [0.0, 200.0] |
| BIOMD0000000115 | Somogyi1990 | Taxonomy-Rattus | 2 | 8 | [0.0, 15.0] |

| Model ID | Reference | Domain | Sys. Dim. | Par. Dim. | Time Span |
|---|---|---|---|---|---|
| BIOMD0000000117 | Dupont1991 | Taxonomy-cellular organisms | 2 | 15 | [0.0, 40.0] |
| BIOMD0000000118 | Golomb2006 | Taxonomy-Rattus | 5 | 45 | [0.0, 420.0] |
| BIOMD0000000119 | Golomb2006 | Taxonomy-Rattus | 9 | 69 | [0.0, 230.0] |
| BIOMD0000000121 | Clancy2001 | Taxonomy-Cavia porcellus | 6 | 19 | [0.0, 1500.0] |
| BIOMD0000000122 | Fisher2006 | Taxonomy-Mammalia | 12 | 23 | [0.0, 300.0] |
| BIOMD0000000123 | Fisher2006 | Taxonomy-Mammalia | 14 | 22 | [0.0, 679.2] |
| BIOMD0000000124 | Wu2006 | Taxonomy-Rattus | 7 | 57 | [0.0, 800.0] |
| BIOMD0000000125 | Komarova2005 | Taxonomy-Opisthokonta | 5 | 7 | [0.0, 10.0] |
| BIOMD0000000126 | Clancy2002 | Taxonomy-Chordata | 9 | 24 | [0.0, 300.0] |
| BIOMD0000000128 | Bertram2006 | Taxonomy-Rattus | 7 | 50 | [0.0, 500.0] |
| BIOMD0000000131 | Izhikevich2004 | Taxonomy-Mammalia | 2 | 8 | [0.0, 200.0] |
| BIOMD0000000134 | Izhikevich2004 | Taxonomy-Mammalia | 2 | 8 | [0.0, 180.0] |
| BIOMD0000000135 | Izhikevich2004 | Taxonomy-Mammalia | 2 | 8 | [0.0, 200.0] |
| BIOMD0000000137 | Sedaghat2002 | Taxonomy-Homo sapiens | 21 | 40 | [0.0, 70.0] |
| BIOMD0000000138 | Tabak2007 | Taxonomy-Chordata | 4 | 40 | [0.0, 500.0] |
| BIOMD0000000143 | Olsen2003B | Taxonomy-Mammalia | 20 | 24 | [0.0, 52.0] |
| BIOMD0000000144 | Calzone2007 | Taxonomy-Drosophila | 17 | 46 | [0.0, 300.0] |
| BIOMD0000000145 | Wang2007 | Taxonomy-cellular organisms\|Gene Ontology-positive regulation of cytosolic calcium ion concentration involved in phospholipase C-activating G-protein coupled signaling pathway\|KEGG Pathway-Calcium signaling pathway - Homo sapiens (human) | 7 | 32 | [0.0, 600.0] |
| BIOMD0000000146 | Hatakeyama2003 | Taxonomy-Cricetinae\|Gene Ontology-regulation of MAP kinase activity\|Gene Ontology-regulation of protein kinase B signaling | 36 | 75 | [0.0, 15000.0] |
| BIOMD0000000147 | ODea2007 | Taxonomy-Mus musculus | 24 | 73 | [0.0, 445.8] |
| BIOMD0000000148 | Komarova2003 | Taxonomy-Chordata | 7 | 13 | [0.0, 120.0] |
| BIOMD0000000149 | Kim2007 | Taxonomy-Homo sapiens\|Gene Ontology-MAPK cascade\|Gene Ontology-canonical Wnt signaling pathway\|KEGG Pathway-MAPK signaling pathway - Homo sapiens (human)\|KEGG Pathway-Wnt signaling pathway - Homo sapiens (human) | 28 | 59 | [0.0, 2100.0] |
| BIOMD0000000151 | Singh2006 | Taxonomy-Eukaryota\|Gene Ontology-MAPK cascade\|Gene Ontology-JAK-STAT cascade\|KEGG Pathway-MAPK signaling pathway - Mus musculus (mouse)\|KEGG Pathway-Jak-STAT signaling pathway - Mus musculus (mouse) | 66 | 105 | [0.0, 2000.0] |
| BIOMD0000000156 | Zatorsky2006 | Taxonomy-Homo sapiens | 3 | 7 | [0.0, 15.0] |
| BIOMD0000000157 | Zatorsky2006 | Taxonomy-Homo sapiens | 3 | 8 | [0.0, 17.0] |
| BIOMD0000000158 | Zatorsky2006 | Taxonomy-Homo sapiens | 3 | 14 | [0.0, 13.0] |
| BIOMD0000000159 | Zatorsky2006 | Taxonomy-Homo sapiens | 3 | 7 | [0.0, 50.0] |
| BIOMD0000000160 | Xie2007 | Taxonomy-Drosophila melanogaster | 24 | 47 | [0.0, 52.0] |
| BIOMD0000000161 | Eungdamrong 2007 | Taxonomy-cellular organisms\|Gene Ontology-Ras protein signal transduction | 43 | 131 | [0.0, 250.0] |
| BIOMD0000000162 | Hernjak2005 | Taxonomy-Chordata | 24 | 202 | [0.0, 35.0] |
| BIOMD0000000163 | Zi2007 | Taxonomy-cellular organisms | 16 | 20 | [0.0, 494.0] |
| BIOMD0000000164 | SmithAE2002 | Taxonomy-Mammalia | 26 | 64 | [0.0, 20.4] |
| BIOMD0000000165 | Saucerman2006 | Taxonomy-Rattus norvegicus | 36 | 62 | [0.0, 192.2] |
| BIOMD0000000166 | Zhu2007 | Taxonomy-cellular organisms | 3 | 22 | [0.0, 1.25] |

| Model ID | Reference | Domain | Sys. Dim. | Par. Dim. | Time Span |
|---|---|---|---|---|---|
| BIOMD0000000167 | Mayya2005 | Taxonomy-cellular organisms\|KEGG Pathway-Jak-STAT signaling pathway - Homo sapiens (human) | 8 | 14 | [0.0, 5000.0] |
| BIOMD0000000168 | Obeyesekere 1999 | Taxonomy-Eukaryota | 7 | 21 | [0.0, 60.0] |
| BIOMD0000000169 | Aguda1999 | Taxonomy-Mammalia | 11 | 32 | [0.0, 1000.0] |
| BIOMD0000000170 | Weimann2004 | Taxonomy-Mammalia | 7 | 27 | [0.0, 52.0] |
| BIOMD0000000171 | Leloup1998 | Taxonomy-Drosophila melanogaster | 10 | 41 | [0.0, 50.0] |
| BIOMD0000000172 | Pritchard2002 | Taxonomy-Saccharomyces cerevisiae | 17 | 87 | [0.0, 1.2] |
| BIOMD0000000174 | Zerial2008 | Taxonomy-Homo sapiens | 4 | 18 | [0.0, 1000.0] |
| BIOMD0000000175 | Birtwistle2007 | Taxonomy-Homo sapiens | 118 | 240 | [0.0, 800.0] |
| BIOMD0000000176 | Conant2007 | Taxonomy-Saccharomyces cerevisiae\| Gene Ontology-glycolytic process\|KEGG Pathway-Glycolysis / Gluconeogenesis - Saccharomyces cerevisiae (budding yeast) | 17 | 98 | [0.0, 5.2] |
| BIOMD0000000177 | Conant2007 | Taxonomy-Saccharomyces cerevisiae\| Gene Ontology-glycolytic process\|KEGG Pathway-Glycolysis / Gluconeogenesis - Saccharomyces cerevisiae (budding yeast) | 18 | 97 | [0.0, 6.2] |
| BIOMD0000000178 | Lebeda2008 | Taxonomy-Mus musculus\|Taxonomy-Rattus norvegicus\|Taxonomy-Homo sapiens | 5 | 5 | [0.0, 30000.0] |
| BIOMD0000000179 | Kim2007 | Taxonomy-cellular organisms | 7 | 18 | [0.0, 30.0] |
| BIOMD0000000180 | Kim2007 | Taxonomy-cellular organisms | 8 | 20 | [0.0, 80.0] |
| BIOMD0000000181 | Sriram2007 | Taxonomy-Opisthokonta | 6 | 25 | [0.0, 225.0] |
| BIOMD0000000182 | Neves2008 | Taxonomy-Rattus norvegicus\|Gene Ontology-MAPK cascade\|Gene Ontology-activation of adenylate cyclase activity | 37 | 83 | [0.0, 1643.6] |
| BIOMD0000000184 | Lavrentovich 2008 | Taxonomy-Mammalia\|Gene Ontology-calcium-mediated signaling | 3 | 14 | [0.0, 400.0] |
| BIOMD0000000185 | Locke2008 | Taxonomy-Mammalia | 8 | 19 | [0.0, 55.0] |
| BIOMD0000000188 | Proctor2008 | Taxonomy-Homo sapiens | 18 | 21 | [0.0, 40000.0] |
| BIOMD0000000189 | Proctor2008 | Taxonomy-Homo sapiens | 16 | 16 | [0.0, 50000.0] |
| BIOMD0000000190 | Caso2006 | Taxonomy-Mammalia\|Taxonomy-Rodentia | 11 | 56 | [0.0, 4000.0] |
| BIOMD0000000191 | ez2008 | Taxonomy-Mammalia | 2 | 18 | [0.0, 120.0] |
| BIOMD0000000192 | rlich2003 | Taxonomy-Homo sapiens | 11 | 16 | [0.0, 60.0] |
| BIOMD0000000195 | Tyson2001 | Taxonomy-Saccharomyces cerevisiae | 11 | 37 | [0.0, 330.0] |
| BIOMD0000000197 | Bartholome2007 | Taxonomy-Mammalia | 5 | 12 | [0.0, 5000.0] |
| BIOMD0000000198 | Stone1996 | Taxonomy-Bos taurus | 11 | 16 | [0.0, 1.2] |
| BIOMD0000000199 | Santolini2001 | Taxonomy-Rattus norvegicus | 9 | 10 | [0.0, 8.0] |
| BIOMD0000000201 | Goldbeter2008 | Taxonomy-Amniota | 22 | 71 | [0.0, 225.0] |
| BIOMD0000000202 | ChenXF2008 | Taxonomy-cellular organisms | 10 | 38 | [0.0, 60.0] |
| BIOMD0000000203 | Chickarmane 2006 | Taxonomy-Mus musculus\|Taxonomy-Homo sapiens | 5 | 32 | [0.0, 5.2] |
| BIOMD0000000204 | Chickarmane 2006 | Taxonomy-Mus musculus\|Taxonomy-Homo sapiens | 5 | 32 | [0.0, 5.2] |
| BIOMD0000000205 | Ung2008 | Taxonomy-Eukaryota | 194 | 313 | [0.0, 5000.0] |
| BIOMD0000000206 | Wolf2000 | Taxonomy-Saccharomyces cerevisiae | 9 | 18 | [0.0, 0.35] |
| BIOMD0000000207 | Romond1999 | Taxonomy-Eukaryota | 6 | 32 | [0.0, 2500.0] |
| BIOMD0000000208 | Deineko2003 | Taxonomy-Mammalia | 6 | 18 | [0.0, 1000.0] |
| BIOMD0000000209 | Chickarmane 2008 | Taxonomy-Homo sapiens | 8 | 49 | [0.0, 94.0] |
| BIOMD0000000210 | Chickarmane 2008 | Taxonomy-Homo sapiens | 8 | 46 | [0.0, 1329.2] |
| BIOMD0000000211 | Albert2005 | Taxonomy-Trypanosoma brucei | 24 | 81 | [0.0, 1.2] |

*Continued on next page*

| Model ID | Reference | Domain | Sys. Dim. | Par. Dim. | Time Span |
|---|---|---|---|---|---|
| BIOMD0000000212 | Curien2009 | Taxonomy-Arabidopsis thaliana | 8 | 64 | [0.0, 1933.2] |
| BIOMD0000000213 | Nijhout2004 | Taxonomy-Mammalia | 6 | 38 | [0.0, 0.3] |
| BIOMD0000000214 | Akman2008 | Taxonomy-Neurospora crassa | 16 | 38 | [0.0, 55.0] |
| BIOMD0000000216 | Hong2009 | Taxonomy-Mammalia | 5 | 18 | [0.0, 55.0] |
| BIOMD0000000217 | Bruggeman2005 | Taxonomy-Escherichia coli | 12 | 98 | [0.0, 4.4] |
| BIOMD0000000218 | Singh2006 | Taxonomy-Mycobacterium tuberculosis | 9 | 66 | [0.0, 150.0] |
| BIOMD0000000219 | Singh2006 | Taxonomy-Mycobacterium tuberculosis | 9 | 70 | [0.0, 197.6] |
| BIOMD0000000220 | Albeck2008 | Taxonomy-Homo sapiens | 58 | 77 | [0.0, 1000000.0] |
| BIOMD0000000221 | Singh2006 | Taxonomy-Escherichia coli | 8 | 49 | [0.0, 2.5] |
| BIOMD0000000222 | Singh2006 | Taxonomy-Escherichia coli | 8 | 49 | [0.0, 48.6] |
| BIOMD0000000223 | Borisov2009 | Taxonomy-Homo sapiens | 86 | 162 | [0.0, 972.8] |
| BIOMD0000000224 | Meyer1991 | Taxonomy-Rattus rattus | 4 | 11 | [0.0, 90.0] |
| BIOMD0000000225 | Westermark2003 | Taxonomy-Mammalia | 2 | 19 | [0.0, 500.0] |
| BIOMD0000000226 | Radulescu2008 | Taxonomy-Mus musculus | 14 | 31 | [0.0, 18000.0] |
| BIOMD0000000227 | Radulescu2008 | Taxonomy-Mus musculus | 45 | 88 | [0.0, 200000.0] |
| BIOMD0000000228 | Swat2004 | Taxonomy-Mammalia | 9 | 40 | [0.0, 1970.0] |
| BIOMD0000000229 | Ma2002 | Taxonomy-Dictyostelium discoideum\|KEGG Pathway-Bacterial chemotaxis - Xanthomonas campestris pv. campestris ATCC 33913 | 7 | 14 | [0.0, 17.0] |
| BIOMD0000000230 | Ihekwaba2004 | Taxonomy-Mus musculus | 24 | 64 | [0.0, 16000.0] |
| BIOMD0000000231 | Valero2006 | Taxonomy-Bos taurus | 5 | 9 | [0.0, 200.4] |
| BIOMD0000000233 | Wilhelm2009 | Taxonomy-cellular organisms | 2 | 4 | [0.0, 10.0] |
| BIOMD0000000236 | Westermark2003 | Taxonomy-Mammalia | 3 | 26 | [0.0, 420.0] |
| BIOMD0000000237 | Schaber2006 | Taxonomy-Saccharomyces cerevisiae | 24 | 46 | [0.0, 600.0] |
| BIOMD0000000238 | Overgaard2007 | Taxonomy-Macaca fascicularis | 3 | 56 | [0.0, 350.0] |
| BIOMD0000000239 | Jiang2007 | Taxonomy-Homo sapiens | 59 | 306 | [0.0, 464.4] |
| BIOMD0000000240 | Veening2008 | Taxonomy-Octodon degus\|Taxonomy-Bacillus subtilis | 6 | 23 | [0.0, 50000.0] |
| BIOMD0000000241 | Shi1993 | Taxonomy-Homo sapiens | 5 | 19 | [0.0, 30.0] |
| BIOMD0000000242 | Bai2003 | Taxonomy-Murinae | 6 | 26 | [0.0, 75.0] |
| BIOMD0000000243 | Neumann2010 | Taxonomy-Homo sapiens | 23 | 17 | [0.0, 800.0] |
| BIOMD0000000244 | Kotte2010 | Taxonomy-Escherichia coli | 47 | 213 | [0.0, 16000.0] |
| BIOMD0000000247 | Ralser2007 | Taxonomy-Saccharomyces cerevisiae | 24 | 135 | [0.0, 15.8] |
| BIOMD0000000249 | Restif2006 | Taxonomy-Bordetella pertussis\|Taxonomy-Homo sapiens | 9 | 21 | [0.0, 20.0] |
| BIOMD0000000250 | Nakakuki2010 | Taxonomy-Homo sapiens | 47 | 141 | [0.0, 1250.0] |
| BIOMD0000000253 | Teusink1998 | Taxonomy-Saccharomyces cerevisiae | 3 | 26 | [0.0, 7.0] |
| BIOMD0000000254 | Bier2000 | Taxonomy-Saccharomyces cerevisiae | 4 | 7 | [0.0, 140.0] |
| BIOMD0000000255 | Chen2009 | Taxonomy-Homo sapiens | 500 | 239 | [0.0, 10000.0] |
| BIOMD0000000256 | Rehm2006 | Taxonomy-Homo sapiens\|Brenda Tissue Ontology-HeLa cell | 27 | 99 | [0.0, 28.8] |
| BIOMD0000000257 | Piedrafita2010 | Taxonomy-cellular organisms | 8 | 19 | [0.0, 400.0] |
| BIOMD0000000258 | Ortega2006 | Taxonomy-cellular organisms | 3 | 9 | [0.0, 21.6] |
| BIOMD0000000259 | Tiago2010 | Taxonomy-Mus musculus | 17 | 29 | [0.0, 400.0] |
| BIOMD0000000260 | Tiago2010 | Taxonomy-Mus musculus | 17 | 29 | [0.0, 600.0] |
| BIOMD0000000261 | Tiago2010 | Taxonomy-Mus musculus | 17 | 29 | [0.0, 600.0] |
| BIOMD0000000262 | Fujita2010 | Taxonomy-Rattus norvegicus | 9 | 24 | [0.0, 1300.0] |
| BIOMD0000000263 | Fujita2010 | Taxonomy-Rattus norvegicus | 9 | 24 | [0.0, 1962.0] |
| BIOMD0000000264 | Fujita2010 | Taxonomy-Rattus norvegicus | 11 | 26 | [0.0, 738.4] |
| BIOMD0000000265 | Conradie2010 | Taxonomy-Mammalia | 23 | 88 | [0.0, 30.0] |
| BIOMD0000000267 | Lebeda2008 | Taxonomy-Mus musculus\|Taxonomy-Rattus norvegicus\|Taxonomy-Homo sapiens | 4 | 4 | [0.0, 398.8] |

| Model ID | Reference | Domain | Sys. Dim. | Par. Dim. | Time Span |
|---|---|---|---|---|---|
| BIOMD0000000268 | Reed2008 | Taxonomy-Homo sapiens\|Brenda Tissue Ontology-hepatocyte | 34 | 159 | [0.0, 60.0] |
| BIOMD0000000269 | Liu2010 | Taxonomy-Arabidopsis thaliana | 9 | 32 | [0.0, 100.0] |
| BIOMD0000000270 | Schilling2009 | Taxonomy-Mus musculus | 33 | 39 | [0.0, 80.0] |
| BIOMD0000000271 | Becker2010 | Taxonomy-Murinae\|Brenda Tissue Ontology-hematopoietic cell line | 6 | 10 | [0.0, 600.0] |
| BIOMD0000000272 | Becker2010 | Taxonomy-Murinae\|Brenda Tissue Ontology-hematopoietic cell line | 6 | 9 | [0.0, 50000.0] |
| BIOMD0000000273 | Pokhilko2010 | Taxonomy-Arabidopsis thaliana | 19 | 99 | [0.0, 55.0] |
| BIOMD0000000274 | Rattanakul2003 | Taxonomy-Chordata | 3 | 13 | [0.0, 8000.0] |
| BIOMD0000000275 | Goldbeter2007 | Taxonomy-Vertebrata | 5 | 24 | [0.0, 43.2] |
| BIOMD0000000276 | Shrestha2010 | Taxonomy-Homo sapiens | 3 | 21 | [0.0, 700.0] |
| BIOMD0000000277 | Shrestha2010 | Taxonomy-Homo sapiens | 3 | 21 | [0.0, 800.0] |
| BIOMD0000000281 | Chance1960 | Taxonomy-Homo sapiens\|Brenda Tissue Ontology-ascites tumor cell | 32 | 54 | [0.0, 65.6] |
| BIOMD0000000282 | Chance1952 | Taxonomy-Equus caballus | 5 | 4 | [0.0, 0.4] |
| BIOMD0000000283 | Chance1943 | Taxonomy-Armoracia rusticana | 4 | 2 | [0.0, 32.6] |
| BIOMD0000000284 | Hofmeyer1986 | Taxonomy-Bacteria | 6 | 0 | [0.0, 48.2] |
| BIOMD0000000286 | Proctor2010 | Taxonomy-Mammalia | 54 | 58 | [0.0, 75000.0] |
| BIOMD0000000287 | Passos2010 | Taxonomy-Homo sapiens | 21 | 36 | [0.0, 300000.0] |
| BIOMD0000000288 | Wang2009 | Taxonomy-Homo sapiens | 19 | 45 | [0.0, 136.4] |
| BIOMD0000000289 | Alexander2010 | Taxonomy-Metazoa | 4 | 15 | [0.0, 150.0] |
| BIOMD0000000290 | Alexander2010 | Taxonomy-Metazoa | 4 | 16 | [0.0, 200.0] |
| BIOMD0000000291 | Nikolaev2005 | Taxonomy-Eukaryota | 10 | 13 | [0.0, 245.0] |
| BIOMD0000000292 | Rovers1995 | Taxonomy-Viridiplantae | 3 | 6 | [0.0, 500.0] |
| BIOMD0000000293 | Proctor2010 | Taxonomy-Homo sapiens | 131 | 92 | [0.0, 877.6] |
| BIOMD0000000294 | Restif2007 | Taxonomy-Bordetella pertussis\|Taxonomy-Homo sapiens | 11 | 22 | [0.0, 100.0] |
| BIOMD0000000295 | Akman2008 | Taxonomy-Neurospora crassa | 5 | 17 | [0.0, 50.0] |
| BIOMD0000000296 | Balagaddé2008 | Taxonomy-Escherichia coli | 5 | 12 | [0.0, 350.0] |
| BIOMD0000000297 | Ciliberto2003 | Taxonomy-Schizosaccharomyces pombe | 19 | 81 | [0.0, 290.0] |
| BIOMD0000000298 | Leloup1999 | Taxonomy-Drosophila melanogaster | 10 | 39 | [0.0, 55.0] |
| BIOMD0000000299 | Leloup1999 | Taxonomy-Neurospora crassa | 3 | 11 | [0.0, 50.0] |
| BIOMD0000000300 | Schmierer2010 | Taxonomy-Homo sapiens | 3 | 16 | [0.0, 7.6] |
| BIOMD0000000301 | Friedland2009 | Taxonomy-Escherichia coli (strain K12) | 8 | 36 | [0.0, 980.0] |
| BIOMD0000000303 | Liu2011 | Taxonomy-Homo sapiens | 42 | 90 | [0.0, 50000.0] |
| BIOMD0000000304 | Plant1981 | Taxonomy-Aplysia\|Brenda Tissue Ontology-abdominal ganglion\|Brenda Tissue Ontology-neuron | 5 | 34 | [0.0, 23000.0] |
| BIOMD0000000305 | Kolomeisky2003 | Taxonomy-Eukaryota | 6 | 15 | [0.0, 0.3] |
| BIOMD0000000309 | Tyson2003 | Taxonomy-cellular organisms | 1 | 9 | [0.0, 2000.0] |
| BIOMD0000000312 | Tyson2003 | Taxonomy-cellular organisms | 2 | 5 | [0.0, 40.0] |
| BIOMD0000000313 | Raia2010 | Taxonomy-Homo sapiens\|Reactome-Signaling by Interleukins\|KEGG Pathway-Jak-STAT signaling pathway - Homo sapiens (human) | 15 | 18 | [0.0, 50000.0] |
| BIOMD0000000314 | Raia2011 | Taxonomy-Homo sapiens\|Reactome-Signaling by Interleukins\|KEGG Pathway-Jak-STAT signaling pathway - Homo sapiens (human) | 11 | 12 | [0.0, 50000.0] |
| BIOMD0000000315 | Montagne2011 | Taxonomy-cellular organisms | 19 | 47 | [0.0, 250.0] |
| BIOMD0000000318 | Yao2008 | Taxonomy-Mammalia | 7 | 31 | [0.0, 20.0] |
| BIOMD0000000319 | Decroly1982 | Taxonomy-cellular organisms | 3 | 7 | [0.0, 220.0] |
| BIOMD0000000320 | Grange2001 | Taxonomy-Rattus norvegicus | 9 | 31 | [0.0, 18.4] |
| BIOMD0000000321 | Grange2001 | Taxonomy-Rattus norvegicus | 3 | 18 | [0.0, 12.0] |

| Model ID | Reference | Domain | Sys. Dim. | Par. Dim. | Time Span |
|----------|-----------|--------|-----------|-----------|-----------|
| BIOMD0000000322 | Kim2011 | Taxonomy-cellular organisms | 4 | 14 | [0.0, 160.0] |
| BIOMD0000000323 | Kim2011 | Taxonomy-cellular organisms | 3 | 6 | [0.0, 25.0] |
| BIOMD0000000325 | Palini2011 | Taxonomy-Saccharomyces cerevisiae | 5 | 15 | [0.0, 3000.0] |
| BIOMD0000000326 | DellOrco2009 | Taxonomy-Mus musculus | 71 | 96 | [0.0, 0.4] |
| BIOMD0000000327 | Whitcomb2004 | Taxonomy-Homo sapiens\|Brenda Tissue Ontology-pancreatic duct | 4 | 44 | [0.0, 998.4] |
| BIOMD0000000328 | Bucher2011 | Taxonomy-Homo sapiens | 18 | 30 | [0.0, 2000.0] |
| BIOMD0000000329 | Kummer2000 | Taxonomy-Rattus norvegicus | 3 | 12 | [0.0, 35.0] |
| BIOMD0000000330 | Larsen2004 | Taxonomy-Rattus norvegicus | 5 | 21 | [0.0, 50.0] |
| BIOMD0000000331 | Larsen2004 | Taxonomy-Rattus norvegicus | 7 | 27 | [0.0, 31.0] |
| BIOMD0000000332 | Bungay2006 | Taxonomy-Homo sapiens | 78 | 113 | [0.0, 423.8] |
| BIOMD0000000333 | Bungay2006 | Taxonomy-Homo sapiens | 54 | 75 | [0.0, 747.2] |
| BIOMD0000000334 | Bungay2003 | Taxonomy-Homo sapiens | 74 | 110 | [0.0, 268.8] |
| BIOMD0000000337 | Pfeiffer2001 | Taxonomy-cellular organisms | 3 | 2 | [0.0, 36.4] |
| BIOMD0000000342 | Zi2011 | Taxonomy-Homo sapiens | 19 | 34 | [0.0, 8.0] |
| BIOMD0000000344 | Proctor2011 | Taxonomy-Chordata | 50 | 65 | [0.0, 759.8] |
| BIOMD0000000346 | FitzHugh1961 | Taxonomy-cellular organisms\|Brenda Tissue Ontology-nerve | 2 | 4 | [0.0, 25.0] |
| BIOMD0000000347 | Bachmann2011 | Taxonomy-Mus musculus\|Brenda Tissue Ontology-erythroid progenitor cell | 26 | 32 | [0.0, 519.0] |
| BIOMD0000000348 | Fridlyand2010 | Taxonomy-Homo sapiens | 7 | 97 | [0.0, 1939.6] |
| BIOMD0000000350 | Troein2011 | Taxonomy-Ostreococcus tauri | 14 | 29 | [0.0, 55.0] |
| BIOMD0000000351 | Vernoux2011 | Taxonomy-Arabidopsis | 6 | 23 | [0.0, 1448.6] |
| BIOMD0000000352 | Vernoux2011 | Taxonomy-Arabidopsis | 6 | 23 | [0.0, 1750.0] |
| BIOMD0000000354 | Abell2011 | Taxonomy-Drosophila | 6 | 20 | [0.0, 88.0] |
| BIOMD0000000355 | Abell2011 | Taxonomy-Drosophila | 9 | 22 | [0.0, 90.0] |
| BIOMD0000000357 | Lee2010 | Taxonomy-Eukaryota | 9 | 12 | [0.0, 582.0] |
| BIOMD0000000358 | Stortelder1997 | Taxonomy-Eukaryota | 12 | 14 | [0.0, 10.4] |
| BIOMD0000000359 | Panteleev2002 | Taxonomy-Eukaryota | 9 | 15 | [0.0, 1455.2] |
| BIOMD0000000360 | Panteleev2002B | Taxonomy-Eukaryota | 9 | 15 | [0.0, 1497.6] |
| BIOMD0000000361 | Panteleev2002 | Taxonomy-Eukaryota | 8 | 9 | [0.0, 20.0] |
| BIOMD0000000363 | Lee2010 | Taxonomy-Eukaryota | 4 | 4 | [0.0, 1749.4] |
| BIOMD0000000364 | Lee2010 | Taxonomy-Homo sapiens | 14 | 22 | [0.0, 5000.0] |
| BIOMD0000000366 | Orfao2008 | Taxonomy-Eukaryota | 12 | 14 | [0.0, 23.8] |
| BIOMD0000000368 | Beltrami1995 | Taxonomy-Eukaryota | 8 | 10 | [0.0, 29.4] |
| BIOMD0000000369 | Beltrami1995 | Taxonomy-Eukaryota | 7 | 10 | [0.0, 50.0] |
| BIOMD0000000370 | Vinod2011 | Taxonomy-Saccharomyces cerevisiae | 32 | 105 | [0.0, 150.0] |
| BIOMD0000000371 | DeVries2000 | Taxonomy-Homo sapiens | 3 | 24 | [0.0, 4000.0] |
| BIOMD0000000372 | Tolic2000 | Taxonomy-Homo sapiens | 6 | 29 | [0.0, 250.0] |
| BIOMD0000000373 | Bertram2004 | Taxonomy-Homo sapiens | 7 | 113 | [0.0, 50000.0] |
| BIOMD0000000374 | Bertram1995 | Taxonomy-Homo sapiens | 5 | 50 | [0.0, 30000.0] |
| BIOMD0000000375 | Mears1997 | Taxonomy-Homo sapiens | 5 | 61 | [0.0, 31000.0] |
| BIOMD0000000376 | Bertram2007 | Taxonomy-Homo sapiens | 11 | 151 | [0.0, 5000.0] |
| BIOMD0000000377 | Bertram2000 | Taxonomy-Homo sapiens | 4 | 32 | [0.0, 5500.0] |
| BIOMD0000000378 | Chay1997 | Taxonomy-Homo sapiens | 6 | 49 | [0.0, 40.0] |
| BIOMD0000000379 | DallaMan2007 | Taxonomy-Homo sapiens | 12 | 57 | [0.0, 944.2] |
| BIOMD0000000382 | Sturis1991 | Taxonomy-Homo sapiens | 6 | 13 | [0.0, 250.0] |
| BIOMD0000000383 | Arnold2011 | Taxonomy-Viridiplantae | 2 | 11 | [0.0, 10.2] |
| BIOMD0000000384 | Arnold2011 | Taxonomy-Viridiplantae | 2 | 14 | [0.0, 30.0] |
| BIOMD0000000385 | Arnold2011 | Taxonomy-Viridiplantae | 2 | 20 | [0.0, 2.0] |
| BIOMD0000000386 | Arnold2011 | Taxonomy-Viridiplantae | 2 | 15 | [0.0, 1176.8] |
| BIOMD0000000387 | Arnold2011 | Taxonomy-Viridiplantae | 2 | 17 | [0.0, 24.6] |
| BIOMD0000000388 | Arnold2011 | Taxonomy-Viridiplantae | 5 | 16 | [0.0, 1226.4] |
| BIOMD0000000389 | Arnold2011 | Taxonomy-Viridiplantae | 19 | 19 | [0.0, 800.0] |

| Model ID | Reference | Domain | Sys. Dim. | Par. Dim. | Time Span |
|---|---|---|---|---|---|
| BIOMD0000000390 | Arnold2011 | Taxonomy-Viridiplantae | 7 | 19 | [0.0, 34.4] |
| BIOMD0000000391 | Arnold2011 | Taxonomy-Viridiplantae | 16 | 75 | [0.0, 0.3] |
| BIOMD0000000392 | Arnold2011 | Taxonomy-Viridiplantae\|Taxonomy-Mammalia | 23 | 135 | [0.0, 31.2] |
| BIOMD0000000393 | Arnold2011 | Taxonomy-Viridiplantae | 28 | 181 | [0.0, 350.0] |
| BIOMD0000000394 | Sivakumar2011 | Taxonomy-Mammalia | 23 | 38 | [0.0, 16.0] |
| BIOMD0000000395 | Sivakumar2011 | Taxonomy-Mammalia | 23 | 25 | [0.0, 81.6] |
| BIOMD0000000396 | Sivakumar2011 | Taxonomy-Mammalia | 36 | 56 | [0.0, 1668.6] |
| BIOMD0000000397 | Sivakumar2011 | Taxonomy-Mammalia | 50 | 56 | [0.0, 208.0] |
| BIOMD0000000398 | Sivakumar2011 | Taxonomy-Mammalia | 27 | 37 | [0.0, 6.8] |
| BIOMD0000000399 | Jenkinson2011 | Taxonomy-Homo sapiens | 93 | 90 | [0.0, 437.2] |
| BIOMD0000000400 | Cooling2007 | Taxonomy-Mus musculus | 13 | 55 | [0.0, 10000.0] |
| BIOMD0000000401 | Ayati2010 | Taxonomy-Chordata | 3 | 15 | [0.0, 153.2] |
| BIOMD0000000402 | Ayati2010 | Taxonomy-Chordata | 4 | 21 | [0.0, 5000.0] |
| BIOMD0000000405 | Cookson2011 | Taxonomy-cellular organisms | 6 | 5 | [0.0, 47.2] |
| BIOMD0000000406 | Moriya2011 | Taxonomy-Schizosaccharomyces pombe | 24 | 118 | [0.0, 190.0] |
| BIOMD0000000407 | Schliemann2011 | Taxonomy-Homo sapiens\|Brenda Tissue Ontology-rhabdomyosarcoma cell line | 47 | 106 | [0.0, 6000.0] |
| BIOMD0000000409 | Queralt2006 | Taxonomy-Saccharomyces cerevisiae | 13 | 54 | [0.0, 521.6] |
| BIOMD0000000410 | Wegner2012 | Taxonomy-Mus musculus\|Brenda Tissue Ontology-hepatocyte | 53 | 151 | [0.0, 300.0] |
| BIOMD0000000411 | Heiland2012 | Taxonomy-Chlamydomonas reinhardtii | 7 | 35 | [0.0, 4.8] |
| BIOMD0000000412 | Pokhilko2012 | Taxonomy-Arabidopsis thaliana | 79 | 120 | [0.0, 60.0] |
| BIOMD0000000413 | Band2012 | Taxonomy-Arabidopsis | 5 | 10 | [0.0, 175.0] |
| BIOMD0000000414 | Band2012 | Taxonomy-Arabidopsis | 1 | 4 | [0.0, 200.0] |
| BIOMD0000000416 | Muraro2011 | Taxonomy-Arabidopsis thaliana\|Gene Ontology-auxin-activated signaling pathway\|Gene Ontology-regulation of cytokinin-activated signaling pathway | 32 | 50 | [0.0, 250.0] |
| BIOMD0000000417 | Ratushny2012 | Taxonomy-Saccharomyces cerevisiae | 1 | 12 | [0.0, 67.6] |
| BIOMD0000000418 | Ratushny2012 | Taxonomy-Saccharomyces cerevisiae | 1 | 11 | [0.0, 134.8] |
| BIOMD0000000419 | Ratushny2012 | Taxonomy-Saccharomyces cerevisiae | 3 | 13 | [0.0, 134.8] |
| BIOMD0000000420 | Ratushny2012 | Taxonomy-Saccharomyces cerevisiae | 2 | 13 | [0.0, 90.0] |
| BIOMD0000000421 | Ratushny2012 | Taxonomy-Saccharomyces cerevisiae | 2 | 13 | [0.0, 90.0] |
| BIOMD0000000422 | Middleton2012 | Taxonomy-Arabidopsis | 22 | 47 | [0.0, 141.0] |
| BIOMD0000000423 | Nyman2012 | Taxonomy-Murinae\|Brenda Tissue Ontology-adipocyte | 9 | 19 | [0.0, 26.0] |
| BIOMD0000000424 | Faratian2009 | Taxonomy-Homo sapiens | 55 | 114 | [0.0, 1537.6] |
| BIOMD0000000425 | Tan2012 | Taxonomy-Escherichia coli | 1 | 6 | [0.0, 21.2] |
| BIOMD0000000426 | Mosca2012 | Taxonomy-Homo sapiens\|Brenda Tissue Ontology-HeLa cell | 21 | 186 | [0.0, 0.0005] |
| BIOMD0000000427 | Bianconi2012 | Taxonomy-Homo sapiens | 21 | 54 | [0.0, 1500.0] |
| BIOMD0000000428 | Achcar2012 | Taxonomy-Trypanosoma brucei | 26 | 84 | [0.0, 12.2] |
| BIOMD0000000429 | Schaber2012 | Taxonomy-Saccharomyces cerevisiae | 15 | 88 | [0.0, 5000.0] |
| BIOMD0000000430 | Sarma2012 | Taxonomy-cellular organisms | 27 | 45 | [0.0, 1380.4] |
| BIOMD0000000431 | Sarma2012 | Taxonomy-cellular organisms | 27 | 45 | [0.0, 1275.2] |
| BIOMD0000000432 | Sarma2012 | Taxonomy-cellular organisms | 11 | 26 | [0.0, 916.0] |
| BIOMD0000000433 | Sarma2012 | Taxonomy-cellular organisms | 11 | 26 | [0.0, 535.2] |
| BIOMD0000000434 | McAuley2012 | Taxonomy-Homo sapiens | 26 | 43 | [0.0, 500.0] |
| BIOMD0000000435 | deBack2012 | Taxonomy-Murinae | 4 | 18 | [0.0, 22.8] |
| BIOMD0000000436 | Gupta2009 | Taxonomy-Mammalia | 12 | 77 | [0.0, 300.0] |
| BIOMD0000000439 | Smith2009 | Taxonomy-Schizosaccharomyces pombe | 20 | 19 | [0.0, 50.0] |
| BIOMD0000000440 | Sarma2012 | Taxonomy-cellular organisms | 11 | 25 | [0.0, 2200.0] |
| BIOMD0000000441 | Sarma2012 | Taxonomy-cellular organisms | 11 | 24 | [0.0, 800.0] |
| BIOMD0000000442 | Sarma2012B | Taxonomy-cellular organisms | 13 | 30 | [0.0, 850.0] |

*Continued on next page*

| Model ID | Reference | Domain | Sys. Dim. | Par. Dim. | Time Span |
|---|---|---|---|---|---|
| BIOMD0000000443 | Sarma2012 | Taxonomy-cellular organisms | 19 | 47 | [0.0, 2800.0] |
| BIOMD0000000444 | Sarma2012 | Taxonomy-cellular organisms | 18 | 47 | [0.0, 5000.0] |
| BIOMD0000000445 | Pokhilko2013 | Taxonomy-Arabidopsis thaliana | 32 | 143 | [0.0, 62.0] |
| BIOMD0000000446 | Erguler2013 | Taxonomy-cellular organisms | 27 | 94 | [0.0, 548.8] |
| BIOMD0000000447 | Venkatraman 2012 | Taxonomy-Rattus rattus | 13 | 23 | [0.0, 250.0] |
| BIOMD0000000448 | nnmark2013 | Taxonomy-Homo sapiens | 27 | 68 | [0.0, 100.0] |
| BIOMD0000000449 | nnmark2013 | Taxonomy-Homo sapiens | 27 | 68 | [0.0, 38.0] |
| BIOMD0000000452 | Bidkhori2012 | Taxonomy-cellular organisms | 109 | 187 | [0.0, 800.0] |
| BIOMD0000000453 | Bidkhori2012 | Taxonomy-cellular organisms | 109 | 188 | [0.0, 26.4] |
| BIOMD0000000458 | Smallbone2013 | Taxonomy-Escherichia coli | 2 | 10 | [0.0, 5.8] |
| BIOMD0000000459 | Liebal2012 | Taxonomy-Bacillus subtilis | 3 | 7 | [0.0, 1996.4] |
| BIOMD0000000460 | Liebal2012 | Taxonomy-Bacillus subtilis | 3 | 7 | [0.0, 300000.0] |
| BIOMD0000000461 | Liebal2012 | Taxonomy-Bacillus subtilis | 3 | 6 | [0.0, 157.6] |
| BIOMD0000000462 | Proctor2012 | Taxonomy-Homo sapiens | 4 | 9 | [0.0, 700.0] |
| BIOMD0000000463 | Heldt2012 | Taxonomy-Influenza A virus\|Taxonomy-Mammalia | 35 | 55 | [0.0, 75.0] |
| BIOMD0000000464 | Koo2013 | Taxonomy-Homo sapiens\|Gene Ontology-calcium ion import\|Reactome-eNOS synthesizes NO | 14 | 28 | [0.0, 1985.0] |
| BIOMD0000000465 | Koo2013 | Taxonomy-Homo sapiens | 16 | 23 | [0.0, 1744.2] |
| BIOMD0000000466 | Koo2013 | Taxonomy-Homo sapiens | 34 | 60 | [0.0, 5000.0] |
| BIOMD0000000467 | Koo2013 | Taxonomy-Homo sapiens | 19 | 18 | [0.0, 2000.0] |
| BIOMD0000000468 | Koo2013 | Taxonomy-Homo sapiens | 79 | 135 | [0.0, 1956.8] |
| BIOMD0000000475 | Amara2013 | Taxonomy-Saccharomyces cerevisiae | 23 | 26 | [0.0, 1985.6] |
| BIOMD0000000476 | Adams2012 | Taxonomy-Arabidopsis thaliana | 16 | 83 | [0.0, 55.0] |
| BIOMD0000000477 | Mol2013 | Taxonomy-Homo sapiens\|Reactome-Signaling by EGFR | 43 | 82 | [0.0, 30.0] |
| BIOMD0000000478 | Besozzi2012 | Taxonomy-Saccharomyces cerevisiae\|Gene Ontology-protein kinase A signaling\|Gene Ontology-cAMP-mediated signaling\|Pathway Ontology-protein kinase A (PKA) signaling pathway | 30 | 39 | [0.0, 155.0] |
| BIOMD0000000479 | Croft2013 | Taxonomy-Saccharomyces cerevisiae | 28 | 42 | [0.0, 100.0] |
| BIOMD0000000481 | tzel2012 | Taxonomy-Bos taurus | 16 | 69 | [0.0, 45.0] |
| BIOMD0000000482 | Noguchi2013 | Taxonomy-Rattus | 23 | 56 | [0.0, 100000.0] |
| BIOMD0000000484 | Cao2013 | Taxonomy-cellular organisms | 2 | 2 | [0.0, 276.8] |
| BIOMD0000000485 | Cao2013 | Taxonomy-cellular organisms | 2 | 8 | [0.0, 4.2] |
| BIOMD0000000488 | Proctor2013 | Taxonomy-Mammalia\|Pathway Ontology-Alzheimer disease pathway | 64 | 73 | [0.0, 907.4] |
| BIOMD0000000489 | Sharp2013 | Taxonomy-Mus musculus | 35 | 87 | [0.0, 30000.0] |
| BIOMD0000000490 | Demin2013 | Taxonomy-Homo sapiens | 33 | 263 | [0.0, 150.0] |
| BIOMD0000000491 | Pathak2013 | Taxonomy-Viridiplantae | 57 | 172 | [0.0, 56.0] |
| BIOMD0000000492 | Pathak2013 | Taxonomy-Viridiplantae | 52 | 176 | [0.0, 36.0] |
| BIOMD0000000493 | Schittler2010 | Taxonomy-cellular organisms | 3 | 23 | [0.0, 850.0] |
| BIOMD0000000502 | Messiha2013 | Taxonomy-Saccharomyces cerevisiae | 8 | 52 | [0.0, 800.0] |
| BIOMD0000000503 | Messiha2013 | Taxonomy-Saccharomyces cerevisiae\|Gene Ontology-glycolytic process\|Gene Ontology-pentose-phosphate shunt | 28 | 192 | [0.0, 256.2] |
| BIOMD0000000504 | Proctor2013 | Taxonomy-Homo sapiens | 73 | 131 | [0.0, 5000.0] |
| BIOMD0000000505 | vanEunen2013 | Taxonomy-Rattus | 45 | 224 | [0.0, 11.2] |
| BIOMD0000000506 | vanEunen2013 | Taxonomy-Rattus | 45 | 224 | [0.0, 70.0] |
| BIOMD0000000507 | Gardner2000 | Taxonomy-Escherichia coli | 3 | 11 | [0.0, 8.0] |
| BIOMD0000000508 | Barrack2014 | Taxonomy-Mammalia | 16 | 48 | [0.0, 52.5] |
| BIOMD0000000509 | Barrack2014 | Taxonomy-Mammalia | 16 | 48 | [0.0, 60.0] |

*Continued on next page*

| Model ID | Reference | Domain | Sys. Dim. | Par. Dim. | Time Span |
|---|---|---|---|---|---|
| BIOMD0000000510 | Kerkhoven2013 | Taxonomy-Trypanosoma brucei\|Gene Ontology-glycolytic process\|Gene Ontology-pentose-phosphate shunt | 38 | 156 | [0.0, 13.0] |
| BIOMD0000000511 | Kerkhoven2013 | Taxonomy-Trypanosoma brucei\|Gene Ontology-glycolytic process\|Gene Ontology-pentose-phosphate shunt | 37 | 154 | [0.0, 39.2] |
| BIOMD0000000512 | Benson2014 | Taxonomy-Homo sapiens | 39 | 155 | [0.0, 1490.4] |
| BIOMD0000000513 | Kerkhoven2013 | Taxonomy-Trypanosoma brucei | 24 | 90 | [0.0, 14.8] |
| BIOMD0000000514 | Kerkhoven2013 | Taxonomy-Trypanosoma brucei\|Gene Ontology-glycolytic process\|Gene Ontology-pentose-phosphate shunt | 37 | 150 | [0.0, 40.0] |
| BIOMD0000000515 | Kerkhoven2013 | Taxonomy-Trypanosoma brucei\|Gene Ontology-glycolytic process\|Gene Ontology-pentose-phosphate shunt | 40 | 181 | [0.0, 34.8] |
| BIOMD0000000516 | Kerkhoven2013 | Taxonomy-Trypanosoma brucei\|Gene Ontology-glycolytic process\|Gene Ontology-pentose-phosphate shunt | 39 | 179 | [0.0, 392.4] |
| BIOMD0000000517 | Smallbone2013 | Taxonomy-Murinae | 4 | 20 | [0.0, 187.8] |
| BIOMD0000000521 | Ribba2012 | Taxonomy-Homo sapiens | 4 | 10 | [0.0, 500.0] |
| BIOMD0000000522 | Muraro2014 | Taxonomy-Arabidopsis | 16 | 69 | [0.0, 6.0] |
| BIOMD0000000523 | Kallenberger 2014 | Taxonomy-Homo sapiens | 18 | 12 | [0.0, 672.6] |
| BIOMD0000000524 | Kallenberger 2014 | Taxonomy-Homo sapiens | 18 | 12 | [0.0, 900.0] |
| BIOMD0000000525 | Kallenberger 2014 | Taxonomy-Homo sapiens | 18 | 15 | [0.0, 758.8] |
| BIOMD0000000526 | Kallenberger 2014 | Taxonomy-Homo sapiens | 18 | 15 | [0.0, 1200.0] |
| BIOMD0000000528 | Fribourg2014 | Taxonomy-H1N1 swine influenza virus\| Taxonomy-Homo sapiens | 12 | 49 | [0.0, 300.0] |
| BIOMD0000000529 | Fribourg2014 | Taxonomy-Influenza A virus (A/New Caledonia/20/1999(H1N1))\|Taxonomy-Homo sapiens | 12 | 49 | [0.0, 300.0] |
| BIOMD0000000530 | Schmitz2014 | Taxonomy-Homo sapiens | 10 | 17 | [0.0, 5.4] |
| BIOMD0000000534 | Dwivedi2014 | Taxonomy-Homo sapiens\|Gene Ontology-JAK-STAT cascade | 39 | 51 | [0.0, 400.0] |
| BIOMD0000000535 | Dwivedi2014 | Taxonomy-Homo sapiens\|Gene Ontology-JAK-STAT cascade | 39 | 51 | [0.0, 0.4] |
| BIOMD0000000537 | Dwivedi2014 | Taxonomy-Homo sapiens\|Gene Ontology-JAK-STAT cascade | 40 | 53 | [0.0, 200.0] |
| BIOMD0000000539 | ois2005 | Taxonomy-cellular organisms | 6 | 11 | [0.0, 400.0] |
| BIOMD0000000540 | Yugi2014 | Taxonomy-cellular organisms | 21 | 22 | [0.0, 100.0] |
| BIOMD0000000541 | Yugi2014 | Taxonomy-cellular organisms | 30 | 44 | [0.0, 3000.0] |
| BIOMD0000000543 | Qi2013 | Taxonomy-cellular organisms\|Gene Ontology-interferon-gamma-mediated signaling pathway\|Gene Ontology-interleukin-6-mediated signaling pathway | 105 | 204 | [0.0, 8000.0] |
| BIOMD0000000544 | Qi2013 | Taxonomy-cellular organisms\|Gene Ontology-interferon-gamma-mediated signaling pathway\|Gene Ontology-interleukin-6-mediated signaling pathway | 102 | 192 | [0.0, 8000.0] |
| BIOMD0000000545 | Ouyang2014 | Taxonomy-Arabidopsis thaliana | 14 | 25 | [0.0, 20.4] |
| BIOMD0000000546 | Miao2010 | Taxonomy-Influenza A virus (strain A/X-31 H3N2)\|Taxonomy-Murinae | 3 | 5 | [0.0, 12.0] |
| BIOMD0000000547 | Talemi2014 | Taxonomy-Saccharomyces cerevisiae | 11 | 52 | [0.0, 200.0] |
| BIOMD0000000548 | Sneppen2009 | Taxonomy-Homo sapiens | 3 | 4 | [0.0, 85.0] |
| BIOMD0000000552 | Ehrenstein2000 | Taxonomy-Homo sapiens | 2 | 4 | [0.0, 451.2] |

| Model ID | Reference | Domain | Sys. Dim. | Par. Dim. | Time Span |
|---|---|---|---|---|---|
| BIOMD0000000553 | Ehrenstein1997 | Taxonomy-Homo sapiens | 2 | 4 | [0.0, 451.2] |
| BIOMD0000000557 | Reiterer2013 | Taxonomy-Homo sapiens | 25 | 50 | [0.0, 300.0] |
| BIOMD0000000558 | Cloutier2012 | Taxonomy-Homo sapiens | 2 | 8 | [0.0, 198.6] |
| BIOMD0000000559 | Ouzounoglou 2014 | Taxonomy-Homo sapiens | 89 | 23 | [0.0, 5000.0] |
| BIOMD0000000560 | Hui2016 | Taxonomy-Mus musculus | 62 | 113 | [0.0, 1231.0] |
| BIOMD0000000563 | Pritchard2014 | Taxonomy-Pseudomonas syringae\| Taxonomy-Arabidopsis | 10 | 19 | [0.0, 102.4] |
| BIOMD0000000564 | Gould2013 | Taxonomy-Arabidopsis thaliana | 19 | 115 | [0.0, 55.0] |
| BIOMD0000000565 | Machado2014 | Taxonomy-Escherichia coli | 26 | 150 | [0.0, 1844.6] |
| BIOMD0000000568 | Mueller2015 | Taxonomy-Mus musculus | 24 | 79 | [0.0, 427.8] |
| BIOMD0000000572 | Costa2014 | Taxonomy-Lactococcus lactis | 26 | 112 | [0.0, 1430.6] |
| BIOMD0000000573 | Aguilera2014 | Taxonomy-Homo sapiens | 2 | 8 | [0.0, 1750.0] |
| BIOMD0000000575 | Sass2009 | Taxonomy-Homo sapiens | 49 | 245 | [0.0, 1000.0] |
| BIOMD0000000576 | Kolodkin2013 | Taxonomy-Homo sapiens | 34 | 61 | [0.0, 1185.2] |
| BIOMD0000000579 | Sengupta2015 | Taxonomy-Homo sapiens | 240 | 272 | [0.0, 500.0] |
| BIOMD0000000580 | Sonntag2012 | Taxonomy-Mus musculus\|Taxonomy-Homo sapiens | 26 | 37 | [0.0, 1636.2] |
| BIOMD0000000581 | DallePezze2012 | Taxonomy-Mus musculus\|Taxonomy-Homo sapiens | 25 | 32 | [0.0, 631.6] |
| BIOMD0000000582 | DallePezze2014 | Taxonomy-Homo sapiens | 23 | 55 | [0.0, 64.0] |
| BIOMD0000000584 | Mandlik2015 | Taxonomy-Leishmania | 21 | 34 | [0.0, 500.0] |
| BIOMD0000000586 | Karapetyan2016 | Taxonomy-Saccharomyces cerevisiae | 10 | 25 | [0.0, 800.0] |
| BIOMD0000000587 | Karapetyan2016 | Taxonomy-Saccharomyces cerevisiae | 10 | 25 | [0.0, 400.0] |
| BIOMD0000000590 | Hermansen2015 | - | 9 | 30 | [0.0, 12.0] |
| BIOMD0000000591 | Boehm2014 | Taxonomy-Mus musculus | 8 | 7 | [0.0, 700.0] |
| BIOMD0000000596 | Philipson2015 | Helicobacter pylori, Mus musculus \| biological process involved in symbiotic interaction; innate immune response; response to bacterium | 15 | 56 | [0.0, 3000.0] |
| BIOMD0000000597 | Flis2015 | Taxonomy-Arabidopsis thaliana | 28 | 121 | [0.0, 60.0] |
| BIOMD0000000598 | Flis2015 | Taxonomy-Arabidopsis thaliana | 28 | 121 | [0.0, 62.0] |
| BIOMD0000000599 | Coggins2014 | Taxonomy-Homo sapiens | 30 | 40 | [0.0, 1984.0] |
| BIOMD0000000600 | re2011 | transforming growth factor beta receptor signaling pathway | 17 | 20 | [0.0, 60000.0] |
| BIOMD0000000604 | Elzbieta_ Petelenz_ Kurdziel2013 | Taxonomy-Saccharomyces cerevisiae | 29 | 127 | [0.0, 5500.0] |
| BIOMD0000000605 | PetelenzKuehn_ osmoadaptation_ HOG1att | Taxonomy-Saccharomyces cerevisiae | 29 | 127 | [0.0, 4000.0] |
| BIOMD0000000607 | PetelenzKuehn_ osmoadaptation_ fps1D1 | Taxonomy-Saccharomyces cerevisiae | 29 | 127 | [0.0, 80000.0] |
| BIOMD0000000611 | Nayak2015 | Taxonomy-Homo sapiens | 66 | 106 | [0.0, 3000.0] |
| BIOMD0000000612 | Proctor2016 | Taxonomy-Homo sapiens | 35 | 72 | [0.0, 200000.0] |
| BIOMD0000000613 | Peterson2010 | Taxonomy-Homo sapiens | 31 | 268 | [0.0, 2000.0] |
| BIOMD0000000614 | Kamihira2000 | Taxonomy-Homo sapiens | 1 | 3 | [0.0, 8000.0] |
| BIOMD0000000615 | Kuznetsov2016 | Taxonomy-Homo sapiens | 4 | 12 | [0.0, 8000.0] |
| BIOMD0000000616 | Dunster2014 | Taxonomy-Homo sapiens | 4 | 8 | [0.0, 300.0] |
| BIOMD0000000617 | Walsh2014 | Taxonomy-Homo sapiens | 1 | 26 | [0.0, 500.0] |
| BIOMD0000000620 | Palmer2014 | Taxonomy-Homo sapiens | 34 | 52 | [0.0, 50000.0] |
| BIOMD0000000621 | Palmer2014 | Taxonomy-Homo sapiens | 34 | 52 | [0.0, 50000.0] |
| BIOMD0000000622 | NguyenLK2011 | Taxonomy-cellular organisms | 10 | 24 | [0.0, 1910.0] |
| BIOMD0000000623 | Orton2009 | Taxonomy-Rattus norvegicus | 25 | 54 | [0.0, 197.6] |

*Continued on next page*

| Model ID | Reference | Domain | Sys. Dim. | Par. Dim. | Time Span |
|---|---|---|---|---|---|
| BIOMD0000000624 | Sluka2016 | Taxonomy-Homo sapiens | 6 | 9 | [0.0, 3000.0] |
| BIOMD0000000625 | Leber2016 | Taxonomy-Mus musculus\|Taxonomy-Helicobacter pylori | 17 | 50 | [0.0, 550.0] |
| BIOMD0000000626 | Ray2013 | Taxonomy-Saccharomyces cerevisiae | 6 | 20 | [0.0, 55.2] |
| BIOMD0000000630 | Venkatraman 2011 | Taxonomy-cellular organisms | 4 | 12 | [0.0, 450.4] |
| BIOMD0000000631 | DeCaluwe2016 | Taxonomy-Arabidopsis thaliana | 12 | 47 | [0.0, 55.0] |
| BIOMD0000000640 | DallePezze2016 | Taxonomy-Mammalia | 31 | 60 | [0.0, 54.8] |
| BIOMD0000000642 | Mufudza2012 | Taxonomy-Homo sapiens | 3 | 18 | [0.0, 50.0] |
| BIOMD0000000646 | Barr2016 | Taxonomy-Homo sapiens | 11 | 26 | [0.0, 5000.0] |
| BIOMD0000000647 | Kwang2003 | KEGG Pathway-MAPK signaling pathway | 11 | 11 | [0.0, 68.8] |
| BIOMD0000000648 | Padala2017 | Taxonomy-Homo sapiens\|Gene Ontology-MAPK cascade\|KEGG Pathway-MAPK signaling pathway\|KEGG Pathway-PI3K-Akt signaling pathway\|KEGG Pathway-Wnt signaling pathway | 53 | 103 | [0.0, 882.4] |
| BIOMD0000000650 | Owen1998 | regulation of immune response to tumor cell | 3 | 13 | [0.0, 546.8] |
| BIOMD0000000651 | Nguyen2016 | Taxonomy-Homo sapiens\|Pathway Ontology-the extracellular signal-regulated Raf/Mek/Erk signaling pathway\|Pathway Ontology-mTOR signaling pathway | 29 | 34 | [0.0, 1600.0] |
| BIOMD0000000652 | Padala2017 | Taxonomy-Homo sapiens\|Gene Ontology-MAPK cascade\|KEGG Pathway-MAPK signaling pathway\|KEGG Pathway-PI3K-Akt signaling pathway\|KEGG Pathway-Wnt signaling pathway | 53 | 102 | [0.0, 1600.0] |
| BIOMD0000000653 | Padala2017 | Taxonomy-Homo sapiens\|Gene Ontology-MAPK cascade\|KEGG Pathway-MAPK signaling pathway\|KEGG Pathway-PI3K-Akt signaling pathway\|KEGG Pathway-Wnt signaling pathway | 53 | 99 | [0.0, 1196.0] |
| BIOMD0000000654 | Padala2017 | Taxonomy-Homo sapiens\|Gene Ontology-MAPK cascade\|KEGG Pathway-MAPK signaling pathway\|KEGG Pathway-PI3K-Akt signaling pathway\|KEGG Pathway-Wnt signaling pathway | 53 | 101 | [0.0, 50000.0] |
| BIOMD0000000655 | Padala2017 | Taxonomy-Homo sapiens\|Gene Ontology-MAPK cascade\|KEGG Pathway-MAPK signaling pathway\|KEGG Pathway-PI3K-Akt signaling pathway\|KEGG Pathway-Wnt signaling pathway | 53 | 101 | [0.0, 600.0] |
| BIOMD0000000656 | Padala2017 | Taxonomy-Homo sapiens\|Gene Ontology-MAPK cascade\|KEGG Pathway-MAPK signaling pathway\|KEGG Pathway-PI3K-Akt signaling pathway\|KEGG Pathway-Wnt signaling pathway | 53 | 101 | [0.0, 1868.4] |
| BIOMD0000000657 | Araujo2016 | Taxonomy-Homo sapiens\|Pathway Ontology-M phase pathway | 3 | 21 | [0.0, 3000.0] |
| BIOMD0000000660 | Barr2017 | Taxonomy-Homo sapiens | 15 | 31 | [0.0, 1904.4] |
| BIOMD0000000662 | Moore2004 | Taxonomy-Homo sapiens | 3 | 12 | [0.0, 60.0] |
| BIOMD0000000663 | Wodarz2007 | Taxonomy-Human immunodeficiency virus 1\|Taxonomy-Homo sapiens | 3 | 7 | [0.0, 19.2] |
| BIOMD0000000664 | Muller2008 | Taxonomy-Homo sapiens | 6 | 20 | [0.0, 80.0] |

| Model ID | Reference | Domain | Sys. Dim. | Par. Dim. | Time Span |
|---|---|---|---|---|---|
| BIOMD0000000666 | Pappalardo2016 | Experimental Factor Ontology-response to dabrafenib\|Reactome-PI3K/AKT Signaling in Cancer\|Reactome-MAPK family signaling cascades\|KEGG Pathway-MAPK signaling pathway\|KEGG Pathway-PI3K-Akt signaling pathway | 35 | 87 | [0.0, 45.0] |
| BIOMD0000000667 | Hornberg2005 | regulation of MAPK cascade | 103 | 83 | [0.0, 15000.0] |
| BIOMD0000000668 | Zhu2015 | KEGG Drug-Gemcitabine (USAN/INN)\|KEGG Drug-Birinapant (USAN/INN)\|Brenda Tissue Ontology-pancreas\|NCIt-Combination Chemotherapy\|KEGG Drug-Gemcitabine (USAN/INN)\|KEGG Drug-Birinapant (USAN/INN)\|BioModels Database-MODEL1604270000 | 10 | 26 | [0.0, 400.0] |
| BIOMD0000000670 | Owen1998 | regulation of immune response to tumor cell | 3 | 8 | [0.0, 122.4] |
| BIOMD0000000676 | Chen2006 | nitric oxide biosynthetic process | 13 | 21 | [0.0, 2.0] |
| BIOMD0000000677 | Holmes2006 | muscle contraction | 1 | 10 | [0.0, 4.0] |
| BIOMD0000000678 | Tomida2003 | Taxonomy-Homo sapiens | 2 | 11 | [0.0, 60.0] |
| BIOMD0000000679 | Waugh2006 | Taxonomy-Homo sapiens | 3 | 11 | [0.0, 24.0] |
| BIOMD0000000682 | Wierschem2004 | Taxonomy-Homo sapiens | 5 | 34 | [0.0, 600000.0] |
| BIOMD0000000683 | Wodarz1999 | Taxonomy-Homo sapiens | 4 | 12 | [0.0, 45.0] |
| BIOMD0000000684 | Wodarz2003 | Taxonomy-Homo sapiens | 10 | 28 | [0.0, 231.8] |
| BIOMD0000000685 | Wodarz2003 | Taxonomy-Homo sapiens\|Brenda Tissue Ontology-cytotoxic T-lymphocyte\|Cell Type Ontology-professional antigen presenting cell | 4 | 17 | [0.0, 55.0] |
| BIOMD0000000686 | Wodarz2007 | Taxonomy-Mus | 5 | 16 | [0.0, 76.0] |
| BIOMD0000000687 | Wodarz2007 | Taxonomy-Mus\|Experimental Factor Ontology-cytomegalovirus infection\|Brenda Tissue Ontology-cytotoxic T-lymphocyte | 15 | 16 | [0.0, 60.0] |
| BIOMD0000000688 | Wodarz2007 | Taxonomy-Mus\|Experimental Factor Ontology-cytomegalovirus infection\|Brenda Tissue Ontology-cytotoxic T-lymphocyte\|Brenda Tissue Ontology-natural killer cell | 16 | 20 | [0.0, 60.0] |
| BIOMD0000000691 | Wolf2000 | Taxonomy-Saccharomyces | 13 | 14 | [0.0, 4.0] |
| BIOMD0000000692 | Phillips2003 | Taxonomy-Escherichia coli\|KEGG Orthology-neurofibromin 1 | 8 | 12 | [0.0, 0.8] |
| BIOMD0000000695 | FelixGarza2017 | Experimental Factor Ontology-psoriasis\|Brenda Tissue Ontology-keratinocyte\|Gene Ontology-response to blue light | 12 | 55 | [0.0, 2000.0] |
| BIOMD0000000696 | Boada2016 | - | 11 | 41 | [0.0, 700.0] |
| BIOMD0000000697 | Ciliberto2003 | NCIt-Xenopus laevis\|Gene Ontology-regulation of cell cycle | 13 | 25 | [0.0, 1000.0] |
| BIOMD0000000698 | Reed2004 | Taxonomy-Homo sapiens | 5 | 23 | [0.0, 10.0] |
| BIOMD0000000700 | Heldt2018 | Taxonomy-Homo sapiens\|Gene Ontology-DNA damage response, detection of DNA damage\|Gene Ontology-regulation of cell cycle | 23 | 56 | [0.0, 1976.8] |
| BIOMD0000000703 | Diedrichs2018 | Reactome-unfolded protein [endoplasmic reticulum lumen] | 11 | 81 | [0.0, 3000.0] |
| BIOMD0000000704 | Aguda1999 | Taxonomy-Vertebrata\|Gene Ontology-G2 DNA damage checkpoint | 16 | 40 | [0.0, 70.0] |

| Model ID | Reference | Domain | Sys. Dim. | Par. Dim. | Time Span |
|---|---|---|---|---|---|
| BIOMD0000000705 | Smith2010 | Taxonomy-Eukaryota\|NCIt-Apoptosis\|Gene Ontology-metabolic process\|Gene Ontology-post-translational protein modification\|Gene Ontology-regulation of cell cycle | 32 | 19 | [0.0, 1982.8] |
| BIOMD0000000706 | Smith2010 | Taxonomy-Eukaryota\|Gene Ontology-post-translational protein modification | 45 | 128 | [0.0, 120.0] |
| BIOMD0000000707 | Revilla2003 | Taxonomy-Human immunodeficiency virus 1\|Taxonomy-Homo sapiens | 5 | 10 | [0.0, 200.0] |
| BIOMD0000000708 | Liu2017 | Taxonomy-Aves\|Taxonomy-Homo sapiens | 5 | 12 | [0.0, 30000.0] |
| BIOMD0000000709 | Liu2017 | Taxonomy-Aves\|Taxonomy-Homo sapiens | 5 | 12 | [0.0, 80000.0] |
| BIOMD0000000710 | Vargas2012 | Taxonomy-Influenza A virus\|Taxonomy-Homo sapiens | 7 | 17 | [0.0, 90.8] |
| BIOMD0000000713 | Aston2018 | Taxonomy-Hepacivirus C\|Taxonomy-Homo sapiens | 3 | 8 | [0.0, 500.0] |
| BIOMD0000000714 | Reynolds2006 | Taxonomy-Homo sapiens | 4 | 25 | [0.0, 765.4] |
| BIOMD0000000715 | Huo2017 | Taxonomy-Homo sapiens | 4 | 11 | [0.0, 15.0] |
| BIOMD0000000716 | Lee2018 | Taxonomy-H5N6 subtype\|Taxonomy-Aves\|Taxonomy-Homo sapiens | 4 | 20 | [0.0, 1500.0] |
| BIOMD0000000717 | Lee2018 | Taxonomy-H5N6 subtype\|Taxonomy-Aves\|Taxonomy-Homo sapiens | 4 | 20 | [0.0, 3000.0] |
| BIOMD0000000718 | Li2008 | Taxonomy-Caulobacter vibrioides | 16 | 50 | [0.0, 600.0] |
| BIOMD0000000719 | Tsai2014 | Taxonomy-Xenopus laevis\|Gene Ontology-regulation of cell cycle | 5 | 19 | [0.0, 150.0] |
| BIOMD0000000720 | Yan2012 | Taxonomy-Mammalia\|Gene Ontology-cell cycle | 8 | 29 | [0.0, 120.0] |
| BIOMD0000000721 | Graham2013 | Taxonomy-Homo sapiens | 5 | 25 | [0.0, 55.0] |
| BIOMD0000000722 | Bianchi2015 | Taxonomy-Mus musculus\|Taxonomy-Rattus norvegicus | 5 | 33 | [0.0, 243.6] |
| BIOMD0000000724 | Theinmozhi2018 | Taxonomy-Homo sapiens | 28 | 51 | [0.0, 10000.0] |
| BIOMD0000000725 | Sora2016 | - | 38 | 93 | [0.0, 1200.0] |
| BIOMD0000000726 | Ruan2017 | Taxonomy-Rabies lyssavirus\|Taxonomy-Homo sapiens\|Taxonomy-Canis lupus familiaris | 8 | 16 | [0.0, 971.2] |
| BIOMD0000000727 | Li2009 | Taxonomy-Caulobacter vibrioides | 30 | 96 | [0.0, 300.0] |
| BIOMD0000000728 | Norel1990 | Taxonomy-Homo sapiens | 2 | 4 | [0.0, 7.0] |
| BIOMD0000000729 | Goldbeter1996 | Taxonomy-Eukaryota | 3 | 15 | [0.0, 45.0] |
| BIOMD0000000730 | Gerard2009 | Taxonomy-Mammalia | 45 | 187 | [0.0, 75.0] |
| BIOMD0000000731 | Robertson-Tessi M 2012 | Taxonomy-Homo sapiens\|NCIt-Tumor-Infiltrating Immune Cell | 16 | 48 | [0.0, 50000.0] |
| BIOMD0000000732 | Kirschner1998 | Taxonomy-Homo sapiens\|NCIt-Tumor-Infiltrating Immune Cell | 3 | 14 | [0.0, 3500.0] |
| BIOMD0000000733 | Moore2004 | Taxonomy-Homo sapiens | 3 | 12 | [0.0, 666.4] |
| BIOMD0000000736 | Mouse_Iron_Distribution_Adequate_iron_diet_No_Tracer_ | Taxonomy-Mus musculus | 10 | 32 | [0.0, 973.6] |
| BIOMD0000000737 | Mouse_Iron_Distribution_Deficient_iro_diet_No_Tracer_ | Taxonomy-Mus musculus | 11 | 32 | [0.0, 1412.0] |
| BIOMD0000000738 | Mouse_Iron_Distribution_Rich_iron_diet_No_Tracer_ | Taxonomy-Mus musculus | 11 | 32 | [0.0, 51.6] |
| BIOMD0000000742 | Victor_Garcia 2018 | Taxonomy-Homo sapiens | 2 | 6 | [0.0, 48.8] |

| Model ID | Reference | Domain | Sys. Dim. | Par. Dim. | Time Span |
|---|---|---|---|---|---|
| BIOMD0000000743 | Gallaher2018 | Taxonomy-Homo sapiens | 4 | 31 | [0.0, 349.8] |
| BIOMD0000000744 | Hu2019 | Taxonomy-Homo sapiens | 5 | 23 | [0.0, 630.4] |
| BIOMD0000000746 | Saad2017 | Taxonomy-Homo sapiens | 4 | 12 | [0.0, 50.0] |
| BIOMD0000000747 | Nagashima2002 | Homo sapiens \| blood coagulation | 33 | 44 | [0.0, 10000.0] |
| BIOMD0000000748 | Phan2017 | Taxonomy-Homo sapiens | 4 | 8 | [0.0, 200.0] |
| BIOMD0000000750 | Lolas2016 | Taxonomy-Homo sapiens | 8 | 41 | [0.0, 3.6] |
| BIOMD0000000752 | Wilkie2013r | Taxonomy-Homo sapiens | 3 | 9 | [0.0, 500.0] |
| BIOMD0000000753 | Figueredo2013 | Taxonomy-Homo sapiens | 2 | 8 | [0.0, 9.8] |
| BIOMD0000000754 | Figueredo2013 | Taxonomy-Homo sapiens | 3 | 13 | [0.0, 1000.0] |
| BIOMD0000000756 | Figueredo2013 | Taxonomy-Homo sapiens | 4 | 20 | [0.0, 3000.0] |
| BIOMD0000000757 | Abernathy2016 | Taxonomy-Homo sapiens | 7 | 32 | [0.0, 300.0] |
| BIOMD0000000759 | Breems2015 | Taxonomy-Homo sapiens | 6 | 24 | [0.0, 64.4] |
| BIOMD0000000761 | Cappuccio2006 | Taxonomy-Mus musculus\|Gene Ontology-T cell mediated cytotoxicity directed against tumor cell target\|Gene Ontology-natural killer cell mediated cytotoxicity directed against tumor cell target | 6 | 22 | [0.0, 434.8] |
| BIOMD0000000762 | Kuznetsov1994 | T cell mediated immune response to tumor cell | 2 | 8 | [0.0, 1000.0] |
| BIOMD0000000763 | Dritschel2018 | T cell mediated immune response to tumor cell | 3 | 8 | [0.0, 50.0] |
| BIOMD0000000765 | Mager2005 | Taxonomy-Unknown | 4 | 13 | [0.0, 80.0] |
| BIOMD0000000766 | Macnamara2015 | Taxonomy-Homo sapiens | 5 | 15 | [0.0, 80.0] |
| BIOMD0000000767 | Macnamara2015 | Taxonomy-Homo sapiens | 3 | 8 | [0.0, 13.6] |
| BIOMD0000000768 | Eftimie2010 | Taxonomy-Homo sapiens | 5 | 23 | [0.0, 10.4] |
| BIOMD0000000769 | Eftimie2017 | Taxonomy-Homo sapiens | 5 | 27 | [0.0, 53.6] |
| BIOMD0000000770 | Eftimie2017 | Taxonomy-Homo sapiens | 4 | 16 | [0.0, 50.0] |
| BIOMD0000000771 | Bajzer2008 | - | 3 | 8 | [0.0, 38.0] |
| BIOMD0000000773 | Wodarz2018 | Taxonomy-Homo sapiens | 3 | 21 | [0.0, 4000.0] |
| BIOMD0000000774 | Wodarz2018 | Taxonomy-Homo sapiens | 2 | 13 | [0.0, 5000.0] |
| BIOMD0000000775 | Iarosz2015 | Taxonomy-Homo sapiens | 4 | 15 | [0.0, 621.6] |
| BIOMD0000000776 | Monro2008 | Taxonomy-Homo sapiens | 2 | 7 | [0.0, 1660.8] |
| BIOMD0000000777 | Chakrabarty2010 | - | 3 | 9 | [0.0, 244.0] |
| BIOMD0000000778 | Wei2017 | Taxonomy-Homo sapiens | 3 | 14 | [0.0, 35000.0] |
| BIOMD0000000779 | dePillis2009 | - | 6 | 44 | [0.0, 26.2] |
| BIOMD0000000780 | Wang2016 | Taxonomy-Homo sapiens | 4 | 15 | [0.0, 1658.4] |
| BIOMD0000000781 | Wang2016 | Taxonomy-Homo sapiens | 3 | 10 | [0.0, 400.0] |
| BIOMD0000000782 | Wang2016 | Taxonomy-Homo sapiens | 2 | 6 | [0.0, 187.2] |
| BIOMD0000000783 | Dong2014 | - | 3 | 8 | [0.0, 500.0] |
| BIOMD0000000784 | Lopez2014 | Taxonomy-Mus musculus | 3 | 16 | [0.0, 36.8] |
| BIOMD0000000785 | Costa2003 | T cell mediated immune response to tumor cell | 2 | 3 | [0.0, 100.0] |
| BIOMD0000000786 | Lipniacki2004 | Mus musculus \| response to tumor necrosis factor | 15 | 35 | [0.0, 40000.0] |
| BIOMD0000000787 | Frascoli2014 | response to tumor cell; T cell mediated cytotoxicity; T cell mediated cytotoxicity directed against tumor cell target | 2 | 6 | [0.0, 285.6] |
| BIOMD0000000788 | Schropp2019 | Taxonomy-Homo sapiens | 8 | 21 | [0.0, 97.2] |
| BIOMD0000000789 | Jenner2018 | Taxonomy-Homo sapiens | 3 | 8 | [0.0, 150.0] |
| BIOMD0000000790 | Alvarez2019 | immune response to tumor cell | 4 | 18 | [0.0, 783.4] |
| BIOMD0000000791 | Wilson2012 | Taxonomy-Homo sapiens | 5 | 13 | [0.0, 347.6] |
| BIOMD0000000792 | Hu2019 | immune response to tumor cell | 6 | 28 | [0.0, 500.0] |
| BIOMD0000000793 | Chen2011 | Taxonomy-Homo sapiens | 2 | 4 | [0.0, 66.0] |
| BIOMD0000000794 | Benary2019 | Mus musculus \| response to tumor necrosis factor | 16 | 33 | [0.0, 8500.0] |

| Model ID | Reference | Domain | Sys. Dim. | Par. Dim. | Time Span |
|---|---|---|---|---|---|
| BIOMD0000000795 | Chen2011 | Taxonomy-Homo sapiens | 2 | 6 | [0.0, 1083.0] |
| BIOMD0000000796 | Yang2012 | Taxonomy-Homo sapiens | 5 | 27 | [0.0, 800.0] |
| BIOMD0000000797 | Hu2018 | immune response to tumor cell | 4 | 19 | [0.0, 50.0] |
| BIOMD0000000798 | Sharp2019 | Taxonomy-Homo sapiens | 5 | 14 | [0.0, 46.4] |
| BIOMD0000000799 | Cucuianu2010 | - | 2 | 6 | [0.0, 100.0] |
| BIOMD0000000800 | Precup2012 | - | 3 | 10 | [0.0, 1000.0] |
| BIOMD0000000801 | Sturrock2015 | Taxonomy-Homo sapiens | 4 | 17 | [0.0, 10000.0] |
| BIOMD0000000802 | Hoffman2018 | Taxonomy-Homo sapiens | 4 | 11 | [0.0, 5000.0] |
| BIOMD0000000803 | Park2019 | T cell homeostasis | 9 | 10 | [0.0, 10.0] |
| BIOMD0000000804 | Koenders2015 | Taxonomy-Homo sapiens | 3 | 20 | [0.0, 180.0] |
| BIOMD0000000805 | Husari2013 | Taxonomy-Homo sapiens | 4 | 14 | [0.0, 6.6] |
| BIOMD0000000806 | Eftimie2019 | Taxonomy-Mus musculus\|Taxonomy-Vesicular stomatitis virus | 6 | 32 | [0.0, 476.6] |
| BIOMD0000000808 | Kronik2008 | - | 6 | 40 | [0.0, 3200.0] |
| BIOMD0000000809 | Malinzi2018 | regulation of immune response to tumor cell; dormancy process | 5 | 13 | [0.0, 19.6] |
| BIOMD0000000810 | Ganguli2018 | Human Disease Ontology-breast cancer | 13 | 70 | [0.0, 1000.0] |
| BIOMD0000000811 | He2017 | - | 6 | 52 | [0.0, 1267.8] |
| BIOMD0000000812 | Galante2012 | Homo sapiens \| programmed cell death | 4 | 16 | [0.0, 300.0] |
| BIOMD0000000813 | Anderson2015 | Homo sapiens | 3 | 12 | [0.0, 400.0] |
| BIOMD0000000814 | Víctor2019 | - | 3 | 14 | [0.0, 2000.0] |
| BIOMD0000000815 | Chrobak2011 | Taxonomy-Mus musculus | 2 | 6 | [0.0, 280.4] |
| BIOMD0000000818 | Lee2008 | Taxonomy-Mammalia | 10 | 18 | [0.0, 12.0] |
| BIOMD0000000819 | Nazari2018 | - | 10 | 34 | [0.0, 500.0] |
| BIOMD0000000821 | Yazdjer2019 | angiogenesis | 3 | 8 | [0.0, 72.0] |
| BIOMD0000000823 | Varusai2018 | Taxonomy-Homo sapiens | 16 | 38 | [0.0, 800.0] |
| BIOMD0000000824 | Lewkiewics2019 | Taxonomy-Thymus | 1 | 7 | [0.0, 200.0] |
| BIOMD0000000825 | Greene2019 | - | 2 | 12 | [0.0, 80.0] |
| BIOMD0000000826 | Sung_Young_Shin2018 | Taxonomy-Homo sapiens | 13 | 89 | [0.0, 1956.8] |
| BIOMD0000000828 | Jung2019 | signaling | 5 | 17 | [0.0, 10.0] |
| BIOMD0000000829 | Jung2019 | signaling | 11 | 51 | [0.0, 200.8] |
| BIOMD0000000830 | GiantsosAdams 2013 | Homo sapiens | 6 | 7 | [0.0, 1089.4] |
| BIOMD0000000831 | Smith1980 | Taxonomy-Bos taurus\|\|Taxonomy-Bos taurus\|Gene Ontology-gonadotropin secretion\|Gene Ontology-luteinizing hormone secretion\|Mathematical Modelling Ontology-Ordinary differential equation model\|BioModels Database-MODEL7898438988 | 3 | 9 | [0.0, 7.2] |
| BIOMD0000000832 | Shin2016 | Taxonomy-Homo sapiens | 20 | 61 | [0.0, 100.0] |
| BIOMD0000000833 | DiCamillo2016 | \|Taxonomy-Homo sapiens\|Gene Ontology-insulin receptor signaling pathway\|Bio Models Database-MODEL1604100005 | 61 | 222 | [0.0, 10000.0] |
| BIOMD0000000835 | Rao2014 | Taxonomy-Rattus norvegicus | 31 | 202 | [0.0, 60.0] |
| BIOMD0000000837 | Hanson2016 | - | 8 | 22 | [0.0, 50000.0] |
| BIOMD0000000838 | Tsur2019 | - | 3 | 8 | [0.0, 155.2] |
| BIOMD0000000839 | Almeida2019 | circadian rhythm | 8 | 29 | [0.0, 45.0] |
| BIOMD0000000840 | Caldwell2019 | - | 5 | 10 | [0.0, 268.0] |

| Model ID | Reference | Domain | Sys. Dim. | Par. Dim. | Time Span |
|---|---|---|---|---|---|
| BIOMD0000000842 | Heitzler2012 | Taxonomy-Homo sapiens\|\|Taxonomy-Homo sapiens\|Brenda Tissue Ontology-kidney\|Gene Ontology-transmembrane receptor protein serine/threonine kinase signaling pathway\|Gene Ontology-angiotensin-activated signaling pathway\| BioModels Database-MODEL1012080000 | 20 | 35 | [0.0, 339.6] |
| BIOMD0000000843 | Dudziuk2019 | - | 10 | 39 | [0.0, 409.6] |
| BIOMD0000000844 | Viertel2019 | - | 9 | 82 | [0.0, 1400.0] |
| BIOMD0000000845 | Gulbudak2019 | - | 3 | 12 | [0.0, 250.0] |
| BIOMD0000000846 | Gulbudak2019 | - | 3 | 12 | [0.0, 400.0] |
| BIOMD0000000847 | Adams2019 | - | 4 | 17 | [0.0, 100.0] |
| BIOMD0000000848 | FatehiChenar 2018 | immune response | 9 | 28 | [0.0, 13.2] |
| BIOMD0000000849 | M_Berg2017 | Bos taurus | 13 | 72 | [0.0, 60.0] |
| BIOMD0000000850 | Jenner2019 | - | 3 | 4 | [0.0, 659.2] |
| BIOMD0000000851 | Ho2019 | - | 5 | 17 | [0.0, 35.6] |
| BIOMD0000000852 | Andersen2017 | - | 6 | 35 | [0.0, 20000.0] |
| BIOMD0000000853 | Smolen2018 | - | 9 | 31 | [0.0, 4000.0] |
| BIOMD0000000854 | Gray2016 | Taxonomy-Mus musculus | 4 | 9 | [0.0, 14.4] |
| BIOMD0000000855 | Cooper2015 | Taxonomy-Homo sapiens | 4 | 39 | [0.0, 20.0] |
| BIOMD0000000857 | Larbat2016 | sucrose metabolic process; phenol-containing compound metabolic process; starch metabolic process | 9 | 34 | [0.0, 170.0] |
| BIOMD0000000858 | Larbat2016 | phenol-containing compound metabolic process; sucrose metabolic process; starch metabolic process | 8 | 35 | [0.0, 50.0] |
| BIOMD0000000859 | Larbat2016 | starch metabolic process; phenol-containing compound metabolic process; sucrose metabolic process | 12 | 46 | [0.0, 55.0] |
| BIOMD0000000861 | Bachmann2011 | Taxonomy-Mus musculus | 26 | 32 | [0.0, 300.0] |
| BIOMD0000000862 | Proctor2017B | Homo sapiens | 8 | 11 | [0.0, 40000.0] |
| BIOMD0000000863 | Kosinsky2018 | Taxonomy-Mus musculus | 6 | 27 | [0.0, 40.2] |
| BIOMD0000000864 | Proctor2017 | Homo sapiens | 6 | 9 | [0.0, 5000.0] |
| BIOMD0000000865 | Nikolaev2019 | Taxonomy-Mus musculus | 4 | 58 | [0.0, 20.0] |
| BIOMD0000000866 | Simon2019 | Homo sapiens | 3 | 6 | [0.0, 3000.0] |
| BIOMD0000000867 | Coulibaly2019 | Taxonomy-Homo sapiens | 10 | 57 | [0.0, 70.0] |
| BIOMD0000000868 | Simon2019 | Homo sapiens | 4 | 7 | [0.0, 1990.0] |
| BIOMD0000000869 | Simon2019 | Homo sapiens | 4 | 12 | [0.0, 1448.0] |
| BIOMD0000000870 | Simon2019 | Homo sapiens | 6 | 14 | [0.0, 1943.0] |
| BIOMD0000000871 | NIK_dependent_ p100_processing_ into_p52 | Homo sapiens | 8 | 20 | [0.0, 500.0] |
| BIOMD0000000872 | Verma2016 | Homo sapiens | 7 | 24 | [0.0, 1000.0] |
| BIOMD0000000873 | Soni2018 | Leishmania | 29 | 57 | [0.0, 125.2] |
| BIOMD0000000874 | Perelson1993 | Taxonomy-Homo sapiens | 4 | 10 | [0.0, 4500.0] |
| BIOMD0000000875 | Nelson2000 | Taxonomy-Homo sapiens | 4 | 8 | [0.0, 30.0] |
| BIOMD0000000876 | Aavani2019 | Taxonomy-Homo sapiens | 4 | 11 | [0.0, 70.0] |
| BIOMD0000000877 | Ontah2019 | Taxonomy-Homo sapiens | 4 | 12 | [0.0, 70.0] |
| BIOMD0000000878 | Lenbury2001 | Taxonomy-Homo sapiens | 3 | 13 | [0.0, 420.0] |
| BIOMD0000000879 | Rodrigues2019 | Taxonomy-Homo sapiens | 3 | 18 | [0.0, 1.5] |
| BIOMD0000000880 | Trisilowati2018 | Taxonomy-Homo sapiens | 4 | 14 | [0.0, 183.0] |
| BIOMD0000000881 | Kogan2013 | - | 4 | 27 | [0.0, 3000.0] |
| BIOMD0000000882 | Munz2009 | Taxonomy-Homo sapiens | 3 | 5 | [0.0, 6.0] |
| BIOMD0000000883 | Giani2019 | Taxonomy-Homo sapiens | 63 | 134 | [0.0, 77.6] |

| Model ID | Reference | Domain | Sys. Dim. | Par. Dim. | Time Span |
|---|---|---|---|---|---|
| BIOMD0000000884 | Cortes2019 | Taxonomy-Homo sapiens | 3 | 9 | [0.0, 50.0] |
| BIOMD0000000885 | Sumana2018 | Taxonomy-Homo sapiens | 4 | 11 | [0.0, 600.0] |
| BIOMD0000000887 | Lim2014 | Taxonomy-Homo sapiens | 4 | 10 | [0.0, 549.2] |
| BIOMD0000000888 | Unni2019 | Homo sapiens | 4 | 21 | [0.0, 80.0] |
| BIOMD0000000889 | Fribourg2014 | Taxonomy-Homo sapiens | 12 | 50 | [0.0, 150.0] |
| BIOMD0000000890 | Bhattacharya 2014 | Taxonomy-Homo sapiens | 3 | 11 | [0.0, 5000.0] |
| BIOMD0000000891 | Khajanchi2019 | Taxonomy-Homo sapiens | 3 | 12 | [0.0, 430.8] |
| BIOMD0000000892 | Sandip2013 | Taxonomy-Homo sapiens | 4 | 11 | [0.0, 27.4] |
| BIOMD0000000893 | Gonzalez Miranda2013 | - | 3 | 6 | [0.0, 9.0] |
| BIOMD0000000894 | Bose2011 | Taxonomy-Homo sapiens | 3 | 8 | [0.0, 143.6] |
| BIOMD0000000895 | Schokker2013 | Taxonomy-Gallus gallus | 9 | 49 | [0.0, 120.4] |
| BIOMD0000000896 | Szymanska2009 | Taxonomy-Homo sapiens | 9 | 18 | [0.0, 1118.0] |
| BIOMD0000000897 | Khajanchi2015 | Taxonomy-Homo sapiens | 2 | 10 | [0.0, 800.0] |
| BIOMD0000000898 | Jiao2018 | Taxonomy-Homo sapiens | 5 | 22 | [0.0, 150.0] |
| BIOMD0000000899 | Ota2015 | Homo sapiens \| GDP-dissociation inhibitor binding | 12 | 15 | [0.0, 1000.0] |
| BIOMD0000000903 | perez2019 | Taxonomy-Homo sapiens | 5 | 28 | [0.0, 53.6] |
| BIOMD0000000905 | Dubey2007 | Taxonomy-Homo sapiens | 5 | 15 | [0.0, 72.0] |
| BIOMD0000000906 | Dubey2007 | Taxonomy-Homo sapiens | 3 | 9 | [0.0, 29.2] |
| BIOMD0000000907 | HeberleRazquin Navas2019 | Taxonomy-Homo sapiens | 25 | 68 | [0.0, 214.4] |
| BIOMD0000000908 | dePillis2013 | Taxonomy-Homo sapiens | 7 | 43 | [0.0, 1500.0] |
| BIOMD0000000909 | dePillis2003 | Taxonomy-Homo sapiens | 4 | 17 | [0.0, 100.0] |
| BIOMD0000000910 | Isaeva2008 | Taxonomy-Homo sapiens | 3 | 10 | [0.0, 250.0] |
| BIOMD0000000911 | Merola2008 | Taxonomy-Homo sapiens | 3 | 9 | [0.0, 137.2] |
| BIOMD0000000912 | Caravagna2010 | Taxonomy-Homo sapiens | 3 | 14 | [0.0, 500.0] |
| BIOMD0000000913 | dePillis2008 | Taxonomy-Homo sapiens | 6 | 37 | [0.0, 350.0] |
| BIOMD0000000914 | Guillen2013 | Taxonomy-Mus musculus | 5 | 10 | [0.0, 85.0] |
| BIOMD0000000915 | Sun2018 | Taxonomy-Homo sapiens | 9 | 30 | [0.0, 18000.0] |
| BIOMD0000000916 | Kraan199 | Taxonomy-Homo sapiens | 5 | 5 | [0.0, 4.4] |
| BIOMD0000000917 | Phillips2007 | Taxonomy-Homo sapiens | 3 | 21 | [0.0, 48.0] |
| BIOMD0000000919 | Ledzewicz2013 | Taxonomy-Homo sapiens\|\|Taxonomy-Homo sapiens\|Mathematical Modelling Ontology-Ordinary differential equation model | 2 | 9 | [0.0, 15.2] |
| BIOMD0000000920 | Jarrett2015 | Taxonomy-Mus musculus | 4 | 16 | [0.0, 594.0] |
| BIOMD0000000921 | Khajanchi2017 | Taxonomy-Homo sapiens | 5 | 24 | [0.0, 225.2] |
| BIOMD0000000922 | Turner2015 | Taxonomy-Lutzia\|Experimental Factor Ontology-malaria | 3 | 9 | [0.0, 22.8] |
| BIOMD0000000924 | Smith2011 | Taxonomy-Homo sapiens | 6 | 28 | [0.0, 79.8] |
| BIOMD0000000925 | Dunster2016 | Homo sapiens \| blood coagulation | 14 | 22 | [0.0, 6000.0] |
| BIOMD0000000926 | Rhodes2019 | Taxonomy-Homo sapiens | 8 | 78 | [0.0, 600.0] |
| BIOMD0000000927 | Grigolon2018 | Taxonomy-unidentified plant | 3 | 9 | [0.0, 444.4] |
| BIOMD0000000928 | Baker2017 | - | 4 | 14 | [0.0, 18.0] |
| BIOMD0000000929 | Li2016 | Taxonomy-Homo sapiens | 5 | 39 | [0.0, 500.0] |
| BIOMD0000000930 | Liu2017 | Taxonomy-Homo sapiens | 4 | 17 | [0.0, 800.0] |
| BIOMD0000000931 | Voliotis2019 | Taxonomy-Mus | 3 | 22 | [0.0, 35.0] |
| BIOMD0000000933 | Kosiuk2015 | Taxonomy-unclassified eukaryotes | 3 | 14 | [0.0, 27.0] |
| BIOMD0000000934 | Linke2017 | NCIt-Saccharomyces cerevisiae\|Gene Ontology-regulation of cell cycle | 13 | 23 | [0.0, 80.0] |
| BIOMD0000000935 | Ferrel2011 | Taxonomy-Xenopus laevis\|Gene Ontology-regulation of cell cycle | 2 | 8 | [0.0, 10.6] |

*Continued on next page*

| Model ID | Reference | Domain | Sys. Dim. | Par. Dim. | Time Span |
|---|---|---|---|---|---|
| BIOMD0000000936 | ferrel2011 | Taxonomy-Xenopus laevis\|Gene Ontology-regulation of cell cycle | 1 | 4 | [0.0, 7.6] |
| BIOMD0000000937 | Ferrel2011 | Taxonomy-Xenopus laevis\|Gene Ontology-regulation of cell cycle | 3 | 12 | [0.0, 12.0] |
| BIOMD0000000938 | Gerard2013 | Taxonomy-Mus musculus\|Gene Ontology-regulation of cell cycle | 3 | 12 | [0.0, 60.0] |
| BIOMD0000000939 | Iwamoto2010 | Taxonomy-Homo sapiens\|Gene Ontology-DNA damage response, detection of DNA damage | 54 | 138 | [0.0, 5500.0] |
| BIOMD0000000940 | Tang2019 | Taxonomy-Homo sapiens | 20 | 42 | [0.0, 1368.4] |
| BIOMD0000000941 | Gerard2010 | Taxonomy-Mammalia\|Gene Ontology-regulation of cell cycle | 8 | 22 | [0.0, 50.0] |
| BIOMD0000000942 | Sible2007 | Taxonomy-Xenopus laevis\|Gene Ontology-regulation of cell cycle | 7 | 27 | [0.0, 200.0] |
| BIOMD0000000943 | Hat2016 | Taxonomy-Homo sapiens\|Gene Ontology-cell cycle | 33 | 101 | [0.0, 150000.0] |
| BIOMD0000000944 | Goldbeter2013 | OMIT-Amphibians | 3 | 15 | [0.0, 40.0] |
| BIOMD0000000947 | Lee2017 | Taxonomy-Homo sapiens\|KEGG Drug-Acetaminophen (JP18/USP) | 9 | 12 | [0.0, 188.8] |
| BIOMD0000000948 | Landberg2009 | KEGG Compound-Resorcinol\|ChEBI-5-alkylresorcinol | 4 | 6 | [0.0, 80.0] |
| BIOMD0000000949 | Chitnis2008 | Taxonomy-Homo sapiens\|Experimental Factor Ontology-malaria | 7 | 24 | [0.0, 12000.0] |
| BIOMD0000000952 | Rodenfels2019 | Taxonomy-Danio rerio\|Gene Ontology-regulation of cell cycle | 11 | 26 | [0.0, 70.0] |
| BIOMD0000000953 | Queralt2006 | Taxonomy-Saccharomyces cerevisiae (strain ATCC 204508 / S288c) | 13 | 46 | [0.0, 500.0] |
| BIOMD0000000954 | Pandey2018 | Taxonomy-Homo sapiens | 18 | 53 | [0.0, 1536.4] |
| BIOMD0000000959 | Kok2020 | Homo sapiens \| regulation of type I interferon-mediated signaling pathway | 41 | 87 | [0.0, 50.0] |
| BIOMD0000000965 | LeBeau1999 | Homo sapiens \| calcium-mediated signaling | 5 | 17 | [0.0, 250.0] |
| BIOMD0000000966 | Cui2008 | Taxonomy-Escherichia coli | 7 | 7 | [0.0, 5.0] |
| BIOMD0000000967 | McLean1991 | Taxonomy-Human immunodeficiency virus\|Taxonomy-Homo sapiens | 4 | 10 | [0.0, 100.0] |
| BIOMD0000000968 | Palmer2008 | Taxonomy-Homo sapiens | 10 | 11 | [0.0, 100.0] |
| BIOMD0000000972 | Tang2020 | Taxonomy-Severe acute respiratory syndrome coronavirus 2\|Taxonomy-Homo sapiens | 8 | 22 | [0.0, 150.0] |
| BIOMD0000000973 | Dasgupta2020 | Taxonomy-Homo sapiens | 2 | 8 | [0.0, 100.0] |
| BIOMD0000000979 | Malkov2020 | Taxonomy-Severe acute respiratory syndrome coronavirus 2\|Taxonomy-Homo sapiens | 5 | 6 | [0.0, 350.0] |
| BIOMD0000000985 | Fabry1984 | receptor-mediated endocytosis | 7 | 13 | [0.0, 80.0] |
| BIOMD0000000986 | Aubry1995 | Taxonomy-Dictyostelium discoideum AX2 | 6 | 12 | [0.0, 293.6] |
| BIOMD0000000987 | Aubry1995 | Taxonomy-Dictyostelium discoideum AX2 | 9 | 11 | [0.0, 152.8] |
| BIOMD0000001004 | Intosalmi2015 | Taxonomy-Mus sp. | 10 | 18 | [0.0, 100.0] |
| BIOMD0000001005 | Bae2017 | Taxonomy-Mus musculus\|Taxonomy-Rattus norvegicus | 41 | 133 | [0.0, 100.0] |
| BIOMD0000001006 | Ciliberto2005 | Taxonomy-Homo sapiens | 7 | 27 | [0.0, 1250.0] |
| BIOMD0000001007 | Zhang2007 | Taxonomy-Homo sapiens | 3 | 32 | [0.0, 150.0] |
| BIOMD0000001009 | Zhang2007 | Taxonomy-Homo sapiens | 4 | 33 | [0.0, 28.8] |
| BIOMD0000001010 | Zhang2007 | Taxonomy-Homo sapiens | 2 | 26 | [0.0, 126.0] |
| BIOMD0000001011 | Triana2020 | Taxonomy-Homo sapiens | 3 | 5 | [0.0, 550.0] |
| BIOMD0000001012 | Triana2020 | Taxonomy-Homo sapiens | 4 | 10 | [0.0, 1000.0] |
| BIOMD0000001015 | Jarrah2014 | Taxonomy-Mus musculus | 6 | 20 | [0.0, 200.0] |

*Continued on next page*

| Model ID | Reference | Domain | Sys. Dim. | Par. Dim. | Time Span |
|---|---|---|---|---|---|
| BIOMD0000001016 | Bakshi2020 | Taxonomy-Homo sapiens | 8 | 14 | [0.0, 100.0] |
| BIOMD0000001017 | Bakshi2020 | Taxonomy-Homo sapiens | 13 | 22 | [0.0, 20000.0] |
| BIOMD0000001018 | Bakshi2020 | Taxonomy-Homo sapiens | 18 | 31 | [0.0, 20000.0] |
| BIOMD0000001021 | Lavigne2021 | Taxonomy-Homo sapiens | 6 | 10 | [0.0, 8.8] |
| BIOMD0000001022 | Creemers2021 | Taxonomy-Homo sapiens | 4 | 7 | [0.0, 80.0] |
| BIOMD0000001023 | Alharbi2020 | Taxonomy-Homo sapiens | 3 | 16 | [0.0, 38.8] |
| BIOMD0000001024 | Chaudhury2020 | Taxonomy-Homo sapiens | 2 | 4 | [0.0, 240.0] |
| BIOMD0000001025 | Chaudhury2020 | Taxonomy-Homo sapiens\|Human Disease Ontology-leukemia | 3 | 14 | [0.0, 1000.0] |
| BIOMD0000001026 | Kurlovics2021 | Taxonomy-Homo sapiens\|\|Brenda Tissue Ontology-erythrocyte\|ChEBI-metformin\| Gene Ontology-drug transport\| Mathematical Modelling Ontology-Ordinary differential equation model\|Bio Models Database-MODEL2103170002 | 1 | 12 | [0.0, 200.0] |
| BIOMD0000001027 | Zake2021 | Taxonomy-Mus sp. | 20 | 97 | [0.0, 15.0] |
| BIOMD0000001028 | Zake2021 | Taxonomy-Homo sapiens | 21 | 104 | [0.0, 150.0] |
| BIOMD0000001029 | Zake2021 | Taxonomy-Homo sapiens | 21 | 104 | [0.0, 80.0] |
| BIOMD0000001030 | Sontag2017 | Taxonomy-Vertebrata | 2 | 6 | [0.0, 200.0] |
| BIOMD0000001031 | Tuwairqi2020 | Taxonomy-Homo sapiens | 3 | 4 | [0.0, 30.0] |
| BIOMD0000001032 | Tuwairqi2020 | Taxonomy-Homo sapiens | 4 | 7 | [0.0, 49.6] |
| BIOMD0000001033 | Almuallem2020 | Taxonomy-Homo sapiens | 6 | 31 | [0.0, 113.8] |
| BIOMD0000001034 | Mendrazitsky 2007 | Taxonomy-Mycobacterium tuberculosis variant bovis BCG\|Taxonomy-Homo sapiens | 4 | 12 | [0.0, 50.0] |
| BIOMD0000001035 | Tuwairqi2020 | Taxonomy-Homo sapiens | 4 | 11 | [0.0, 259.6] |
| BIOMD0000001036 | Cappuccio2007 | Taxonomy-Homo sapiens | 3 | 13 | [0.0, 1500.0] |
| BIOMD0000001037 | Alharbi2019 | Taxonomy-Homo sapiens | 2 | 6 | [0.0, 10.0] |
| BIOMD0000001038 | Alharbi2019 | Taxonomy-Homo sapiens | 3 | 10 | [0.0, 100.0] |
| BIOMD0000001039 | Zake2021 | Taxonomy-Mus sp. | 20 | 97 | [0.0, 6.0] |
| BIOMD0000001040 | Kurlovics2021 | Taxonomy-Homo sapiens\|\|Brenda Tissue Ontology-erythrocyte\|ChEBI-metformin\| Gene Ontology-drug transport\| Mathematical Modelling Ontology-Ordinary differential equation model\|Bio Models Database-MODEL2103170001 | 1 | 11 | [0.0, 120.0] |
| BIOMD0000001043 | Wodarz2001 | Taxonomy-Homo sapiens | 3 | 9 | [0.0, 1500.0] |
| BIOMD0000001045 | Moore2004 | Taxonomy-Influenza A virus (strain A/ Hong Kong/1/1968 H3N2)\|Taxonomy-Homo sapiens | 3 | 2 | [0.0, 130.0] |
| BIOMD0000001047 | Collier1996 | Taxonomy-Drosophila | 4 | 14 | [0.0, 24.4] |
| BIOMD0000001048 | Siddhartha2002 | Taxonomy-Homo sapiens | 3 | 11 | [0.0, 1994.4] |
| BIOMD0000001052 | Alharbi2020 | Taxonomy-Homo sapiens | 3 | 16 | [0.0, 30.0] |
| BIOMD0000001053 | Garde2020 | Taxonomy-Bacillus subtilis | 6 | 6 | [0.0, 5.5] |
| BIOMD0000001054 | Pearce2021 | Taxonomy-Homo sapiens | 7 | 11 | [0.0, 200.0] |
| BIOMD0000001056 | Chulian2021 | Taxonomy-Homo sapiens | 3 | 9 | [0.0, 4000.0] |
| BIOMD0000001057 | Nikolov2020 | Taxonomy-Homo sapiens | 2 | 7 | [0.0, 1298.0] |
| BIOMD0000001058 | Novak2022 | Taxonomy-Homo sapiens | 10 | 48 | [0.0, 212.0] |
| BIOMD0000001059 | Stucki2005 | Taxonomy-Mus | 10 | 23 | [0.0, 1062.6] |
| BIOMD0000001060 | Frank2021 | Taxonomy-Homo sapiens | 2 | 20 | [0.0, 24.4] |
| BIOMD0000001072 | Phillips2013 | Taxonomy-Mammalia | 6 | 53 | [0.0, 270000.0] |
| BIOMD0000001077 | Adlung2021 | Taxonomy-Homo sapiens | 15 | 29 | [0.0, 130.0] |
| BIOMD0000001102 | Burbano2023 | - | 23 | 28 | [0.0, 62.0] |
| MODEL0910896131 | Guyton1972 | Taxonomy-Homo sapiens | 2 | 8 | [0.0, 60000.0] |
| MODEL0975191032 | Chang2008 | Taxonomy-Homo sapiens | 115 | 219 | [0.0, 1000.0] |

| Model ID | Reference | Domain | Sys. Dim. | Par. Dim. | Time Span |
|---|---|---|---|---|---|
| MODEL1002160000 | Cabrero2011 | Taxonomy-Homo sapiens | 8 | 64 | [0.0, 100.0] |
| MODEL1004010000 | Kuwahara2010 | Taxonomy-Escherichia coli | 31 | 55 | [0.0, 2.0] |
| MODEL1004010001 | Kuwahara2010 | Taxonomy-Escherichia coli | 31 | 55 | [0.0, 3.0] |
| MODEL1004010002 | Kuwahara2010 | Taxonomy-Escherichia coli | 31 | 55 | [0.0, 0.15] |
| MODEL1004070000 | Haut1974 | Rattus \| pentose-phosphate shunt | 31 | 49 | [0.0, 750.0] |
| MODEL1004070001 | Vaseghi1999 | Saccharomyces cerevisiae \| pentose-phosphate shunt | 6 | 18 | [0.0, 1986.0] |
| MODEL1005050000 | Salazar2009 | Taxonomy-Arabidopsis thaliana | 15 | 76 | [0.0, 60.0] |
| MODEL1005200000 | Twycross2010 | Taxonomy-Embryophyta | 43 | 16 | [0.0, 1000.0] |
| MODEL1007200000 | Nijhout2006 | Taxonomy-Homo sapiens | 23 | 108 | [0.0, 5.0] |
| MODEL1008060000 | Munz2009 | Taxonomy-Homo sapiens | 3 | 9 | [0.0, 107.6] |
| MODEL1008060002 | Munz2009 | Taxonomy-Homo sapiens | 5 | 10 | [0.0, 400.0] |
| MODEL1009220000 | Martins2004 | Saccharomyces cerevisiae | 19 | 110 | [0.0, 290.0] |
| MODEL1009230000 | Munz2009 | Taxonomy-Homo sapiens | 3 | 6 | [0.0, 1985.2] |
| MODEL1011010000 | Bruck2008 | Saccharomyces cerevisiae \| glycolytic process | 17 | 93 | [0.0, 0.4] |
| MODEL1101100000 | Bakker1997 | Taxonomy-Trypanosoma brucei | 38 | 50 | [0.0, 55.0] |
| MODEL1101170000 | Nakano2010 | Taxonomy-Homo sapiens | 182 | 286 | [0.0, 60.0] |
| MODEL1102210000 | Wei2011 | Taxonomy-Homo sapiens | 190 | 316 | [0.0, 84.8] |
| MODEL1102210001 | Telesco2011 | Taxonomy-Homo sapiens | 119 | 58 | [0.0, 1712.0] |
| MODEL1103210001 | Jamshidi01_RBC_Metabolic Network | Homo sapiens | 39 | 410 | [0.0, 150.0] |
| MODEL1108260010 | Jesty1993 | Taxonomy-Homo sapiens | 4 | 8 | [0.0, 12.0] |
| MODEL1108260015 | Qiao2004 | Taxonomy-Homo sapiens | 6 | 20 | [0.0, 503.2] |
| MODEL1109160000 | Kogan2001 | - | 30 | 32 | [0.0, 33.2] |
| MODEL1109160001 | Kogan2001 | - | 40 | 37 | [0.0, 11.8] |
| MODEL1112100000 | nsson2005 | Taxonomy-Arabidopsis thaliana | 1012 | 522 | [0.0, 2000.0] |
| MODEL1112110004 | Silber2007 | Taxonomy-Homo sapiens | 7 | 22 | [0.0, 1980.0] |
| MODEL1112150000 | Tiemann2011 | Taxonomy-Mus musculus | 8 | 23 | [0.0, 6.0] |
| MODEL1112260002 | Smith2010 | Taxonomy-Homo sapiens | 54 | 135 | [0.0, 300.0] |
| MODEL1202030000 | Dupeux2011 | Taxonomy-Arabidopsis thaliana | 14 | 24 | [0.0, 10.0] |
| MODEL1202030001 | Dupeux2011 | Taxonomy-Arabidopsis thaliana | 8 | 12 | [0.0, 10.0] |
| MODEL1202170000 | Nazaret2008 | Taxonomy-Eukaryota | 12 | 49 | [0.0, 94.8] |
| MODEL1203220000 | Mol2013 | Taxonomy-Leishmania | 64 | 131 | [0.0, 200.0] |
| MODEL1204040000 | Houser2012 | Taxonomy-Saccharomyces cerevisiae | 15 | 46 | [0.0, 8000.0] |
| MODEL1204060000 | Kubota2012 | Taxonomy-Rattus | 23 | 26 | [0.0, 350.0] |
| MODEL1204240000 | Sorokina2011 | Taxonomy-Ostreococcus tauri | 38 | 73 | [0.0, 236.2] |
| MODEL1204280001 | Sarma2012 | Taxonomy-Xenopus laevis | 12 | 23 | [0.0, 500.0] |
| MODEL1204280002 | Sarma2012 | Taxonomy-Mus musculus | 11 | 22 | [0.0, 350.0] |
| MODEL1204280003 | Sarma2012 | Taxonomy-Mus musculus | 11 | 22 | [0.0, 500.0] |
| MODEL1204280004 | Sarma2012 | Taxonomy-Mus musculus | 11 | 26 | [0.0, 600.0] |
| MODEL1204280005 | Sarma2012 | Taxonomy-Xenopus laevis | 12 | 23 | [0.0, 266.8] |
| MODEL1204280006 | Sarma2012 | Taxonomy-Mus musculus | 11 | 22 | [0.0, 400.0] |
| MODEL1204280007 | Sarma2012 | Taxonomy-Mus musculus | 11 | 22 | [0.0, 300.0] |
| MODEL1204280008 | Sarma2012 | Taxonomy-Mus musculus | 11 | 26 | [0.0, 294.8] |
| MODEL1204280009 | Sarma2012 | Taxonomy-Xenopus laevis | 12 | 23 | [0.0, 500.0] |
| MODEL1204280010 | Sarma2012 | Taxonomy-Mus musculus | 11 | 22 | [0.0, 350.0] |
| MODEL1204280011 | Sarma2012 | Taxonomy-Mus musculus | 11 | 22 | [0.0, 500.0] |
| MODEL1204280012 | Sarma2012 | Taxonomy-Mus musculus | 11 | 26 | [0.0, 600.0] |
| MODEL1204280013 | Sarma2012 | Taxonomy-Xenopus laevis | 12 | 23 | [0.0, 400.0] |
| MODEL1204280014 | Sarma2012 | Taxonomy-Mus musculus | 11 | 22 | [0.0, 350.0] |
| MODEL1204280015 | Sarma2012 | Taxonomy-Mus musculus | 11 | 22 | [0.0, 280.0] |
| MODEL1204280016 | Sarma2012 | Taxonomy-Mus musculus | 11 | 26 | [0.0, 350.0] |

*Continued on next page*

| Model ID | Reference | Domain | Sys. Dim. | Par. Dim. | Time Span |
|----------|-----------|--------|-----------|-----------|-----------|
| MODEL1204280017 | Sarma2012 | Taxonomy-Xenopus laevis | 25 | 37 | [0.0, 1000.0] |
| MODEL1204280018 | Sarma2012 | Taxonomy-Mus musculus | 24 | 37 | [0.0, 1000.0] |
| MODEL1204280019 | Sarma2012 | Taxonomy-Mus musculus | 24 | 37 | [0.0, 1503.0] |
| MODEL1204280021 | Sarma2012 | Taxonomy-Xenopus laevis | 25 | 37 | [0.0, 1541.2] |
| MODEL1204280022 | Sarma2012 | Taxonomy-Mus musculus | 24 | 37 | [0.0, 1000.0] |
| MODEL1204280023 | Sarma2012 | Taxonomy-Mus musculus | 24 | 37 | [0.0, 1572.4] |
| MODEL1204280025 | Sarma2012 | Taxonomy-Xenopus laevis | 25 | 37 | [0.0, 1200.0] |
| MODEL1204280029 | Sarma2012 | Taxonomy-Xenopus laevis | 25 | 37 | [0.0, 1200.0] |
| MODEL1204280030 | Sarma2012 | Taxonomy-Mus musculus | 24 | 37 | [0.0, 5000.0] |
| MODEL1204280031 | Sarma2012 | Taxonomy-Mus musculus | 24 | 37 | [0.0, 5000.0] |
| MODEL1204280032 | Sarma2012 | Taxonomy-Mus musculus | 27 | 45 | [0.0, 5000.0] |
| MODEL1204280033 | Sarma2012 | Taxonomy-Xenopus laevis | 12 | 23 | [0.0, 700.0] |
| MODEL1204280034 | Sarma2012 | Taxonomy-Mus musculus | 11 | 22 | [0.0, 421.6] |
| MODEL1204280035 | Sarma2012 | Taxonomy-Mus musculus | 11 | 22 | [0.0, 559.6] |
| MODEL1204280037 | Sarma2012 | Taxonomy-Xenopus laevis | 12 | 23 | [0.0, 201.2] |
| MODEL1204280038 | Sarma2012 | Taxonomy-Mus musculus | 11 | 22 | [0.0, 500.0] |
| MODEL1204280039 | Sarma2012 | Taxonomy-Mus musculus | 11 | 22 | [0.0, 346.4] |
| MODEL1208030000 | Mandlik2013 | Taxonomy-Leishmania | 6 | 27 | [0.0, 30.0] |
| MODEL1303010000 | Lopez2013 | Taxonomy-Homo sapiens | 74 | 106 | [0.0, 10000.0] |
| MODEL1303140000 | Mazemondet 2012 | Taxonomy-Homo sapiens | 5 | 12 | [0.0, 302.0] |
| MODEL1303260000 | Smallbone2013 | Saccharomyces cerevisiae \| glycolytic process | 15 | 93 | [0.0, 154.4] |
| MODEL1303260001 | Smallbone2013 | Saccharomyces cerevisiae \| glycolytic process | 15 | 95 | [0.0, 154.0] |
| MODEL1303260002 | Smallbone2013 | Saccharomyces cerevisiae \| glycolytic process | 15 | 105 | [0.0, 182.0] |
| MODEL1303260003 | Smallbone2013 | Saccharomyces cerevisiae \| glycolytic process | 15 | 106 | [0.0, 100.0] |
| MODEL1303260004 | Smallbone2013 | Saccharomyces cerevisiae \| glycolytic process | 16 | 111 | [0.0, 97.2] |
| MODEL1303260005 | Smallbone2013 | Saccharomyces cerevisiae \| glycolytic process | 16 | 111 | [0.0, 103.8] |
| MODEL1303260006 | Smallbone2013 | Saccharomyces cerevisiae \| glycolytic process | 20 | 128 | [0.0, 156.8] |
| MODEL1303260007 | Smallbone2013 | Saccharomyces cerevisiae \| glycolytic process | 20 | 129 | [0.0, 109.0] |
| MODEL1303260008 | Smallbone2013 | Saccharomyces cerevisiae \| glycolytic process | 20 | 132 | [0.0, 472.0] |
| MODEL1303260009 | Smallbone2013 | Saccharomyces cerevisiae \| glycolytic process | 20 | 142 | [0.0, 518.4] |
| MODEL1303260010 | Smallbone2013 | Saccharomyces cerevisiae \| glycolytic process | 20 | 142 | [0.0, 465.6] |
| MODEL1303260011 | Smallbone2013 | Saccharomyces cerevisiae \| glycolytic process | 20 | 142 | [0.0, 300.0] |
| MODEL1303260012 | Smallbone2013 | Saccharomyces cerevisiae \| glycolytic process | 20 | 145 | [0.0, 200.0] |
| MODEL1303260013 | Smallbone2013 | Saccharomyces cerevisiae \| glycolytic process | 20 | 146 | [0.0, 607.6] |
| MODEL1303260014 | Smallbone2013 | Saccharomyces cerevisiae \| glycolytic process | 20 | 155 | [0.0, 573.6] |
| MODEL1303260015 | Smallbone2013 | Saccharomyces cerevisiae \| glycolytic process | 20 | 155 | [0.0, 575.4] |
| MODEL1303260016 | Smallbone2013 | Saccharomyces cerevisiae \| glycolytic process | 20 | 166 | [0.0, 526.6] |

| Model ID | Reference | Domain | Sys. Dim. | Par. Dim. | Time Span |
|---|---|---|---|---|---|
| MODEL1303260017 | Smallbone2013 | Saccharomyces cerevisiae \| glycolytic process | 20 | 168 | [0.0, 546.2] |
| MODEL1303260018 | Smallbone2013 | Saccharomyces cerevisiae \| glycolytic process | 20 | 168 | [0.0, 305.0] |
| MODEL1304300000 | Hofmeyr1996 | regulation of catalytic activity | 3 | 8 | [0.0, 2.0] |
| MODEL1308080003 | Cao2010 | Taxonomy-Escherichia coli | 14 | 30 | [0.0, 100000.0] |
| MODEL1403250001 | vanEunen2012 | Taxonomy-Saccharomyces cerevisiae | 13 | 88 | [0.0, 2.0] |
| MODEL1403250002 | Rao2014 | Taxonomy-Saccharomyces cerevisiae | 8 | 50 | [0.0, 3.6] |
| MODEL1409050001 | Rutkis2013 | - | 20 | 87 | [0.0, 6500.0] |
| MODEL1409240002 | Qosa2014 | Taxonomy-Mus musculus\|Taxonomy-Homo sapiens | 6 | 14 | [0.0, 46.2] |
| MODEL1411130000 | Miao2014 | - | 8 | 12 | [0.0, 10.0] |
| MODEL1412200000 | Talemi2015 | - | 5 | 19 | [0.0, 284.4] |
| MODEL1502270000 | Weisse2015 | - | 22 | 32 | [0.0, 0.15] |
| MODEL1503180002 | Smallbone2015 | - | 4 | 7 | [0.0, 10000.0] |
| MODEL1503180004 | Smallbone2015 | - | 3 | 11 | [0.0, 15.0] |
| MODEL1503180005 | Smallbone2015 | - | 3 | 9 | [0.0, 100.0] |
| MODEL1503180006 | Smallbone2015 | - | 3 | 12 | [0.0, 66.0] |
| MODEL1504010000 | Shestov2014 | - | 21 | 144 | [0.0, 4.4] |
| MODEL1504080004 | Costa2015 | - | 18 | 145 | [0.0, 200.0] |
| MODEL1504080005 | Costa2015 | - | 18 | 140 | [0.0, 200.0] |
| MODEL1504080006 | Costa2015 | - | 18 | 149 | [0.0, 200.0] |
| MODEL1504160000 | Wu2011 | - | 6 | 11 | [0.0, 30.0] |
| MODEL1505110000 | Millard2016 | - | 64 | 440 | [0.0, 87.8] |
| MODEL1506070001 | Nguyen2014 | - | 21 | 60 | [0.0, 1250.0] |
| MODEL1509020000 | Ashall2009 | - | 14 | 38 | [0.0, 50000.0] |
| MODEL1509050002 | Perrett2014 | - | 11 | 39 | [0.0, 275.2] |
| MODEL1510230001 | Rantasalo2016 | - | 13 | 33 | [0.0, 3.2] |
| MODEL1510230002 | Rantasalo2016 | - | 13 | 33 | [0.0, 0.7] |
| MODEL1510230003 | Rantasalo2016 | - | 13 | 33 | [0.0, 2.8] |
| MODEL1510230004 | Rantasalo2016 | - | 18 | 37 | [0.0, 0.3] |
| MODEL1510230005 | Rantasalo2016 | - | 18 | 37 | [0.0, 0.06] |
| MODEL1601250000 | Tortolina2015 | - | 456 | 862 | [0.0, 264.6] |
| MODEL1603150000 | Saa2016 | - | 5 | 40 | [0.0, 1000.0] |
| MODEL1603270000 | D1 LTP time window | - | 87 | 143 | [0.0, 20.0] |
| MODEL1604100000 | Mukhopadhyay 2013 | - | 53 | 22 | [0.0, 0.4] |
| MODEL1608100000 | Aguilera2017 | - | 2 | 4 | [0.0, 267.2] |
| MODEL1608100001 | Aguilera2017 | - | 7 | 18 | [0.0, 1500.0] |
| MODEL1609100001 | Adams2013 | - | 6 | 7 | [0.0, 1000.0] |
| MODEL1610100003 | Proctor2017 | - | 7 | 9 | [0.0, 10.0] |
| MODEL1611030000 | Guisoni2016 | - | 5 | 5 | [0.0, 27.2] |
| MODEL1611160001 | Leber2016 | - | 42 | 117 | [0.0, 43.4] |
| MODEL1611160002 | Leber2016 | - | 37 | 76 | [0.0, 860.8] |
| MODEL1701170000 | Yapo2017 | - | 52 | 115 | [0.0, 800.0] |
| MODEL1701170001 | Yapo2017 | - | 64 | 152 | [0.0, 500.0] |
| MODEL1704110000 | Proctor2017 | - | 9 | 17 | [0.0, 500000.0] |
| MODEL1704110001 | Proctor2017 | - | 16 | 31 | [0.0, 1879.2] |
| MODEL1704110002 | Proctor2017 | - | 11 | 17 | [0.0, 500000.0] |
| MODEL1704110003 | Proctor2017 | - | 14 | 21 | [0.0, 2000000.0] |
| MODEL1704110004 | Proctor2017 | - | 46 | 84 | [0.0, 1917.8] |

| Model ID | Reference | Domain | Sys. Dim. | Par. Dim. | Time Span |
|---|---|---|---|---|---|
| MODEL1704190000 | Phosphatase_activities_on_PI_3_4_5_P3_and_PI_3_4_P2 | - | 12 | 31 | [0.0, 1187.6] |
| MODEL1705030000 | DallePezze2016 | - | 29 | 50 | [0.0, 30.0] |
| MODEL1705030001 | DallePezze2016 | - | 29 | 53 | [0.0, 48.8] |
| MODEL1705170000 | Proctor2017 | - | 10 | 15 | [0.0, 1989.4] |
| MODEL1705170001 | Proctor2017 | - | 5 | 8 | [0.0, 5000.0] |
| MODEL1705170002 | Proctor2017 | - | 7 | 11 | [0.0, 10000.0] |
| MODEL1705170003 | Proctor2017 | - | 12 | 18 | [0.0, 1988.6] |
| MODEL1705170004 | Proctor2017 | - | 17 | 25 | [0.0, 10000.0] |
| MODEL1705170005 | Proctor2017 | - | 50 | 83 | [0.0, 1981.8] |
| MODEL1708210000 | PanRTK_model_for_single_cell_line | Taxonomy-Homo sapiens | 59 | 124 | [0.0, 600.0] |
| MODEL1808280013 | Ducrot2009 | - | 10 | 35 | [0.0, 1990.6] |
| MODEL1812050001 | Rohan_D_Gidvani2012 | - | 13 | 24 | [0.0, 1987.2] |
| MODEL1907260001 | dePillis2005 | Taxonomy-Mus musculus | 3 | 23 | [0.0, 52.8] |
| MODEL1907310002 | Arciero2004 | Taxonomy-Homo sapiens | 4 | 20 | [0.0, 255.2] |
| MODEL1908270001 | Nikolopoulou 2018 | - | 3 | 31 | [0.0, 210.8] |
| MODEL1908270002 | ODea2007 | - | 24 | 70 | [0.0, 1531.8] |
| MODEL1908290001 | Hetmanski2019 | - | 58 | 97 | [0.0, 1978.8] |
| MODEL1909050002 | Ribba2018 | - | 2 | 13 | [0.0, 300.0] |
| MODEL1909090002 | Wei2019 | - | 5 | 27 | [0.0, 17.6] |
| MODEL1909090003 | Shariatpanahi 2018 | - | 4 | 29 | [0.0, 195.4] |
| MODEL1909100002 | Louzoun2014 | - | 4 | 24 | [0.0, 494.6] |
| MODEL1909160002 | Reyes2017 | Taxonomy-Mus musculus | 8 | 29 | [0.0, 3000.0] |
| MODEL1910240001 | Petrov2018 | - | 5 | 13 | [0.0, 8000.0] |
| MODEL1911120004 | Gupta2019 | - | 4 | 26 | [0.0, 500.0] |
| MODEL1911130007 | Hutchinson2016 | Taxonomy-Homo sapiens | 16 | 49 | [0.0, 828.4] |
| MODEL1911140002 | Tiwari2019 | Taxonomy-Mus musculus | 17 | 38 | [0.0, 6500.0] |
| MODEL1911150001 | Tsur2019 | - | 3 | 10 | [0.0, 1896.4] |
| MODEL1911190001 | Day2015 | immune response | 6 | 39 | [0.0, 488.8] |
| MODEL1911270001 | Sigal2019 | Taxonomy-Mus musculus | 7 | 35 | [0.0, 500.0] |
| MODEL1911270002 | Yu2019 | Taxonomy-Mus musculus | 4 | 17 | [0.0, 46.2] |
| MODEL1912090002 | Kawka2014 | Wnt signaling pathway | 9 | 19 | [0.0, 20000.0] |
| MODEL1912100004 | Nguyen2013 | Taxonomy-Homo sapiens | 22 | 49 | [0.0, 10000.0] |
| MODEL1912100005 | Nguyen2013 | Taxonomy-Homo sapiens | 7 | 19 | [0.0, 50.4] |
| MODEL1912120002 | Sandip2013 | Taxonomy-Homo sapiens | 4 | 11 | [0.0, 37.2] |
| MODEL1912120005 | Mkango2019 | Taxonomy-Homo sapiens | 6 | 29 | [0.0, 35.0] |
| MODEL2001080001 | Guimera2017 | cell redox homeostasis; mitochondrion | 15 | 9 | [0.0, 803.2] |
| MODEL2001090001 | Leeuwen2007 | Taxonomy-Homo sapiens | 11 | 29 | [0.0, 5.0] |
| MODEL2001090002 | Nanda2013 | Homo sapiens | 5 | 28 | [0.0, 1000.0] |
| MODEL2001130003 | Kronik2010 | Taxonomy-Homo sapiens | 7 | 15 | [0.0, 600.0] |
| MODEL2001160001 | dePillis2007 | Taxonomy-Homo sapiens | 4 | 15 | [0.0, 288.4] |
| MODEL2003030002 | Back2018 | - | 5 | 12 | [0.0, 100.0] |
| MODEL2003060002 | Grigolon2018 | Taxonomy-unidentified plant | 3 | 9 | [0.0, 50.0] |
| MODEL2003170002 | Vibert2017 | - | 6 | 14 | [0.0, 1.4] |
| MODEL2003180001 | Gidvani2012 | Taxonomy-Saccharomyces cerevisiae | 13 | 24 | [0.0, 1.0] |
| MODEL2003190008 | Garira2019 | - | 6 | 65 | [0.0, 1901.0] |
| MODEL2003200001 | Jafarnejad2019 | Taxonomy-Homo sapiens | 49 | 71 | [0.0, 0.6] |
| MODEL2004300002 | Konrath2020 | - | 7 | 41 | [0.0, 10.0] |

| Model ID | Reference | Domain | Sys. Dim. | Par. Dim. | Time Span |
|---|---|---|---|---|---|
| MODEL2005070001 | Hatzimanikatis 1999 | cell cycle | 3 | 19 | [0.0, 0.002] |
| MODEL2005130001 | Ma2005 | Homo sapiens | 3 | 36 | [0.0, 8.0] |
| MODEL2007090001 | Nwabugwu2013 | Taxonomy-Homo sapiens\|NCIt-Tumor-Infiltrating Immune Cell | 16 | 48 | [0.0, 14000.0] |
| MODEL2009230001 | Hardiansyah2019 | Taxonomy-Homo sapiens | 8 | 29 | [0.0, 5.0] |
| MODEL2011030003 | Bouchnita2020 | - | 6 | 11 | [0.0, 250.0] |
| MODEL2012040002 | Khandibharad 2022 | - | 63 | 83 | [0.0, 384.0] |
| MODEL2103010001 | Hetmanski2021 | - | 58 | 99 | [0.0, 1856.0] |
| MODEL2103290001 | Sharma2022 | - | 8 | 88 | [0.0, 5000.0] |
| MODEL2109110006 | Mullinax2016 | Taxonomy-Homo sapiens | 4 | 17 | [0.0, 101.4] |
| MODEL2112290001 | Berzins2022 | BioModels Database-BIOMD0000000222 | 30 | 229 | [0.0, 13.0] |
| MODEL2201210001 | Sivery2016 | - | 7 | 21 | [0.0, 12.8] |
| MODEL2201250001 | Potassium_balance_2021 | - | 15 | 72 | [0.0, 60.0] |
| MODEL2202020001 | Slaviero2021 | - | 27 | 196 | [0.0, 40.0] |
| MODEL2206230001 | Mosbacher2022 | - | 98 | 280 | [0.0, 10.0] |
| MODEL2209260001 | Grignard2022 | - | 6 | 12 | [0.0, 37.6] |
| MODEL2210070001 | Maier2022 | - | 44 | 63 | [0.0, 2500.0] |
| MODEL2301180001 | Li2021 | - | 4 | 11 | [0.0, 16.0] |
| MODEL2306170002 | Harish2009 | - | 38 | 71 | [0.0, 35.0] |
| MODEL2306300001 | Davies2023 | Taxonomy-Homo sapiens | 29 | 108 | [0.0, 433.6] |
| MODEL2307050001 | Sobotta2017 | - | 35 | 74 | [0.0, 1995.2] |
| MODEL2307110001 | Mothes2020 | - | 7 | 17 | [0.0, 502.4] |
| MODEL2307130001 | Konrath2023 | - | 20 | 29 | [0.0, 1990.0] |
| MODEL2311140001 | DeboraSoncini 2024 | - | 18 | 157 | [0.0, 1971.2] |
| MODEL2401050001 | Lang2024 | - | 124 | 325 | [0.0, 2000.0] |
| MODEL2426780967 | MODEL2426 | Taxonomy-Saccharomyces cerevisiae | 17 | 88 | [0.0, 5.2] |
| MODEL2427021978 | MODEL2427 | Taxonomy-Saccharomyces cerevisiae | 17 | 88 | [0.0, 5.2] |
| MODEL2502170001 | Hayashi2013 | Taxonomy-Mus musculus | 51 | 67 | [0.0, 1263.6] |
| MODEL2502170002 | Hayashi2013 | Taxonomy-Mus musculus | 26 | 42 | [0.0, 1643.2] |
| MODEL2503310002 | Kurpad2023 | Taxonomy-Homo sapiens | 3 | 13 | [0.0, 20000.0] |
| MODEL2937159804 | Shimoni2009 | Taxonomy-Escherichia coli | 6 | 14 | [0.0, 640.0] |
| MODEL3631586579 | MODEL3631 | Taxonomy-Embryophyta | 436 | 1661 | [0.0, 0.4] |
| MODEL4780784080 | Hayashi1999 | Taxonomy-Rattus | 10 | 16 | [0.0, 22.0] |
| MODEL4816599063 | mahaney2000 | Taxonomy-Spodoptera frugiperda | 10 | 17 | [0.0, 25.0] |
| MODEL4821294342 | Kierzek2001 | Taxonomy-Escherichia coli | 8 | 18 | [0.0, 269.6] |
| MODEL4992089662 | Banaji2005 | Homo sapiens \| blood circulation | 36 | 106 | [0.0, 220.2] |
| MODEL5954483266 | Koster1988 | Taxonomy-Xenopus laevis | 2 | 12 | [0.0, 600.0] |
| MODEL6185511733 | Qiao2007 | Taxonomy-Eukaryota | 9 | 12 | [0.0, 0.6] |
| MODEL6185746832 | Qiao2007 | Taxonomy-Eukaryota | 14 | 18 | [0.0, 500.0] |
| MODEL6623617994 | Lambeth2002 | Taxonomy-Homo sapiens | 20 | 85 | [0.0, 11.6] |
| MODEL6624091635 | MODEL6624 | Taxonomy-Lactococcus lactis | 23 | 151 | [0.0, 0.5] |
| MODEL6624199343 | Martins2001 | Taxonomy-Saccharomyces cerevisiae | 3 | 13 | [0.0, 50000.0] |
| MODEL6655501972 | Nijhout2004 | - | 6 | 43 | [0.0, 10000.0] |
| MODEL7743212613 | Radulescu2008 | Taxonomy-Mammalia | 5 | 41 | [0.0, 15000.0] |
| MODEL7743315447 | Radulescu2008 | Taxonomy-Mammalia | 6 | 40 | [0.0, 30000.0] |
| MODEL7743358405 | Radulescu2008 | Taxonomy-Mammalia | 8 | 37 | [0.0, 32000.0] |
| MODEL7743444866 | Radulescu2008 | Taxonomy-Mammalia | 18 | 100 | [0.0, 1016.0] |
| MODEL7743528808 | Radulescu2008 | - | 19 | 100 | [0.0, 1972.8] |
| MODEL7743576806 | Radulescu2008 | - | 21 | 100 | [0.0, 60000.0] |
| MODEL7743608569 | Radulescu2008 | Taxonomy-Mammalia | 30 | 95 | [0.0, 60000.0] |

*Continued on next page*

| Model ID | Reference | Domain | Sys. Dim. | Par. Dim. | Time Span |
|---|---|---|---|---|---|
| MODEL7743631122 | Radulescu2008 | Taxonomy-Mammalia | 39 | 90 | [0.0, 60000.0] |
| MODEL8236441887 | Yang2008 | Taxonomy-Homo sapiens | 19 | 73 | [0.0, 78.0] |
| MODEL8236480549 | Yang2008 | Taxonomy-Homo sapiens | 20 | 67 | [0.0, 20000.0] |
| MODEL8236520494 | Yang2008 | Taxonomy-Homo sapiens | 7 | 24 | [0.0, 10.0] |
| MODEL8262229752 | MODEL8262 | Taxonomy-Escherichia coli | 21 | 25 | [0.0, 700.4] |
| MODEL8459127548 | You2010 | Taxonomy-Saccharomyces cerevisiae | 30 | 37 | [0.0, 1946.0] |
| MODEL8478881246 | Basak_Cell_2007 | Taxonomy-Homo sapiens | 32 | 101 | [0.0, 1720.8] |
| MODEL8938094216 | dAlcantara2003 | Taxonomy-Homo sapiens | 14 | 16 | [0.0, 10.0] |
| MODEL9070467164 | Bhalla2002 | Taxonomy-Mus musculus | 86 | 169 | [0.0, 1610.2] |
| MODEL9071122126 | Bhalla1999 | Taxonomy-Homo sapiens | 59 | 108 | [0.0, 513.2] |
| MODEL9071773985 | Bhalla2001 | Taxonomy-Homo sapiens | 66 | 139 | [0.0, 295.8] |
| MODEL9077438479 | Bhalla2002 | - | 26 | 43 | [0.0, 136.8] |
| MODEL9079179924 | Bhalla2002 | Taxonomy-Mus musculus | 75 | 136 | [0.0, 1597.2] |
| MODEL9079740062 | Bhalla2004 | Taxonomy-Mammalia | 26 | 43 | [0.0, 20.0] |
| MODEL9080388197 | Bhalla2004 | Taxonomy-Mammalia | 12 | 20 | [0.0, 87.2] |
| MODEL9080747936 | Bhalla2004 | Taxonomy-Mammalia | 44 | 84 | [0.0, 777.8] |
| MODEL9081220742 | Bhalla2004 | Taxonomy-Mammalia | 174 | 337 | [0.0, 1399.2] |
| MODEL9085850385 | Bhalla2004 | Taxonomy-Mammalia | 54 | 94 | [0.0, 537.8] |
| MODEL9086207764 | Hayer2005 | Taxonomy-Mammalia | 273 | 570 | [0.0, 0.4] |
| MODEL9086518048 | Hayer2005 | Taxonomy-Mammalia | 273 | 580 | [0.0, 140.0] |
| MODEL9086628127 | Hayer2005 | Taxonomy-Mammalia | 14 | 28 | [0.0, 12500.0] |
| MODEL9086926384 | Hayer2005 | Taxonomy-Mammalia | 77 | 148 | [0.0, 148.4] |
| MODEL9086953089 | Hayer2005 | Taxonomy-Mammalia | 104 | 196 | [0.0, 1566.4] |
| MODEL9087255381 | Hayer2005 | Taxonomy-Mammalia | 275 | 586 | [0.0, 882.0] |
| MODEL9087474843 | Hayer2005 | Taxonomy-Mammalia | 276 | 586 | [0.0, 482.0] |
| MODEL9087766308 | Soud1999 | Taxonomy-Homo sapiens | 5 | 4 | [0.0, 0.1] |
| MODEL9087988095 | Soud1999 | Taxonomy-Homo sapiens | 5 | 4 | [0.0, 0.4] |
| MODEL9088169066 | Soud1999 | Taxonomy-Homo sapiens | 5 | 4 | [0.0, 0.3] |
| MODEL9088294310 | Soud1999 | Taxonomy-Homo sapiens | 5 | 4 | [0.0, 0.15] |
| MODEL9089491423 | Ajay_Bhalla_ 2004 | Taxonomy-Rattus | 181 | 349 | [0.0, 205.6] |
| MODEL9089538076 | Ajay_Bhalla_ 2004_PKM_MKP 3_Tuning | Taxonomy-Rattus | 182 | 353 | [0.0, 8000.0] |
| MODEL9089914876 | Ajay_Bhalla_ 2004_Feedback_ Tuning | Taxonomy-Rattus | 179 | 347 | [0.0, 5000.0] |
| MODEL9147091146 | Ajay_Bhalla_ 2007_Bistable | Taxonomy-Rattus | 71 | 132 | [0.0, 1436.8] |
| MODEL9147232940 | Ajay_Bhalla_ 2007_PKM | Taxonomy-Rattus | 59 | 104 | [0.0, 1.6] |
| MODEL9147975215 | Asthagiri2001 | Taxonomy-Cricetulus griseus | 36 | 47 | [0.0, 8000.0] |

