# OpenReview forum: "A Regime-Aware Trajectory Prediction Framework for 1000+ Systems Biology Models"
_ICML.cc/2026/Conference — ICML 2026 regular_

### Official Review · Reviewer_GHR9 · 2026-03-11

**Soundness:** 3
**Presentation:** 3
**Significance:** 3
**Originality:** 2
**Overall Recommendation:** 4
**Confidence:** 1

**Summary:**

The paper presents a large-scale benchmark of over 1,000 ODE-based systems biology models and proposes RegimeFlow, a framework for trajectory prediction. While the scale of the benchmark is impressive, the algorithmic contribution appears to be a relatively straightforward combination of existing techniques (BLR, Flow Matching, and Mamba). The novelty of the "regime-aware" component is limited as this technique have been applied by previous paper.

**Compliance With Llm Reviewing Policy:**

Affirmed.

**Final Justification:**

I thank the authors for the detailed and well-structured rebuttal. I find the additional clarifications helpful, particularly the explanation of how the regime-aware prior (via Bayesian linear regression) is integrated into the flow matching framework, and the formalization of its effect on reducing transport cost. The discussion of design choices and the role of basis functions also improves the overall clarity of the method.

I also appreciate the newly added empirical results, including the comparison to recent time-series foundation models such as Chronos and TimeMoE. These results strengthen the empirical support and better position the proposed approach relative to strong baselines.

Overall, while I still view the method as largely building on existing components, the rebuttal has clarified the contribution and provided stronger empirical evidence. Based on these improvements, I will increase my score from 3 to 4.

**Key Questions For Authors:**

See weaknesses.

**Limitations:**

yes

**Strengths And Weaknesses:**

**Strengths**

Scale and Scope: The construction of a benchmark containing 1,000+ ODE models is a significant effort that could benefit the systems biology community.

Empirical Performance: The model demonstrates improved MAE and CRPS compared to standard time-series baselines and shows high efficiency.


**Weaknesses**

1. Incremental Algorithmic Novelty: The core components of RegimeFlow are well-established in their respective fields. The use of Bayesian Linear Regression (BLR) with Fourier or polynomial bases to capture "regimes" is a textbook approach for trend fitting. Similarly, Conditional Flow Matching (CFM) and Mamba (SSM) are off-the-shelf architectures. The integration of these elements follows a standard "conditional generation" pipeline without a breakthrough in how biological constraints are enforced.

2. The "Regime-Aware Prior" is essentially a curve-fitting prior. It treats biological trajectories as general signals (oscillations, growth) rather than leveraging the specific structural properties of ODEs (e.g., conservation laws, reaction stoichiometry, or sparsity in the vector field). From an algorithmic perspective, this approach could be applied to any time-series data (finance, weather), making the "biological" aspect of the algorithm feel somewhat superficial.

3. Dependence on Quality of Basis Functions: The success of the "regime" identification depends heavily on the pre-defined basis functions (Fourier, Polynomial, etc.). This is a classic feature engineering step rather than a learned representation. If a biological system exhibits a complex or chaotic regime not easily captured by these bases, the framework's "prior" might fail, and the paper doesn't sufficiently address this limitation.

4. Missing Baselines. The evaluation lacks comparisons with recent Time-series Foundation Models (e.g., Chronos, Pandas). Since the paper emphasizes 'cross-system generalization' and 'unified framework,' it is essential to demonstrate whether the proposed regime-aware prior offers any advantage over these large-scale pre-trained models that possess zero-shot capabilities. Without such comparisons, the claim of algorithmic superiority is not fully supported, especially as the current baselines (DLinear, iTransformer) are primarily task-specific architectures rather than foundation models.

---

> ### Author Rebuttal · Authors · 2026-03-31
>
> We thank the reviewer for acknowledging the scale of our benchmark and the empirical performance of our framework. We respectfully address each concern below.
>
> **Q1: Incremental Algorithmic Novelty.**
>
> While relying on established components, our novelty lies in their integration to solve a previously unexplored problem: cross-system prediction across 1,000+ biological models. We prioritize a simple, scalable design over architectural complexity.
> 1. Beyond Standard Condition: Instead of passive embeddings, RegimeFlow uses Bayesian Linear Regression (BLR) to shape the Flow Matching source distribution. Replacing the standard Gaussian with a regime-aware prior aligns the starting manifold with biological dynamics.
> 2. Reducing Transport Distance: This non-trivial integration is formalized in Propositions 3.1-3.2 (Lines 612-686). Aligning the source distribution reduces the Expected Transport Cost under specific SNR conditions, enabling the CFM backbone to solve a significantly simpler transport problem.
> 3. Synergistic Effects: Fig. 5 (Line 392) shows our components are complementary, not redundant: the regime-aware prior minimizes source-target distance, allowing adaptive conditioning helps the backbone model the residual heterogenous dynamics.
> 4. Significant Empirical Improvement: This architectural synthesis is not an incremental tweak: it yields a 48% MSE reduction (0.023→0.012) over the backbone alone. Furthermore, it outperforms 200M-parameter Foundation Models (e.g., Chronos, TimeMoE) by 10x (see Q4 Table). Finally, it achieves highly efficient single-step inference (NFE = 1), bypassing the multi-step sampling of prior flow models (see Reviewer jTZm, Q3).
>
>
> **Q2: "Curve-Fitting Prior" and Biological Relevance.**
>
> Our method does not encode mechanistic ODE structures. This is a deliberate design choice motivated by the cross-system generalization setting. In our benchmark, systems vary widely in topology, dimensionality, and governing equations, and their mechanistic structures (e.g., conservation law, stoichiometry, interaction graphs) are system-specific and not aligned across systems. Incorporating such information would require per-system modeling and hinder transferability. Instead, we focus on regime-level biological structure, which captures macroscopic behaviors that are both biologically meaningful and shared across diverse systems, enabling a unified model.
>
> BLR is not used as a standalone curve-fitting tool, but to construct a regime-aware source distribution for flow matching. This allows biological prior knowledge to act directly at initialization, rather than as a weak auxiliary signal. We view this formulation as methodological generality rather than biological superficiality.
>
>
> **Q3: Dependence on Quality of Basis Functions.**
>
> We clarify that the predefined bases do not lock in the representation. Instead, they are used only to initialize the source prior and warm-start the generative process.  The flow-matching vector field can subsequently steer trajectories away if data exhibits uncaptured dynamics.
>
> Crucially, RegimeFlow does not strictly depend on predefined regimes, allowing trajectories lacking a clear regime can be safely assigned to the Complex (Sec. 3.3.1). Empirically, even on this subset where no strong regime prior is applied, our method outperforms the best baseline (Table 1, Line 349, MSE 0.034 vs. 0.039).
>
> While our current basis library prioritizes simplicity and transferability (Fig. 7, Effect of basis complexity), the framework is modular and can accommodate more expressive or learned bases. We agree that automated basis discovery for chaotic regimes is a valuable future direction.
>
>
> **Q4: Missing Baselines.**
>
> Thank you for the suggestion. While our initial submission included 11 diverse baselines, we have now evaluated four recent zero-shot time-series foundation models, TimeMoE[R3], Sundial[R4], TimesFM[R2], and Chronos[R1].
>
> MSE results show a clear advantage for RegimeFlow, with even the strongest foundation models performing substantially worse than Ours$^†$ across all subsets. MAE follows identical rankings and will be included in the revision.
>
> |Model|Overall|Complex|Stable|Oscillatory|Monotonic|
> |-|-|-|-|-|-|
> |TimeMoE_200M|0.120±0.004|0.099±0.013|0.118±0.003|0.138±0.005|0.138±0.012|
> |Sundial_128M|0.111±0.003|0.095±0.012|0.106±0.002|0.141±0.009|0.133±0.011|
> |TimesFM_200M|0.099±0.005|0.109±0.009|0.086±0.002|0.249±0.032|0.045±0.004|
> |Chronos_200M|0.097±0.004|0.144±0.021|0.078±0.003|0.234±0.013|0.054±0.004|
> |Ours$^†$|0.012±0.002|0.034±0.002|0.006±0.001|0.035±0.002|0.006±0.000|
>
> **Reference**
> - [R1] Ansari et al. Chronos: Learning the language of time series. TMLR, 2024.
> - [R2] Das et al. A decoder-only foundation model for time-series forecasting. ICML, 2024.
> - [R3] Shi et al. Time-moe: Billion-scale time series foundation models with mixture of experts. ICLR, 2025.
> - [R4] Liu et al. Sundial: A family of highly capable time series foundation models. ICML, 2025.

---

> > ### Author Rebuttal · Reviewer_GHR9 · 2026-04-02
> >
> > Thanks for your clear rebuttal. I am happy to increase my rating.

---

> > > ### Author Response · Authors · 2026-04-04
> > >
> > > Thank you so much for your time and positive feedback. We will make sure to carefully integrate these updates into the final manuscript.

---

### Official Review · Reviewer_icBY · 2026-03-12

**Soundness:** 3
**Presentation:** 4
**Significance:** 4
**Originality:** 3
**Overall Recommendation:** 5
**Confidence:** 4

**Summary:**

This paper proposes a regime-aware trajectory prediction framework, RegimeFlow, for forecasting future trajectories across over a thousand biological systems. Motivated by the idea that system dynamics can often be grouped into a small number of coarse regimes, the authors construct regime-aware source distributions using Bayesian linear regression (BLR) with regime-specific basis functions, and then refine these priors via conditional flow matching. The paper reports that the proposed method outperforms several baseline methods on a newly curated benchmark of biological ODE systems.

**Compliance With Llm Reviewing Policy:**

Affirmed.

**Final Justification:**

I appreciate the authors' response! Since they fully resolved my concerns regarding the benchmark dataset, I will increase my score.

**Key Questions For Authors:**

- (Regime assignment) Since the proposed method critically depends on the regime variables, could the authors clarify how $c_{pat}$ and $c_{freq}$ are assigned in practice? In particular, for a high-dimensional biological system in which different coordinates may exhibit different qualitative temporal behaviors, is the regime defined per system or per coordinate, and how is it determined?
- (Independent channel) The paper appears to process each coordinate independently. How are cross-variable interactions handled? If such interactions are not explicitly modeled, I encourage the authors to discuss this more clearly as a limitation.
- (Reproducibility) I encourage the authors to release the code and dataset. Since the construction of a large-scale biological systems benchmark is itself a potentially valuable contribution, public availability would substantially strengthen the impact of the paper. Do the authors plan to release these resources?

**Limitations:**

Please refer to the Weaknesses section above for a more detailed discussion. In summary, the primary limitations of this work are the lack of a clearly specified regime assignment procedure and the independent-channel modeling assumption, which may limit the method’s applicability to systems with strong cross-variable interactions.

**Strengths And Weaknesses:**

**Strengths**
- The motivation is clear, and the paper is generally well written. In particular, Figure 2 is effective in illustrating the overall structure of the proposed method.
- The idea of incorporating dynamical patterns into the prior is interesting and well motivated. While regime-based basis functions themselves are not entirely new, using BLR-derived predictions as the source prior for conditional flow matching is, in my view, a novel and meaningful contribution.
- The empirical evaluation is fairly extensive. The authors test the method across multiple settings, include several ablations (e.g., random masking of regime embeddings, removal of AdaLN), and report strong performance on their benchmark.

**Weakness**
- Since the assignment of dynamical regimes is central to the method, the lack of clarity on how these regime variables are obtained is a significant concern. I elaborate on this point below.
- As I understand it, the model treats each coordinate as an independent channel. However, in many biological dynamical systems, state variables are strongly coupled, and cross-variable interactions are often essential to the dynamics. The paper does not clearly explain how such dependencies are captured under this modeling choice.
- The benchmark dataset is presented as a contribution, but the paper does not clearly state whether it will be publicly released, nor does it provide a repository or data availability statement. This weakens the reproducibility and practical impact of the dataset contribution.

---

> ### Author Rebuttal · Authors · 2026-03-31
>
> **Q1 & W1: Regime assignment.**
>
> Thanks for pointing this out. We will clarify the regime assignment in the revision.
>
> **Regime label is assigned per coordinate (trajectory/variable), not per system.** For each biological model, individual variable trajectories are analyzed independently. As a result, different coordinates from the same system may be assigned different regime labels, reflecting heterogeneous dynamics within a system.
>
> The regime assignment follows a **two-step procedure**.
>
> **Step 1: Automated multi-criteria detector.**
> Each trajectory is classified using a rule-based, multi-criteria detector to assign the pattern label $c_{\text{pat}}$ and, when applicable, the frequency $c_{\text{freq}}$.
>
> - **Oscillatory:** Detected via a consensus of:
>   (i) FFT(Fast Fourier Transform)-based spectral analysis for dominant frequency components; (ii) Peak/Valley periodicity to check for regular extrema spacing; (iii) Zero-crossing regularity around the trajectory mean.
>   These detectors are combined because one alone can fail on certain cases (e.g., pulse-like signals, or truncated cycles).
> - **Monotonic:** Identified by fitting a global linear trend, verifying that the tail slope remains significantly nonzero and consistent across trajectory segments.
> - **Stable:** Identified by checking if the trajectory converges to a constant value, characterized by a near-zero tail slope and decreasing local window variance.
> - **Complex:** A "catch-all" category for trajectories that exhibit high-entropy or hybrid behaviors that do not cleanly satisfy the criteria above.
>
> If a trajectory is classified as oscillatory, $c_{\text{freq}}$ is estimated from the dominant periodic component (using FFT); otherwise $c_{\text{freq}}=0$.
>
> **Step 2: Manual validation.**
> To reduce misclassification, we perform manual inspection of trajectory plots, focusing on ambiguous cases (e.g., weak oscillations vs. damped convergence), and correct obvious inconsistencies.
>
>
> **Q2 & W2: Independent Channel Modeling and Cross-Variable Interactions.**
>
> RegimeFlow adopts a channel-independent (CI) formulation as a deliberate design choice for cross-system generalization. This is a practical choice for scalable transfer across highly heterogeneous systems, and is broadly consistent with recent time-series foundation models that emphasize robust zero-shot generalization [R1-R4].  While multivariate forecasting exists in other foundation model [R5-R6], applying it to our setting causes two challenges.
>
> First, our benchmark contains 1,050 distinct systems with entirely different interaction topologies (e.g., a yeast model vs. a neuron model). Forcing the model to learn a "universal" cross-channel dependency across mismatched systems risks negative transfer, where the model learns spurious correlations that do not generalize to new, unseen systems.
> Second, the biological system dimension spans dramatically from 1 to 1,000+, standard multivariate leads to a prohibitive context-length explosion. Thus, topology-aware relational modules is as a critical direction for future work.
>
> We already note this in the Conclusion section and will expand the discussion in the revision, highlighting topology or relational extensions as an important direction for future work.
>
> **Reference**
>
> - [R1] Ansari et al. "Chronos: Learning the language of time series." *Transactions on Machine Learning Research*, 2024.
> - [R2] Das et al. "A decoder-only foundation model for time-series forecasting." *ICML*, 2024.
> - [R3] Shi et al. "Time-moe: Billion-scale time series foundation models with mixture of experts." *ICLR*, 2025.
> - [R4] Liu et al. "Sundial: A family of highly capable time series foundation models." *ICML*, 2025.
> - [R5] Ansari et al. "Chronos-2: From univariate to universal forecasting." arXiv:2510.15821, 2025.
> - [R6] Cohen et al. "Toto: Time series optimized transformer for observability." arXiv:2407.07874, 2024.
>
>
> **Q3 & W3: Reproducibility.**
>
> The code is already available at Supplementary Material. The full benchmark will be released upon acceptance. We will also add an explicit code/data availability statement in the revised paper.

---

> > ### Author Rebuttal · Reviewer_icBY · 2026-04-02
> >
> > I thank the authors for their response to my review. The authors clarified that regime labels are assigned per coordinate using an automated detector followed by manual validation, which resolves part of my concern.
> >
> > I also continue to view the benchmark dataset as one of the paper’s most valuable contributions, and the paper itself presents it as a central contribution. For this reason, I encourage the authors to provide a concrete data release plan in the revision, including the release scope, trajectory generation pipeline, and related implementation details. If these resources are clearly documented and made available upon publication, this would substantially strengthen the paper’s reproducibility and practical impact.

---

> > > ### Author Response · Authors · 2026-04-04
> > >
> > > We thank the reviewer for recognizing the benchmark dataset as a central contribution of this work. We fully agree that a concrete release plan is essential for reproducibility and practical impact.
> > >
> > > **Our commitment:** upon acceptance, our complete suite of resources will be formally released across Hugging Face Datasets (for large-scale data) and a public GitHub repository (for all codebases).
> > >
> > > ## 1. Release Scope & Data Organization.
> > > We will open-source the full benchmark of 1,050 biological dynamical systems. The dataset is organized as follows:
> > >
> > > ```
> > > SysBio-Traj/
> > > ├── README.md                           # Detailed documentation
> > > ├── SysBio-Traj_index.csv               # Master metadata index
> > > ├── scripts/			     # Generation utilities
> > > │   └── simulator.py
> > > └── Data/
> > >     ├── BIOMD0000000013/
> > >     │   ├── Poolman2004.csv             # Trajectory data
> > >     │   ├── Poolman2004.xml             # Model source
> > >     │   ├── initial_conditions.json     # Initial Condition
> > >     │   └── Poolman2004_conditions.json # Regime labels
> > >     ├── BIOMD0000000144/
> > >     │   ├── Calzone2007.csv
> > >     │   ├── Calzone2007.xml
> > >     │   ├── initial_conditions.json
> > >     │   └── Calzone2007_conditions.json
> > >     └── ...
> > > ```
> > >
> > >
> > >   (i) **Trajectory Data (.csv):** simulated trajectory with 512 uniformly sampled time points across all system species (e.g., C, M, X). An illustrative snippet of the data format is shown below:
> > >   ```
> > > time,C,M,X
> > > 0.0,0.23901494118204017,0.008987225703866234,9.08824000739145e-05
> > > 0.11741682974559686,0.2416656577021649,0.009177260272872067,9.2840083348309e-05
> > > ...
> > >   ```
> > >   (ii) **Model Source (.xml):** Systems Biology Markup Language [R1] file, which is a community-standard model description format encoding the mathematical structure of each biological system, including species, reactions, and kinetic laws.
> > >
> > >   (iii) **Initial Conditions (.json):** Calibrated initial states for reproduction:
> > > ```
> > > {
> > >   "initial_time": 0.0,
> > >   "initial_conditions": {
> > >     "C": 0.23901494118204017,
> > >     "M": 0.008987225703866234,
> > >     "X": 9.08824000739145e-05
> > >   }
> > > }
> > > ```
> > >
> > >   (iv) **Regime Metadata (.json):** Per-trajectory regime labels, defined as:
> > >
> > >   ```
> > > {
> > >   "C": {
> > >     "trajectory_type": 3,
> > >     "trajectory_type_name": "oscillation",
> > >     "bounds": [0.1895, 0.5814],
> > >     "period": 214.0
> > >   },
> > >   ...
> > > }
> > >   ```
> > >   where `trajectory_type` follows our six-class taxonomy (0: complex, 1: increasing-stable, 2: decreasing-stable, 3: oscillation, 4: monotonic increasing, 5: monotonic decreasing), `bounds` records the per-trajectory min/max values, and `period` is expressed in index units.
> > >
> > > > (Note: The top-level SysBio-Traj_index.csv maps each model_id to its time boundaries for programmatic access.)
> > >
> > > `SysBio-Traj_index.csv`
> > >   ```
> > > model_id,model_name,time_start,time_end,time_span
> > > BIOMD0000000013,Poolman2004,0,0.4,0.4
> > > BIOMD0000000144,Calzone2007,0,300,300
> > > ...
> > >   ```
> > >
> > > ## 2. Trajectory Generation Pipeline.
> > > Trajectories are generated by numerically simulating each SBML model using the Tellurium library [R2]. The core generation procedure follows steps:
> > >
> > >   (i) **Model Parsing:** The mathematical structure of the biological system is first parsed and loaded directly from its standardized SBML (`.xml`) file. Notably, while the SBML format inherently encapsulates the complete dynamical context (equations, parameters, and default initial conditions), we manually calibrated the initial states for a specific subset of models (saved in initial_conditions.json). This curation ensures the generation of high-quality and dynamically rich trajectories.
> > >
> > >   (ii) **Numerical Simulation:** The system dynamics are integrated over a specified time span (from start_time to end_time). As described in the manuscript, the simulation is configured to uniformly sample exactly 512 time points to ensure consistency for training and evaluation.
> > >
> > >   (iii) **Structured Formatting:** Finally, the time steps and species trajectories are merged into a table. We use the species IDs as column headers (e.g., time, C, M, X) and save the generated trajectory directly as a `.csv` file.
> > >
> > >   The complete and reproducible generation scripts will be formally released alongside our codebase.
> > >
> > > ## 3. Reproducibility & Implementation.
> > > (i) As provided in our Supplementary (`.zip`), complete implementations, hyperparameter configurations, and training/evaluation pipelines for RegimeFlow and all baselines are already available.
> > >
> > > (ii) Comprehensive metadata for all 1,050 systems is documented in Appendix D and will be integrated into the codebase.
> > >
> > > We believe this formalized plan ensures that SysBio-Traj serves as a robust foundation for future research in AI for Systems Biology.
> > >
> > > **Reference**
> > >   - [R1] Hucka et al. "The systems biology markup language (SBML): a medium for representation and exchange of biochemical network models." Bioinformatics, 2003.
> > >   - [R2] Medley et al. "Tellurium notebooks—an environment for reproducible dynamical modeling in systems biology." PLoS Computational Biology, 2018.

---

### Official Review · Reviewer_TQQz · 2026-03-12

**Soundness:** 3
**Presentation:** 3
**Significance:** 3
**Originality:** 3
**Overall Recommendation:** 5
**Confidence:** 4

**Summary:**

The paper introduces RegimeFlow for long-horizon testing for biological dynamical systems. The authors have also curated a large-scale benchmark of over 1000 ODE based systems biology model. The key innovation is a domain aware prior into conditinional flow matching that is backed with theoretical results. The prior used in conditional flow matching is built from a projection onto a regime specific basis function.

**Compliance With Llm Reviewing Policy:**

Affirmed.

**Final Justification:**

The main skepticism (parameter dependent regime switching and transiet structure handling) has been addressed by the authors and the clarification on the actual Bayesian structure used strengthened my position as to the strengths of the paper.

**Key Questions For Authors:**

#### Questions/Clarifications

1. Can the authors do a validation on a test set that doesn't quite fit the bucketization for e.g. an early oscillatory regime that then stabilizes?

2. Can the authors reframe the theory? The two propositions are mostly sophisticated remarks and the careful benchmarking shows empirical validity but it's not a full blown theory unless the authors derive some computable estimates that then drives their predictions (it seems it can be done within the framework).

3. Clarify where the gains are coming from -- if the covariance regularization collapses it back to your isotropic Gaussian, then what is that covariance doing (it is not used in the best model in the paper)? Are the gains coming from mean alignment? If so, this is completely understandeable but the theory sections makes it seem like the full Bayesian geometry is used. A line or so in the main text stating this would strengthen the paper.

**Limitations:**

Yes

**Strengths And Weaknesses:**

#### Strengths

1. Novelty is in domain aware integration into purely data drive state of the art ML methods and the domain problem itself is important and broadly applicable -- long horizon forecasting in systems biology.

2. Benchmark Dataset: This is the strongest contribution of the paper. This is a large carefully structured dataset!

3. The paper and the set up is easy to understand. Infact the regime specific prior construction is very interpretable and not black boxy.

4. The result section is strong, the claims are well validated and the two propositions support the message.

#### Weaknesses

1. The heterogeneity in the data is bucketed using macroscopic information and this is used to construct the priors. So the claim isn't that the paper deals with heterogeneity in biological dynamical systems but a more narrower: if the heterogeneity is classified according to the proposed buckets, then a structured prior beats out uniformed ones. The paper shows the latter to be true but this becomes more narrower than what the abstract seems to imply.

2. parameter dependent switching: Nonlinear biological dynamical systems show different qualitative behavior dependent on the parameters and depending on how you parameterize them they may fall in one bucket or the other. So parameter-sensitive transitions are completely ignored in this framework.

3. The theory is super standard, its good but maybe a comprehensive remark can capture it? I think the theory section can be strengthened by including some computable estimates (there is a qualitative gap discussion in the appendix) that can then help guide the predictions.

4. I could be wrong here, but the authors in 3.3.3 say
> We replace the anisotropic predictive covariance with
> an isotropic approximation to improve numerical stability

and the authors don't induce the BLR induced variance. So I am confused as to where the  BLR vs isotropic theory is informing the actual implementation.

---

> ### Author Rebuttal · Authors · 2026-03-31
>
> **W1: Scope of Heterogeneity**
>
> While we leverage regime structures to mediate heterogeneity, we clarify that our claims are significantly broader than a simple "if-classified" result:
> 1. Bucket-Free Performance: Our model does not strictly rely on predefined buckets. As shown in Table 1 (Line 342), it outperforms all baselines even when regime information is entirely withheld during training and inference, proving it is a general generative model rather than a classification-dependent approach.
> 2. Robustness to Partial Regime Information: Fig. 6 further shows that RegimeFlow remains highly competitive under varying levels of regime masking during training and across different regime-conditioning scenarios at test time, indicating limited reliance on precise or complete regime information.
> 3. Addressing Unclassifiable Dynamics: To ensure our framework is not limited to simple predefined categories, we include a "Complex" regime. This category acts as a catch-all for heterogeneous dynamics that do not fit the monotonic, oscillatory, or stable patterns. Our model beats all baselines in this category, showing that the architecture is generalizable to out-of-bucket systems.
> 4. Intra-Bucket Heterogeneity: Significant heterogeneity exists even within a single bucket (e.g., circadian vs. calcium oscillators differ in frequency and amplitude). Our framework successfully captures these dynamics, demonstrating that while the regime provides a coarse prior, the model performs the heavy lifting of learning specific vector fields.
>
> **W2: Parameter-Dependent Regime Switching**
>
> We thank the reviewer for this insightful point. We clarify that regime labels are assigned at the trajectory level, not the system level. Trajectories from the same system under different parameters can exhibit and be assigned to different regimes (e.g., stable vs. oscillatory), enabling our model to capture parameter-dependent regime shifts. Our current framework does not explicitly study the ODE parameter sensitivity as we primarily focus on macroscopic forecasting across diverse systems. We will incorporate analysis of parameter-dependent regime transitions in future work.
>
> **Q1: Validation on Transient Regimes**
>
> Table 1 details our performance on the Complex regime, designed to capture trajectories defying canonical patterns.
>
> To address concerns about trajectories defying a single regime (e.g., transient or mixed), we evaluate representative examples under a long-horizon extrapolation setting (96 → 416).
> As shown below, RegimeFlow consistently outperforms the strong TSFlow baseline across these cases:
>
> |Case|Example|Ours$^†$(MSE/MAE)|TSFlow$^†$(MSE/MAE)|
> |-|-|-|-|
> |Complex Oscillation|Golomb2006|0.024/0.092|0.135/0.275|
> | Oscilaltory$\to$Stable|Sharp2013|0.035/0.116|0.046/0.159|
> |Decrease Oscilation|Radulescu2008|0.051/0.151|0.086/0.216|
>
> The predicted trajectories are available at: https://figshare.com/s/fcba7c6cddfe005ffef9.
>
> **W3 & Q2: Reframing the Theory and Computable Estimates**
>
> We appreciate the constructive critique. In the revision, we will reframe the theory accordingly and introduce a  **Regime significance score** ($\hat\Delta$). This acts as a computable estimate offering practical guidance for reliable predictions.
>
> Specifically, we define the computable estimate:
> $\hat\Delta(y_p)=||\mu _ {\text{prior}}(y _ p)||^2/\hat{\sigma} _ \epsilon^2-\mathrm{Tr}(\Sigma _ {\text{design}})$, where $\mu _ {\text{prior}}(y _ p)$ is the BLR-induced mean,
> $\hat{\sigma} _ \epsilon^2$ is the estimated observation noise variance, and $\Sigma _ {\text{design}}$ is the covariance of the basis geometry.
>
> This yields a practical test-time rule: when $\hat\Delta>0$, the regime-aware prior is expected to be beneficial; when $\hat\Delta \le 0$, it indicates lower confidence / possible regime mismatch, in which case regime-agnostic inference is favored or richer basis may be needed.
>
> **W4 & Q3: Source of Gains**
>
> We thank the reviewer for this nuanced observation regarding the transition from anisotropic to isotropic covariance.
>
> 1. In principle, utilizing the full Bayesian geometry (mean and anisotropic covariance) is beneficial and empirically competitive. As shown in Table 3 (Line 721), the full BLR geometry achieves an MSE of 0.014, outperforming the best baseline, TSFlow (0.016).
> 2. In practice, propagating dense, anisotropic covariance matrices across long horizons ($T \gg L$) can lead to numerical instability. Replacing this with an isotropic approximation ($\sigma^2 I$) stabilizes the generation process, further improving MSE to 0.012.
> 3. From the $W_2$ decomposition (App. A.2), the transport cost separates into mean and covariance terms. Thus, mean alignment alone reduces transport distance, independent of covariance choice.
>
> The primary gain in practical model comes from reducing the transport distance via mean alignment, while the isotropic approximation ensures numerical stability. We will clarify this point in the revision.

---

> > ### Author Rebuttal · Reviewer_TQQz · 2026-04-02
> >
> > Thank you for the responses! This answers every doubt I had. I will raise my score

---

> > > ### Author Response · Authors · 2026-04-04
> > >
> > > Thank you for your positive feedback. We are really glad we could clear up your concerns. We will incorporate your constructive suggestions into the revision. Thank you again for your time.

---

### Official Review · Reviewer_jTZm · 2026-03-13

**Soundness:** 3
**Presentation:** 3
**Significance:** 4
**Originality:** 4
**Overall Recommendation:** 5
**Confidence:** 3

**Summary:**

This paper proposes RegimeFlow, a framework for cross-system trajectory prediction of biological dynamical systems. The core idea is to replace the standard Gaussian source distribution in conditional flow matching with a regime-aware prior constructed via Bayesian linear regression over structured basis functions (exponential decay, Fourier, polynomial).

**Compliance With Llm Reviewing Policy:**

Affirmed.

**Final Justification:**

The author’s detailed rebuttal reinforced my prior assessment, so I decided to maintain a positive score and increase my confidence.

**Key Questions For Authors:**

1. How to acquire biological state labels for truly unseen systems during testing?

2. Can you provide some test performance data on real experimental data?

3. Can you provide detailed information comparing training time and computational cost? These details will help improve the reproducibility of the paper.

**Limitations:**

yes

**Strengths And Weaknesses:**

**Strengths:**
1. The benchmarks for 1050 biological systems based on ordinary differential equations  are a valuable community contribution.

2. The experimental evaluations are comprehensive and in-depth, with ablation experiments on prior design, adaptive layer normalization, and hyperparameters being particularly valuable.

3. The theoretical analysis and proofs in Propositions 3.1, 3.2, and Appendix B.1 further enhance the persuasiveness of this paper.

**Weakness:**
1. This paper synthesizes trajectories from known ODE equations, which have controllable noise; actual laboratory data sampling may be irregular, noisy, or have coefficients, and this benchmark may overestimate actual performance.

2. During testing, it was assumed that state labels (stable, oscillating, monotonic) were known. However, in actual biological discoveries, state classification may be uncertain or even impossible to obtain. The relevant experimental section in Figure 6 partially addresses this issue, but it does not completely eliminate my concerns.

---

> ### Author Rebuttal · Authors · 2026-03-31
>
> **Q1&W2: Acquiring Regime Labels for Unseen Systems**
>
> We thank the reviewer for this important concern. RegimeFlow is designed to leverage regime information when available, but does not depend on it, remaining state-of-the-art even in regime label-free settings. We address this from three aspects:
>
> 1. Regime-Agnostic Inference (no labels required): RegimeFlow does **not require regime labels for deployment**. Regime conditioning is an **optional enhancement**. We explicitly evaluate this setting in Table 1 (“w/o Regime Condition”, Line 342), where RegimeFlow still outperforms all baseline in fully regime label-free setting.
>
> 2. Acquisition via Domain Knowledge (when available): In many biological applications, high-level dynamical behaviors are known even when the underlying mechanistic model is not (e.g., cell cycle systems are oscillatory, homeostatic processes are stable). Such qualitative priors are routinely used in systems biology to guide modeling design [R1, R2]. RegimeFlow can incorporate this coarse-grained knowledge when available.
>
>  3. Robustness to Partially Available Labels: We evaluate the setting where regime information is only partially available (e.g., for a subset of systems or trajectories). RegimeFlow remains robust, as shown in Fig. 6, Line 405.
>
>
> **Reference**
>
> - [R1] Tyson et al. "Sniffers, buzzers, toggles and blinkers: dynamics of regulatory and signaling pathways in the cell." *Current opinion in cell biology* 15.2 (2003): 221-231.
> - [R2] Andrews et al.. "Design patterns of biological cells." *BioEssays* 46.3 (2024): 2300188.
>
> **Q2&W1: Performance on Real Experimental Data**
>
> We acknowledge that evaluating on real labotory data is an important direction and remains to be explored. While collecting sufficiently large and diverse real biological dataset for comprehensive benchmarking is challenging within the short rebuttal period, we conducted additional experiments on our benchmark to approximate realistic conditions:
>
> - **Measurement Noise:** We perturbed the observed trajectories with additive Gaussian noise at different scales of 0.1 and 0.3 (i.e., $x + \text{scale} \times \mathcal{N}(0, 1)$), to simulate measurement uncertainty commonly observed in laboratory data.
> - **Irregular Sampling:** We randomly subsampled the observation points (e.g., **Irregular-24** retains only 24 random observations out of 96), mimicking incomplete and non-uniform data collection.
>
> | Setting      | Ours$^†$(MSE) | TSFlow$^†$(MSE) | Improvement |
> | ------------ | ------------- | --------------- | ----------- |
> | Clean        | 0.012±0.002   | 0.016±0.004     | 25%         |
> | Noisy-0.1    | 0.041±0.004   | 0.045±0.003     | 9%          |
> | Noisy-0.3    | 0.059±0.002   | 0.067±0.014     | 12%         |
> | Irregular-48 | 0.038±0.005   | 0.054±0.009     | 30%         |
> | Irregular-24 | 0.080±0.004   | 0.095±0.012     | 16%         |
>
> RegimeFlow consistently outperforms TSFlow under both noise and irregular sampling. Full results will be included in the appendix.
>
> **Q3: Training Time and Computational Cost**
>
> We provide our computational profiling below and will include the full table in the appendix.
> Point-forecasting methods offer fast training and inference but yield inferiro predictive performance, whereas probabilistic methods (e.g., TSFlow) achieve better performance with higher computational cost.
>
> RegimeFlow incurs moderate training cost, but its key advantage lies in inference efficiency. Specifically, it requires only a **single function evaluation** (NFE = 1; Fig. 7/10, Line 400/802) to generate accurate forecasts. In contrast, diffusion/flow-based methods typically require 16–100 NFEs, leading to substantially slower inference.
>
> | Model| Params | Training Time | Inference Time | NFE   |
> |-|-|-|-|-|
> | ***Point Forecasting***         |        |               |                |       |
> | DLinear  | 49.7 K | ~0.3 h        | 0.1 ms         | —     |
> | iTransformer                    | 4.9 M  | ~0.5 h        | 0.9 ms         | —     |
> | PatchTST                        | 4.7 M  | ~0.6 h        | 0.8 ms         | —     |
> | TimeMixer                       | 133 K  | ~0.5 h        | 3.1 ms         | —     |
> | TimeXer                         | 11.5 M | ~0.5 h        | 2.1 ms         | —     |
> | SMamba                          | 3.0 M  | ~0.5 h        | 2.3 ms         | —     |
> | BiMamba4TS                      | 3.1 M  | ~1.5 h        | 2.5 ms         | —     |
> | NSformer                        | 2.0 M  | ~5 h          | 3.0 ms         | —     |
> | ***Probabilistic Forecasting*** |        |               |                |       |
> | CSDI$^†$                        | 413 K  | ~8 h          | 90.8 ms        | 50    |
> | TSDiff$^†$                      | 946 K  | ~10 h         | 552.6 ms       | 100   |
> | TSFlow$^†$                      | 1.9 M  | ~6 h          | 71.9 ms        | 16    |
> | **Ours$^†$**                    | 1.8 M  | ~7 h          | **7.7 ms**     | **1** |

---

> > ### Author Rebuttal · Reviewer_jTZm · 2026-04-04
> >
> > Thank you for the detailed rebuttal; this reinforced my prior assessment, so I decided to maintain a positive score and increase my confidence.

---

> > > ### Author Response · Authors · 2026-04-04
> > >
> > > Thank you for the positive feedback. We will incorporate your suggestions in the revision. We appreciate your time and helpful comments.

---

### Decision · Program_Chairs · 2026-04-30

**Decision:**

Accept (regular)

**Comment:**

This paper introduces a regime-aware trajectory prediction approach to forecast the trajectories of biological dynamical systems. It uses Bayesian linear regression for the regime-aware prior to combine with the conditional diffusion model. It tests the method by establishing a benchmark using ODE-based systems biology models.

The reviewers like the system biology trajectory benchmark. After rebuttal, the authors have also substantially strengthened the benchmark side. The proposed method works well on the benchmark. The salient limitation is that the benchmark only focuses on ODE trajectories.